# Fast and Regret Optimal Best Arm Identification: Fundamental Limits and Low-Complexity Algorithms

**Qining Zhang**
University of Michigan, Ann Arbor
`qiningz@umich.edu`

**Lei Ying**
University of Michigan, Ann Arbor
`leiying@umich.edu`

## Abstract

This paper considers a stochastic Multi-Armed Bandit (MAB) problem with dual objectives: (i) quick identification and commitment to the optimal arm, and (ii) reward maximization throughout a sequence of $T$ consecutive rounds. Though each objective has been individually well-studied, i.e., best arm identification for (i) and regret minimization for (ii), the simultaneous realization of both objectives remains an open problem, despite its practical importance. This paper introduces *Regret Optimal Best Arm Identification* (ROBAI) which aims to achieve these dual objectives. To solve ROBAI with both pre-determined stopping time and adaptive stopping time requirements, we present an algorithm called EOCP and its variants respectively, which not only achieve asymptotic optimal regret in both Gaussian and general bandits, but also commit to the optimal arm in $\mathcal{O}(\log T)$ rounds with pre-determined stopping time and $\mathcal{O}(\log^2 T)$ rounds with adaptive stopping time. We further characterize lower bounds on the commitment time (equivalent to the sample complexity) of ROBAI, showing that EOCP and its variants are sample optimal with pre-determined stopping time, and almost sample optimal with adaptive stopping time. Numerical results confirm our theoretical analysis and reveal an interesting "over-exploration" phenomenon carried by classic UCB algorithms, such that EOCP has smaller regret even though it stops exploration much earlier than UCB, i.e., $\mathcal{O}(\log T)$ versus $\mathcal{O}(T)$, which suggests over-exploration is unnecessary and potentially harmful to system performance.

## 1 Introduction

The stochastic Multi-Armed Bandit (MAB) problem [1], which models a wide range of applications including online recommendations [2, 3, 4], job assignments [5, 6, 7], clinical trials [8, 9], and etc, is a sequential decision-making process between an agent and an environment which consists of a number of arms (actions). In this paper, we will use "arm" and "action" interchangeably. Most existing studies, say [10, 11, 12, 13, 14, 15], formulate a regret minimization problem to study the bandit model, where the agent aims to maximize the cumulative reward through interacting with the environment for a number of consecutive rounds. The well-known UCB algorithm [10] and its variants [12] are among the most popular bandit algorithms for regret minimization, which exhibit outstanding performances both theoretically and empirically in bandit models. Moreover, state-of-the-art reinforcement learning algorithms, e.g., [16, 17], are also motivated by the essence of UCB which sets up confidence intervals to encourage exploration. However, the UCB algorithms do not commit to a single arm. Instead, they continue to switch among all arms based on reward signals received. Since one of the ultimate goals of the MAB model is to learn the optimal arm, it would be ideal if the algorithm would commit to an arm quickly without sacrificing the regret. In fact and in practice, a number of applications such as occupational decisions [18], medicine release and pandemic control [19, 20], and long-term investments [21], require or prefer quick commitment to an action instead of continuous exploration. This motivated us to consider an MAB problem with dual

| Algorithm | Regret | Setting | Optimality | Commitment | Confidence |
|---|---|---|---|---|---|
| UCB [22] | $\frac{2+o(1)}{\Delta}\log T$ | Gaussian | Yes | $T$ | N/A |
| KL-UCB [12] | $\frac{\Delta+o(1)}{\mathsf{KL}(\mu_2,\mu_1)}\log T$ | General | Yes | $T$ | N/A |
| TS [15] | $\frac{\Delta+o(1)}{\mathsf{KL}(\mu_2,\mu_1)}\log T$ | General | Yes | $T$ | N/A |
| BAI-ETC [22] | $\frac{4+o(1)}{\Delta}\log T$ | Gaussian | No | $\mathcal{O}(\log T)$ | $\tilde{\mathcal{O}}(T^{-1})$ |
| DETC [23] | $\frac{2+o(1)}{\Delta}\log T$ | Gaussian | Yes | $\Omega(\log^2 T)$ | $\mathcal{O}(T^{-1})$ |
| UCB$_\alpha$ [24] | $\frac{2\alpha^2+o(1)}{\Delta}\log T$ | Gaussian | No | $\mathcal{O}(\log T)$ | $\tilde{\mathcal{O}}(T^{-1})$ |
| EOCP **(Ours)** | $\frac{2+o(1)}{\Delta}\log T$ | Gaussian | Yes | $\mathcal{O}(\log T)$ | $\mathcal{O}\left(T^{-1}\right)$ |
| EOCP-UG **(Ours)** | $\frac{2+o(1)}{\Delta}\log T$ | Gaussian | Yes | $\mathcal{O}(\log^2 T)$ | $\mathcal{O}\left(T^{-1}\right)$ |
| KL-EOCP **(Ours)** | $\frac{\Delta+o(1)}{\mathsf{KL}(\mu_2,\mu_1)}\log T$ | General | Yes | $\mathcal{O}(\log T)$ | $\mathcal{O}\left(T^{-1}\right)$ |
| LB(pre-determined) | $\frac{2+o(1)}{\Delta}\log T$ | Gaussian | $\mathcal{O}(\log^c T)$ | $\mathcal{O}(\log T)$ | $\mathcal{O}(T^{-1})$ |
| LB(adaptive) | $\frac{2+o(1)}{\Delta}\log T$ | Gaussian | $\mathcal{O}(\log^c T)$ | $\mathcal{O}(\log^{2-c} T)$ | $\mathcal{O}(T^{-1})$ |

Table 1: Caparison under 2-armed bandits where $\Delta = |\mu_1 - \mu_2|$ is the expected reward difference, $\mathsf{KL}(\mu_2,\mu_1)$ is the Kullback-Leibler divergence between reward distributions, and $\alpha > 1$. EOCP and KL-EOCP use a pre-determined stopping time and require the knowledge of $\Delta$, while EOCP-UG uses an adaptive stopping time. Both LBs represent the commitment time lower bound under Gaussian bandits for regret optimal algorithms with $\mathcal{O}(\log^c T)$ finite-time regret violation and $\mathcal{O}(T^{-1})$ confidence.

objectives: (i) quick identification and commitment to the optimal arm, and (ii) minimization of the cumulative regret throughout a sequence of rounds.

**Regret Optimal Best Arm Identification:** The lack of commitment in traditional regret minimization formulation motivates us to propose a new viewpoint towards online decision-making called *Regret Optimal Best Arm Identification* (ROBAI), which intends to manage two goals at the same time: minimizing regret while committing to an arm quickly. Specifically, it is ideal for the agent to quickly commit to the optimal arm which has the highest expected reward while minimizing the exploration regret. To solve ROBAI, we need to answer three fundamental questions: (1) how should the learner explore arms while maintaining a low-regret performance (**exploration strategy**)? (2) when should the learner stop exploration and commit to an arm (**stopping criterion**)? and (3) which arm to commit to when the exploration ends (**action selection strategy**)? All three components need to be designed together to make the algorithm most efficient. The fundamental question this paper addresses is: Can we design an efficient algorithm that is both regret optimal and identifies the optimal arm quickly, and what are the fundamental limits of such algorithms?

**Connection to Best Arm Identification:** One approach people may take to solve ROBAI is the Best Arm Identification (BAI) algorithm, which studies how to learn the optimal arm with the minimum number of samples (rounds) and then commit to the selected arm. However, since BAI focuses purely on sample complexity (or commitment time), the algorithms for BAI explore sub-optimal arms too aggressively and too often, leading to large regret. As shown in [22], the regret is asymptotically at least twice as large as the regret under the classic UCB algorithm. Modifications shown in [23] may lead to better regret performance, but they also make the algorithm too complicated to find the optimal arm quickly. Moreover, these algorithms adapted from BAI often exhibit poor empirical regret performances as shown in our numerical experiments of Fig. 1.

**Our Contributions:** We propose an algorithm called EOCP, which stands for *Explore Optimistically then Commit Pessimistically*, to solve the ROBAI problem. It first uses an optimistic modified version of the classic UCB algorithm to explore arms with a slightly larger exploration function, and then it commits to the optimal arm candidate according to a pessimistic LCB algorithm when the exploration ends, by selecting the arm that has the largest lower confidence bound. The exploration and action identification strategies are respectively motivated by the inflated bonus trick [24] and the principle of pessimism from the literature of offline bandits [25, 26, 27] and offline reinforcement learning [28, 29, 30]. Our greatest contributions include designing new stopping rules with both pre-determined stopping time (vanilla EOCP) and adaptive stopping time (the EOCP-UG variant), which probably balance the trade-off between regret minimization and action identification. We theoretically

show that both algorithms are asymptotically regret optimal in Gaussian bandit models. Moreover, EOCP and EOCP-UG algorithms commit to the optimal arm in $\mathcal{O}(\log T)$ and $\mathcal{O}(\log^2 T)$ number of rounds respectively, both with $\mathcal{O}(T^{-1})$ confidence. We further characterize the fundamental commitment time (sample complexity until commitment) limits of action identification for regret optimal algorithms, which shows that $\mathcal{O}(\log T)$ number of samples is always required with predetermined stopping time, and $\mathcal{O}(\log^{2-c} T)$ number of samples is required with adaptive stopping time if the finite-time regret of the algorithm does not exceed its asymptotic regret rate by $\mathcal{O}(\log^c T)$. This shows that EOCP is sample optimal and EOCP-UG is nearly sample optimal. We also propose an improved algorithm called KL-EOCP to achieve regret optimality in general bandits beyond Gaussian rewards. To the best of our knowledge, KL-EOCP is the first algorithm that not only achieves asymptotic regret optimality in general bandit models but also commits to the optimal arm in $\mathcal{O}(\log T)$ rounds, which also matches the commitment time lower bound. The more detailed comparison between existing algorithms and our proposed algorithms with lower bounds is summarized in Tab. 1. Numerical experiments confirm the superiority of our proposed algorithm and show an interesting "over-exploration" phenomenon carried by UCB algorithms. As shown in Fig. 1, our EOCP algorithm reduces more than 20% of regret compared to the vanilla UCB algorithm by finding and committing to the optimal arm early.

## 2 Preliminaries

**Stochastic Multi-armed Bandits:** A stochastic multi-armed bandit problem is an online decision-making process between an agent and an environment for a number of $T$ consecutive rounds. At each round $t \in \{1, 2, \cdots, T\}$, the agent can choose an action $A_t$ among a set of actions (arms) with cardinality $A$, denoted by $\mathcal{A} = \{1, 2, \cdots, A\}$, to interact with the environment. Each action $a$ is associated with a probability distribution $\nu_a$ and we denote the set of distributions as $\boldsymbol{\nu} = \{\nu_1, \cdots, \nu_A\}$ with respective expectations $\boldsymbol{\mu} = \{\mu_1, \cdots, \mu_A\}$ which is unknown to the agent a priori. The expectations are assumed to be bounded so without loss of generality, we have $\mu_a \in [0, 1]$ for any action $a$. After the agent chooses an action, say action $A_t$ at round $t$, it will observe an independent reward $r_t$ which is sampled from the distribution $\nu_{A_t}$ associated with the action $A_t$ that it chooses. We define the optimal action $a^*$ to be the action which has the highest expected reward, i.e., $\mu_{a^*} = \arg\max_{a \in \mathcal{A}} \mu_a$, and for simplicity, we assume it is unique. Let $\Delta_a = |\mu_{a^*} - \mu_a| \in (0, 1]$ to be the expected reward gap between the optimal action and a sub-optimal action $a$, and we use $\Delta_{\min} = \min_{a:\Delta_a > 0} |\mu_{a^*} - \mu_a|$ to denote the minimum reward gap among all sub-optimal actions.

**Regret:** The goal of the agent is to maximize the expected cumulative reward from the total $T$ rounds of interactions with the environment, i.e., to maximize $\mathbb{E}[r_1 + r_2 + \cdots + r_T]$, where the expectation is taken over all randomness. The performance of any bandit algorithm Alg chosen by the agent is usually measured by the cumulative *Regret* up to round $T$ defined as follows:

$$\mathsf{Reg}_{\boldsymbol{\mu}}^{\mathsf{Alg}}(T) = T\mu_{a^*} - \mathbb{E}_{\boldsymbol{\mu}}\left[\sum_{t=1}^{T} r_t\right],$$

where the subscript $\boldsymbol{\mu}$ denotes the bandit instance represented by the reward expectations. Maximizing reward is equivalent to minimizing the cumulative regret. For most of the algorithms to achieve this goal, the agent will make action-choosing decisions based on two statistics maintained and updated at each round for every action: the empirical mean $\bar{r}_t(a)$ and the number of pulls $N_t(a)$ in previous rounds. They are defined as:

$$N_t(a) = \sum_{k=1}^{t} \mathbb{1}_{A_t=a}, \quad \bar{r}_t(a) = \frac{1}{N_t(a)} \sum_{k=1}^{t} r_t \mathbb{1}_{A_t=a}.$$

The theoretical regret limit of any algorithm Alg is studied and characterized in [13], which shows:

$$\liminf_{T \to \infty} \frac{\mathsf{Reg}_{\boldsymbol{\mu}}^{\mathsf{Alg}}(T)}{\log T} \geqslant \sum_{a:\Delta_a>0} \frac{\Delta_a}{\mathsf{KL}(\nu_a, \nu_{a^*})}, \tag{1}$$

where $\mathsf{KL}(\cdot, \cdot)$ denotes the Kullback–Leibler divergence between two distributions. We call an algorithm Alg regret optimal (asymptotically) if the asymptotic regret performance of Alg achieves this lower bound. Therefore, whether an algorithm is asymptotically regret optimal will depend on

the distributions $\boldsymbol{\nu}$ of the rewards. Specifically, for Gaussian bandits, the KL divergence between distributions $\mathcal{N}(\mu_a, 1)$ and $\mathcal{N}(\mu_{a'}, 1)$ is simply $(\mu_a - \mu_{a'})^2/2$. So the asymptotic regret rate lower bound in the RHS for Gaussian bandits would be $2\sum_{a:\Delta_a>0} \Delta_a^{-1}$.

**Commitment:** In ROBAI, commitment to a single action $\hat{a}$ (ideally, the optimal action) is required. After a stopping time $T_c$, the agent will not be allowed to switch actions and will commit to the same action until the end. We consider two categories of commitment: the pre-determined stopping-time setting and the adaptive stopping time setting. The pre-determined stopping time requires $T_c$ to be pre-specified before the first round of interaction, while the adaptive stopping criterion requires $T_c$ to be a stopping time measurable to the natural filtration. How quickly the agent commits is measured by the *Sample Complexity until Commitment* (also called commitment time) which is the expected number of exploration rounds, i.e., $\mathsf{SCC}_{\boldsymbol{\mu}}^{\mathsf{Alg}}(T) = \mathbb{E}_{\boldsymbol{\mu}}[T_c]$. The accuracy of identifying the optimal action is measured by the confidence, which is the probability that the agent commits to a sub-optimal action, i.e., $\mathbb{P}_{\boldsymbol{\mu}}(\hat{a} \neq a^*)$. An ideal algorithm should minimize the commitment time while maintaining confidence lower than a pre-specified threshold.

**ROBAI Problem Formulation:** We use $\Pi_{\mathrm{RO}}$ to denote the class of regret optimal algorithms which commit to the optimal action with a confidence lower than $\mathcal{O}(T^{-1})$, i.e.,

$$\Pi_{\mathrm{RO}} = \left\{ \mathsf{Alg} \,\middle|\, \limsup_{T\to\infty} \frac{\mathsf{Reg}_{\boldsymbol{\mu}}^{\mathsf{Alg}}(T)}{\log T} \leqslant \sum_{a:\Delta_a>0} \frac{\Delta_a}{\mathsf{KL}(\nu_a, \nu_{a^*})} \text{ and } \mathbb{P}_{\boldsymbol{\mu}}(\hat{a} \neq a^*) = \mathcal{O}\left(T^{-1}\right) \right\}.$$

ROBAI aims to design a regret optimal algorithm $\mathsf{Alg} \in \Pi_{\mathrm{RO}}$ to minimize the commitment time:

$$\min \mathsf{SCC}_{\boldsymbol{\mu}}^{\mathsf{Alg}}(T) = \mathbb{E}_{\boldsymbol{\mu}}[T_c], \quad \text{s.t.,} \quad \mathsf{Alg} \in \Pi_{\mathrm{RO}}.$$

# 3 Low-Complexity Algorithms

In this section, we propose a low-complexity algorithm called EOCP with a pre-determined stopping time to solve ROBAI. Then, we propose a variant called EOCP-UG with adaptive stopping time.

## 3.1 The Pre-determined Stopping Time Setting

The pre-determined stopping time setting is motivated by real-world applications such as A/B tests in medical experiments with operational or budget limits [31], where the number of testers is usually pre-designed and determined before the experiment starts. Therefore, we require the agent to pre-specify the stopping time $T_c$ before the first round. We also assume the algorithm knows a strictly positive lower bound $\Delta_{\mathrm{lb}}$ on the minimum reward gap $\Delta_{\min}$ between the optimal action and sub-optimal actions. We propose our EOCP algorithm in Algorithm. 1.

---
**Algorithm 1** EOCP with Pre-determined Stopping Time

---
**Require:** Exploration function $l$; Lower bound $\Delta_{\mathrm{lb}}$ on minimum reward gap.
1: Let $T_c = \frac{8Al}{\Delta_{\mathrm{lb}}^2} + A$ be the pre-determined stopping time.
2: Initialize by pulling each arm $a$ once.
3: **for** $t = A + 1 : T_c$ **do**
4:     Set uncertainty bonus $b_{t-1}(a) = \sqrt{\frac{2l}{N_{t-1}(a)}}$, and $\mathsf{UCB}_{t-1}(a) = \bar{r}_{t-1}(a) + b_{t-1}(a)$.
5:     Take action $A_t = \arg\max_a \mathsf{UCB}_{t-1}(a)$. // UCB Exploration
6: **end for**
7: Set bonus $b_{T_c}(a) = \sqrt{\frac{2l}{N_{T_c}(a)}}$, and $\mathsf{LCB}_{T_c}(a) = \bar{r}_{T_c}(a) - b_{T_c}(a)$.
8: For $t \in [T_c + 1, T]$, commit to action $\hat{a} = \arg\max_a \mathsf{LCB}_{T_c}(a)$. // LCB Commitment

---

In EOCP, the agent will spend the first $A$ rounds exploring each arm once as a start. Our choice of exploration strategy is a modified version of the classic UCB algorithm where the agent will choose the action with the largest upper confidence bound in terms of empirical reward. The exploration function $l$ controls the intensity of exploration to achieve the optimal trade-off between reducing uncertainty for action identification and minimizing regret. After the pre-determined stopping time $T_c$, the agent will commit to an action that has the largest lower confidence bound of empirical

reward. This LCB commitment strategy is inspired by the principle of pessimism from the literature of offline learning [25, 26, 27, 28, 29, 30], where the empirical reward of each action is penalized by the amount of uncertainty to combat the imbalanced data coverage of actions in the offline dataset. It is also shown that the pessimistic principle works well when the data coverage is concentrated on the optimal action, i.e., the optimal action has the largest number of pulls. This trait of the LCB algorithm matches the trait of UCB exploration, in which the optimal action will be chosen much more often than sub-optimal actions in exploration. So by designing such a proper $T_c$, we will be able to achieve the best of both worlds: a low-regret exploration of the UCB algorithm, and a fast best arm identification through the choice of LCB algorithm.

### 3.2 The Adaptive Stopping Time Setting

In this setting, we do not assume the algorithm is provided a priori with additional information on the lower bound $\Delta_{\mathrm{lb}}$ on the minimum reward gap, so there is no hope of designing a pre-determined stopping time. Instead, we design our stopping criterion based on the samples collected from the explorations, which leads to the fact that $T_c$ is a stopping time measurable to the natural filtration. We propose our EOCP-UG algorithm in Algorithm 2 corresponding to an unknown gap.

---

**Algorithm 2** EOCP-UG with Adaptive Stopping Time

---

**Require:** Exploration function $l$.
 1: Initialize by pulling each arm $a$ once.
 2: **while** $\max_a \min_{a'} N_{t-1}(a) - l N_{t-1}(a') \leqslant 1$ **do**
 3:   Set uncertainty bonus $b_{t-1}(a) = \sqrt{\frac{2l}{N_{t-1}(a)}}$, $\mathsf{UCB}_{t-1}(a) = \bar{r}_{t-1}(a) + b_{t-1}(a)$.
 4:   Take action $A_t = \arg\max_a \mathsf{UCB}_{t-1}(a)$. **// UCB Exploration**
 5: **end while**
 6: Let $T_c \leftarrow t - 1$, bonus $b_{T_c}(a) = \sqrt{\frac{2l}{N_{T_c}(a)}}$, and $\mathsf{LCB}_{T_c}(a) = \bar{r}_{T_c}(a) - b_{T_c}(a)$.
 7: For $t \in [T_c + 1, T]$, commit to the action $\hat{a} = \arg\max_a \mathsf{LCB}_{T_c}(a)$. **// LCB Commitment**

---

In EOCP-UG, we use the same UCB exploration and LCB best action identification strategies as in the pre-determined stopping time setting. The only difference compared to EOCP comes from the new stopping rule based on the number of pulls $N_t(a)$ for each action. Specifically, the exploration ends if there is an imbalanced fraction of $N_t(a)$ among all actions, that is, one action has $l$ times more pulls than all other actions in previous rounds. Here, $l$ is the exploration function and we will select $l$ to be slightly larger than $\log T$ in later sections. The intuition of such a stopping criterion comes from the characteristics of UCB exploration, i.e., as round $t$ increases, the algorithm will slowly adapt to choosing the optimal action more often. When the fraction between the number of pulls for the optimal action and any sub-optimal action is large enough, the optimal action will be identifiable. Note that in action identification, we can simply choose the action that has the largest number of pulls $N_{T_c}(a)$ when we stop, and obtain exactly the same performance guarantees. However, to keep it consistent with the pre-determined stopping time setting, we use the LCB commitment.

## 4 Main Results

In this section, we assume the distributions $\{\nu_1, \cdots, \nu_A\}$ come from a Gaussian family, where $\nu_a$ associated with action $a$ follows a Gaussian distribution with mean $\mu_a$ and unit variance, i.e., $\nu_a \sim \mathcal{N}(\mu_a, 1)$. The results in this section can be easily generalized to sub-Gaussian distributions. The key idea towards the optimal trade-off between controlling regret and identifying the optimal action is inspired by the inflated bonus trick from [24], where we will choose the exploration function $l$ to be slightly larger than the $\log T$ used in vanilla UCB algorithms to encourage more exploration of sub-optimal arms, i.e., we will select $l = \log T + \mathcal{O}(\sqrt{\log T})$.

### 4.1 Regret Optimality for Gaussian Bandits with Pre-Determined Stopping Time

In Theorem. 1, we present the theoretical regret performance guarantee of the EOCP algorithm:

**Theorem 1** *Let $l = \log(T) + \mathcal{O}(\sqrt{\log T})$ and when $T$ is large enough, the expected regret of the* EOCP *algorithm in Algorithm. 1 with pre-determined stopping time can be upper-bounded by:*

$$\mathsf{Reg}_{\boldsymbol{\mu}}^{\mathsf{EOCP}}(T) \leqslant \sum_{a:\Delta_a > 0} \frac{2 \log T}{\Delta_a} + \mathcal{O}\left(\frac{\sqrt{\log T}}{\Delta_{\min}}\right).$$

A direct asymptotic bound can be obtained from Theorem. 1, i.e.,

$$\limsup_{T \to \infty} \frac{\mathsf{Reg}_{\boldsymbol{\mu}}^{\mathsf{EOCP}}(T)}{\log T} \leqslant \sum_{a:\Delta_a > 0} \frac{2}{\Delta_a}.$$

Comparing it to Eq. (1), it is clear that EOCP is asymptotically regret optimal in Gaussian bandits. The commitment time and confidence guarantees can be extracted from the setup of Algorithm. 1 itself and the proof of Theorem. 1. Recall that in Algorithm. 1, we pre-determined the length of exploration $T_c$ to be $\mathcal{O}(\Delta_{\mathrm{lb}}^{-2} l)$. The following corollary characterizes these parts of theoretical performance:

**Corollary 1** *Let $l = \log(T) + \mathcal{O}(\sqrt{\log T})$, the expected commitment time for* EOCP *in Algorithm. 1 is given by*

$$\mathsf{SCC}_{\boldsymbol{\mu}}^{\mathsf{EOCP}}(T) = \mathcal{O}(\Delta_{\mathrm{lb}}^{-2} \log T),$$

*and the confidence level is $\mathcal{O}(T^{-1})$.*

The complete proofs of Theorem. 1 and Corollary. 1 are provided in the supplementary material. In order to upper bound the cumulative regret of $T$ rounds, we divide the total regret into the regret accumulated in exploration and the regret accumulated in commitment.

**Bounding Regret from Exploration:** To bound the regret accumulated in exploration and since we use a variant of UCB exploration, we follow the standard procedure of proofs for UCB algorithms, e.g., proof of Theorem 8 from. [22]. Then, this procedure results in an order $\mathcal{O}(l)$ dominating regret term, which is $\mathcal{O}(\log T)$ by the choice of our exploration function. Through carefully applying any-time concentration inequalities, we are able to show that the constant in front of this dominating regret term is exactly the constant we obtained in Theorem. 1.

**Bounding Regret from Committing to the Wrong Action:** As for the regret accumulated from commitment, the key is to prove the $\mathcal{O}(T^{-1})$ confidence level upper bound presented in Corollary. 1. We follow a procedure similar to the proof of Theorem 1 in [2] to utilize the adaptivity of UCB exploration and the pessimistic LCB commitment. We first show that with high probability, the number of pulls $N_t(a)$ for any sub-optimal actions in the exploration phase is upper-bounded. This is because after a certain number of pulls, the uncertainty bonus $b_{t-1}(a)$ for any sub-optimal action will be so small that the upper confidence bound $\mathsf{UCB}_{t-1}(a)$ cannot be larger than $\mu_{a*}$, thus less than the upper confidence bound of the optimal action. Therefore, sub-optimal actions will not be chosen in future rounds. After our carefully designed $T_c$, we make sure that the optimal action has the largest number of pulls $N_{T_c}(a)$ among all actions, thus its bonus is so small so that its lower confidence bound $\mathsf{LCB}_{T_c}(a^*)$ is larger than the expectations $\mu_a$ of any sub-optimal action $a$, and thus larger than the lower confidence bound of other actions. So with high probability, we will commit to the optimal action. Then, the $\mathcal{O}(T^{-1})$ confidence level will provide us with a constant regret in commitment, and the overall dominating regret comes from exploration.

Combining both bounds, we are able to show the regret performance upper bound in Theorem. 1.

### 4.2 Regret Optimality for Gaussian Bandits with Adaptive Stopping Time

We present the regret performance of EOCP-UG in Theorem. 2. Compared to Theorem. 1, EOCP-UG has exactly the same regret performance guarantees as EOCP even without the knowledge of a lower bound $\Delta_{\mathrm{lb}}$ on the minimum reward gap. This implies that EOCP-UG adapts to the reward gap.

**Theorem 2** *Let $l = \log(T) + \mathcal{O}(\sqrt{\log T})$ and when $T$ is large enough, the expected regret of the* EOCP-UG *algorithm in Algorithm. 2 with adaptive stopping time is upper-bounded by:*

$$\mathsf{Reg}_{\boldsymbol{\mu}}^{\mathsf{EOCP\text{-}UG}}(T) \leqslant \sum_{a:\Delta_a > 0} \frac{2 \log T}{\Delta_a} + \mathcal{O}\left(\frac{\sqrt{\log T}}{\Delta_{\min}}\right).$$

We can also directly derive an asymptotic regret bound which shows that EOCP-UG is regret optimal in Gaussian bandits:

$$\limsup_{T \to \infty} \frac{\mathsf{Reg}_{\boldsymbol{\mu}}^{\mathsf{EOCP\text{-}UG}}(T)}{\log T} \leqslant \sum_{a:\Delta_a>0} \frac{2}{\Delta_a}.$$

With adaptive stopping, the sample complexity until the commitment is not a pre-determined value. However, we can still extract similar guarantees along with the confidence level from the proof of Theorem. 2. We present these results in the following corollary:

**Corollary 2** *Let $l = \log(T) + \mathcal{O}(\sqrt{\log T})$, the sample complexity until commitment for* EOCP-UG *algorithm in Algorithm. 2 is upper-bounded by:*

$$\mathsf{SCC}_{\boldsymbol{\mu}}^{\mathsf{EOCP\text{-}UG}}(T) \leqslant \sum_{a:\Delta_a>0} \frac{8 \log^2 T}{\Delta_a^2} + \mathcal{O}\left( \frac{\log^{\frac{3}{2}} T}{\Delta_{\min}^2} \right),$$

*and the confidence level is upper bounded by $\mathcal{O}(T^{-1})$.*

**Proof Roadmap:** The complete proofs of Theorem. 2 and Corollary 2 are provided in the supplementary material. Compared to the proof of Theorem. 1, the major difference lies in bounding the regret of commitment. Similarly, we are required to bound the probability of committing to a sub-optimal action. We first show that when the agent stops exploration according to Line 2 of Algorithm. 2, the action that has the maximum number of pulls is the optimal action with high probability. Then, we show that under this event we will commit to the optimal action if we use LCB commitment.

**Magic Choice of Exploration Function:** Bounding the regret in both exploration and commitment requires a delicate analysis with any time concentration inequalities. Our choice of exploration function $l$ plays an important role which manages the trade-off between low-regret exploration and high-probability optimal action commitment. If the exploration function is too large, i.e., if $l = 2 \log T$, the regret in exploration will not be optimal. If the exploration function is too small, i.e., if $l = \log T$, the probability of committing to the wrong action can not be bounded by $\mathcal{O}(T^{-1})$. Our magic choice of exploration function $l$ achieves the best of both worlds.

**Loss of SCC from Unknown Gap:** Even though we have the same regret performance, the guarantee for commitment time is $\mathcal{O}(\log^2 T)$ which is worse than $\mathcal{O}(\log T)$ in the pre-determined setting. So, is this order fundamental with adaptive stopping time? In the next section, we provide the answer by investigating the theoretical limits of commitment time for asymptotic regret optimal algorithms.

### 4.3 Fundamental Limits of Sample Complexity until Commitment

In order to answer the question regarding the fundamental sample complexity until commitment for regret optimal algorithms with both pre-determined and adaptive stopping times, we consider a simplified Gaussian bandit model where there are only 2 arms. The reward gap between the two actions is $\Delta$, so $\Delta_{\min} = \Delta$. From Eq. (1), it is clear that the optimal regret of any algorithm is asymptotically $2\Delta^{-1} \log(T)$, so the set of all regret optimal algorithms is characterized by:

$$\Pi_{\mathrm{RO}} = \left\{ \mathsf{Alg} \,\middle|\, \limsup_{T \to \infty} \frac{\mathsf{Reg}_{\boldsymbol{\mu}}^{\mathsf{Alg}}(T)}{\log T} \leqslant \frac{2}{\Delta} \text{ and } \mathbb{P}_{\boldsymbol{\mu}}(\hat{a} \neq a^*) = \mathcal{O}\left(T^{-1}\right) \right\}.$$

Furthermore, for each regret optimal algorithm, we say it has $c$-logarithm regret violation if there exists a constant $c \in (0,1)$ (choose the minimum $c$ if there exists multiple) such that the following inequality is satisfied when $T$ is large enough:

$$\left| \mathsf{Reg}_{\boldsymbol{\mu}}^{\mathsf{Alg}}(T) - \frac{2 \log(T)}{\Delta} \right| = \mathcal{O}(\log^c T).$$

If an algorithm has $c$-logarithm regret violation, the largest additional lower order term in its finite-time regret bound should be $\mathcal{O}(\log^c T)$. On the other hand, it also characterizes the convergence rate of the regret to its asymptote. For example, the vanilla UCB algorithm has at most $1/2$-logarithm regret violation, and our EOCP and EOCP-UG algorithms both have at most $1/2$-logarithm regret violation [22, Theorem. 8]. We then provide the following theorem to characterize the fundamental commitment time limits for algorithms in $\Pi_{\mathrm{RO}}$:

**Theorem 3 (Information-Theoretic Limits of** SCC**)** *Consider a 2-armed Gaussian bandits, for any asymptotically regret optimal algorithm* Alg *which has c-logarithm regret violation, in order to guarantee $\mathcal{O}(T^{-1})$ confidence level, the sample complexity until commitment with pre-determined stopping time is lower bounded by*

$$T_{\mathrm{c}} = \Omega(\Delta^{-2}\log(T)),$$

*and with adaptive stopping time, the sample complexity until commitment is lower-bounded by:*

$$\mathbb{E}_{\boldsymbol{\mu}}[T_{\mathrm{c}}] = \Omega(\Delta^{-2}\log^{2-c}(T)).$$

**Proof Roadmap:** The proof is provided in the supplementary material. The proof idea relies on the well-known "transportation" lemma [32, Lemma. 1] originally derived to prove the theoretical limits of best arm identification algorithms. This lemma characterizes the expected number of pulls for each action by hypothesis testing between the original bandit problem and another bandit instance with a different optimal action. Then by finding a proper bandit instance and combining the lemma with regret optimal algorithms, we will be able to prove Theorem. 3 for both settings.

**Sample Optimality:** It is shown by Corollary. 1 and Theorem. 3 together that our proposed EOCP algorithm achieves the optimal commitment time with $\mathcal{O}(\log T)$ with pre-determined stopping time if the reward gap $\Delta$ is known a priori. However, with adaptive stopping time, our EOCP-UG algorithm has $1/2$-logarithm regret violation and $\mathcal{O}(\log^2 T)$ commitment time which is larger than the $\mathcal{O}(\log^{1.5} T)$ lower bound indicated by Theorem. 3. Even though EOCP-UG is not shown to be exactly sample optimal, we conjure that the performance gap comes from our analysis techniques which makes one of the bounds (maybe both) not tight. Whether EOCP-UG is indeed sample optimal, and if not, how to design regret optimal algorithms with $\mathcal{O}(\log^{1.5} T)$ commitment time remains open.

## 5  Regret Optimality for General Bandits

Even though EOCP is applicable in sub-Gaussian bandits, it is not regret optimal any more beyond Gaussian bandits. This can be seen by comparing Theorem. 1 and the fundamental regret limit (1) with Pinsker's inequality, and it is clear that the lower bound is smaller than our upper bound asymptotically beyond Gaussian bandits. To close this gap, we propose an improved algorithm called KL-EOCP which is provably regret optimal in general bandits.

**Natural Exponential Family:** We assume the reward distributions of each action belong to a natural exponential family, i.e., $\mathcal{P} = \{(\nu_\theta)_{\theta \in \Theta} : d\nu_\theta/d\xi = \exp(\theta x - b(\theta))h(x)\}$, where $\Theta \subset \mathbb{R}$ is the set of all parameters $\theta$ such that the expectation $\mu$ is positive and bounded, i.e., $\mu \in [0, 1]$. $\xi$ is some reference measure on $\mathbb{R}$ and $b : \Theta \to \mathbb{R}$ is a convex twice differentiable function. This distribution $\nu_\theta$ can also be parameterized by its expectation $\mu = b'(\theta)$, the derivative of $b(\cdot)$, and for every $\mu$ we denote by $\nu^\mu$ the unique distribution in $\mathcal{P}$ with expectation $\mu$ and by $\theta^\mu$ its corresponding parameter. Gaussian distribution with unit variance is an example of this family. Moreover, the Kullback-Leibler divergence from $\nu_{\theta_1}$ to $\nu_{\theta_2}$ (with a little abuse of notation) can be expressed as [12]:

$$\mathsf{KL}(\mu_1, \mu_2) = \mathsf{KL}(\nu_{\theta_1}, \nu_{\theta_2}) = b(\theta_2) - b(\theta_1) - b'(\theta_1)(\theta_2 - \theta_1).$$

The set of exponential family bandit models $\boldsymbol{\nu} = (\nu_{\theta_1}, \cdots, \nu_{\theta_A})$ can be characterized by the expectations of the actions $\boldsymbol{\mu} = (\mu_1, \cdots, \mu_A)$. We assume for all $\lambda \in \mathbb{R}$, and $\theta \in \Theta$ the moment generating function $M_{\nu_\theta}(\lambda) = \mathbb{E}_{\nu_\theta}[\exp(\lambda W)]$ for the distribution $\nu_\theta$ is well-defined and is finite.

**Algorithm:** Analog to $\Delta_{\mathrm{lb}}$ in Algorithm. 1, the KL-EOCP algorithm requires the knowledge of a strictly positive lower bound $\mathsf{KL}_{\mathrm{lb}}$ on the "minimum KL divergence", denoted as $\mathsf{KL}_{\mathrm{min}}$, which captures the minimum reward distribution gap (distance) between the optimal action and any sub-optimal action. Considering the asymmetricity of the KL divergence, we define $\mathsf{KL}_{\mathrm{min}}$ as follows:

$$\mathsf{KL}_{\mathrm{min}} = \min_{a \neq a*} \min \left\{ \mathsf{KL}(\mu_a, \mu_{a*}), 4\mathsf{KL}(\mu_a', \mu_a) \right\},$$

where $\mu_a' \in (\mu_a, \mu_{a*})$ such that $4\mathsf{KL}(\mu_a', \mu_{a*}) = \mathsf{KL}(\mu_a, \mu_{a*})$. The term $4\mathsf{KL}(\mu_a', \mu_a)$ reflects the skew of $\mathsf{KL}$ divergence when the two distributions are switched. A simple lower bound to $\mathsf{KL}_{\mathrm{min}}$ can be easily computed given a lower bound $\Delta_{\mathrm{lb}}$ on the minimum reward gap $\Delta_{\mathrm{min}}$ with the expression of the exponential family, while our KL-EOCP algorithm can operate with any lower bound $\mathsf{KL}_{\mathrm{lb}}$.

The KL-EOCP algorithm is summarized in Algorithm. 3. It designs the UCB and LCB bonuses based on the KL divergence of the reward distributions for all actions. In general, these designs would lead to smaller confidence intervals with the same $\mathcal{O}(T^{-1})$ concentration guarantees as first adopted in the KL-UCB algorithm proposed in [12]. If the lower bound information $\mathsf{KL}_{\mathrm{lb}}$ is not known a priori, we can combine Algorithm. 3 with the adaptive stopping time of EOCP-UG to deal with this setting.

---

**Algorithm 3** KL-EOCP with Pre-Determined Stopping Time

---

**Require:** Exploration function $l$; Lower bound $\mathsf{KL}_{\mathrm{lb}}$ on the minimum KL divergence.
1: Let $T_{\mathrm{c}} = \frac{4Al}{\mathsf{KL}_{\mathrm{lb}}^2} + A$ be the length of the exploration phase.
2: Initialize by pulling each arm $a$ once.
3: **for** $t = A + 1 : T_{\mathrm{c}}$ **do**
4:     Set upper confidence bound:

$$\mathsf{UCB}_{t-1}(a) = \underset{\mu \geqslant \bar{r}_{t-1}(a)}{\arg\max} \left\{ N_{t-1}(a)\mathsf{KL}(\bar{r}_{t-1}(a), \mu) \leqslant l \right\}.$$

5:     Take action $A_t = \arg\max_a \mathsf{UCB}_{t-1}(a)$.   // UCB Exploration
6: **end for**
7: Set lower confidence bound:

$$\mathsf{LCB}_{T_{\mathrm{c}}}(a) = \underset{\mu \leqslant \bar{r}_{T_{\mathrm{c}}}(a)}{\arg\min} \left\{ N_{T_{\mathrm{c}}}(a)\mathsf{KL}(\bar{r}_{T_{\mathrm{c}}}(a), \mu) \leqslant l \right\}.$$

8: For $t \in [T_{\mathrm{c}} + 1, T]$, commit to action $\hat{a} = \arg\max_a \mathsf{LCB}_{T_{\mathrm{c}}}(a)$.   // LCB Commitment

---

**Regret Optimality in General Bandits:** The theoretical regret performance of the KL-EOCP is summarized in the following Theorem. To the best of our knowledge, it is the first result achieving asymptotic regret optimality in general bandit problems with commitment.

**Theorem 4** *Let $l = \log(T) + \mathcal{O}(\sqrt{\log T})$, when $T$ is large enough and the reward distributions $\boldsymbol{\nu}$ of each action belong to the same natural exponential family, the expected regret of* KL-EOCP *in Algorithm. 3 is upper-bounded by:*

$$\mathsf{Reg}_{\boldsymbol{\mu}}(T) \leqslant \sum_{a:\Delta_a > 0} \frac{\Delta_a \log T}{\mathsf{KL}(\mu_a, \mu_1)} + \mathcal{O}\left( \frac{\log^{\frac{3}{4}} T}{\mathsf{KL}_{\min}} \right).$$

Therefore, it is clear that we can derive an asymptotic upper bound as follows:

$$\limsup_{T \to \infty} \frac{\mathsf{Reg}_{\boldsymbol{\mu}}(T)}{\log T} \leqslant \sum_{a:\Delta_a > 0} \frac{\Delta_a}{\mathsf{KL}(\mu_a, \mu_1)},$$

which exactly matches the regret limit in Eq. (1). The complete proof of Theorem. 4 is provided in the supplementary materials. Even though the proof roadmap is similar to the proof of Theorem. 1, the major difference comes from the use of a tighter concentration lemma modified from [12, Theorem. 11] which captures the low probability event when the KL divergence of the empirical mean is far away from its expectation. The sample complexity until commitment and confidence level guarantees can also be extracted from the proof. Specifically, $\mathsf{SCC}_{\boldsymbol{\mu}}^{\mathsf{KL\text{-}EOCP}}(T) = \mathcal{O}(\mathsf{KL}_{\mathrm{lb}}^{-1} \log T)$ and the confidence level is upper bounded by $\mathcal{O}(T^{-1})$.

# 6 Numerical Experiments

In this section, we study the empirical performance of our proposed EOCP algorithm with variants compared to existing algorithms in the literature, including BAI-ETC [22], UCB [10], KL-UCB [12], and DETC [23] in both Gaussian and Bernoulli bandit settings. In this section, we only test the algorithms on a two-armed bandit problem because some baselines are only applicable in two-armed settings. The performance of our algorithms in bandit models with multiple arms is demonstrated in Appendix. E. In the Gaussian setting, we test all the algorithms with distributions $\mathcal{N}(\mu_i, 1)$ for arm

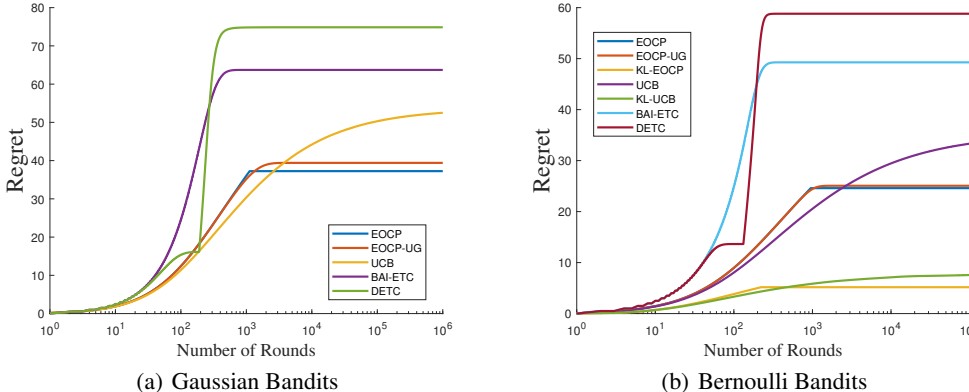

|  | (a) Gaussian Bandits | (b) Bernoulli Bandits |

Figure 1: Comparison of regret performance of EOCP with variants and existing algorithms in the literature. The gap $\Delta$ between the two arms is $0.5$ and results are averaged over $10^5$ iterations.

$i = 1, 2$ with a total of $10^6$ rounds, and in the Bernoulli bandit setting, we test the algorithms with distribution $\text{Ber}(\mu_i)$ for arm $i = 1, 2$ with a total of $10^5$ rounds. We set $\mu_1 = 0.7$ and $\mu_2 = 0.2$, so the gap between the arms is $\Delta = 0.5$. The results are averaged over $10^5$ iterations and in Fig. 1.

In the Gaussian bandit setting, it shows that both BAI-ETC and DETC algorithms exhibit unsatisfactorily high regret. On the contrary, our proposed algorithms EOCP and EOCP-UG have lower final regret even compared to the popular UCB algorithm, surprisingly. Even though our algorithms use a larger exploration function than the vanilla UCB algorithm, and accumulate regret more quickly in exploration, i.e., approximately the first 1000 rounds, it allows us to commit to the optimal action. Our algorithms almost find the optimal action in all simulated traces which gives rise to the very slowly-increasing behavior of regret in the commitment phase. This phenomenon coincides with our theoretical analysis which shows that the regret in commitment is $\mathcal{O}(1)$. However, the UCB algorithm continues to explore sub-optimal actions, so its regret continues to grow when the EOCP algorithm has already committed to the optimal action. Numerically, the EOCP algorithm reduces $20\%$ of the final regret compared to UCB through early commitment, and we conjure that preventing the "over-exploration" phenomenon of the UCB algorithm is behind the reason for such empirical regret reduction. Comparing the commitment time of EOCP-UG and EOCP, we can see that both algorithms stop exploration at approximately 1000 rounds. This means that not knowing the gap information won't harm the empirical sample complexity until commitment too much, which in turn may imply that our theoretical analysis of commitment time upper bound in Corollary. 2 is not tight enough. The same trend can be witnessed in the results of the Bernoulli bandits. However, in Bernoulli bandits, KL-UCB and KL-EOCP algorithms have a much better regret performance than other algorithms, which shows that knowledge of the reward distribution family improves the performance significantly.

## 7 Conclusion

We studied ROBAI which intends to both minimize regret and commit to the optimal action. We proposed EOCP with variants, which combine UCB exploration and LCB commitment with novel stopping criteria in both pre-determined and adaptive settings. We showed that both EOCP and EOCP-UG are regret asymptotic optimal in Gaussian bandits with $\mathcal{O}(\log T)$ and $\mathcal{O}(\log^2 T)$ commitment time respectively, almost matching the theoretical limits we derived. For general bandits, we proposed KL-EOCP which is provably regret optimal. Numerical experiments confirmed the superiority of our algorithms and revealed the "over-exploration" phenomenon of UCB algorithms.

## Acknowledgments and Disclosure of Funding

The work of Qining Zhang and Lei Ying is supported in part by NSF under grants 2112471, 2134081, 2207548, and 222897.

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

# A  Related Works

In this section, we provide a more detailed review of related works. We first review two classic problem formulations of the multi-armed bandit model: the regret minimization problem and the best arm identification problem. Then we review previous works on explore-then-commit algorithms which takes commitment into account. Finally, we review the offline stochastic bandit literature which motivated our choice of pessimistic principle in action identification.

## A.1  Regret Minimization

The theoretical limits of regret minimization have been revealed by [13, 14], which shows that the expected regret of any algorithm is lower bounded when horizon $T$ approaches infinity:

$$\liminf_{T\to\infty} \frac{\mathsf{Reg}_{\boldsymbol{\mu}}^{\mathsf{Alg}}(T)}{\log T} \geqslant \sum_{a:\Delta_a>0} \frac{\Delta_a}{\mathsf{KL}(\nu_a, \nu_{a*})},$$

where $\mathsf{KL}(\cdot, \cdot)$ denotes the Kullback–Leibler divergence between two distributions. Based on the asymptotic lower bound, we say an algorithm is asymptotically regret optimal if its regret performance achieves the regret lower bound asymptotically. Two sets of algorithms prevail in the regret minimization literature. One is the family of UCB algorithms [10, 12, 33], which reflects the principal of optimism in action selection to encourage exploration. To be specific, the UCB algorithms will select the action which has the largest upper confidence bound of reward estimation. This upper confidence bound represents the highest possible expected reward given the samples collected from previous rounds under a high probability event. Usually the additional bonus from the empirical mean to the upper confidence bound for a specific action decreases as the number of pulls increases. Therefore, actions which have not been tried frequently in previous rounds will have larger bonus. This trait encourages exploration. By designing the bonus carefully, one can find the optimal trade-off between exploration and exploitation. The other set of prevailing algorithms is the family of Thompson Sampling Algorithms [1, 34, 11, 15, 35]. The TS algorithm assumes each action is associated with a posterior distribution given the reward feedback from previous rounds. Then the agent will collect one virtual sample from each posterior distribution and choose the action which has the largest virtual sample. As the number of pulls grows, the posterior distribution will be more and more concentrated around the expectation so the virtual sample will also be closer to its expectation. On the other hand, the remaining randomness encourages exploration of other under-explored actions. However, both algorithms do not commit to a single action because their action selection policies have to be re-evaluated at each round based on new observations.

## A.2  Best Arm Identification

The best arm identification algorithms consist of three components: an action sampling rule deciding which action to choose at each round, a stopping rule deciding a time $\tau$ to stop collecting new samples, and a decision rule which outputs a best action candidate $\hat{a}$. We call an algorithm $\delta$-PAC if the probability of outputting a sub-optimal action is less than $\delta$, i.e., $\mathbb{P}_{\boldsymbol{\mu}}(\hat{a} \neq 1) \leqslant \delta$. Here, $\delta$ is also called the confidence level. The performance of best arm identification algorithms is measured by both the sample complexity until identifying the optimal action and the confidence level. The theoretical limits for best arm identification algorithms are also well studied in [36, 32], which shows that any $\delta$-PAC algorithm would incur at least $\mathcal{O}(\log \delta^{-1})$ sample complexity asymptotically. The constant in front of the logarithmic term depends on the bandit instance, i.e., the reward expectation and distribution of every action. Based on the this lower bound, the authors of [36] proposed the Track and Stop (TAS) algorithm which proved to be asymptotic optimal if the distribution associated with each action is from a single parameter natural exponential family. The TAS algorithm estimates the expectation of each action and at the same time calculates the optimal proportion of pulls for each action so that the optimal action is identifiable from the sub-optimal ones. Then, it designs a feedback control dynamic to track this proportion, similar to the Max-weight dynamic in queueing systems [37]. The stopping criterion of TAS is based on a generalized likelihood test. When the optimal action is identifiable from sub-optimal ones, the optimal action candidate $\hat{a}$ is chosen to be the action which has the largest empirical reward mean.

### A.3 Explore-Then-Commit Algorithms

A natural bridge between regret minimization and best arm identification problems, which also takes into account action commitment, is the explore-then-commit algorithms [38, 22, 23, 39, 40, 41].Researchers have long been hoping that such algorithms will achieve the best of both worlds: maintaining low regret and identifying the optimal action quickly. In these algorithms, a clear separation of exploration phase and exploitation phase exists. At each round, the agent will only make action selection decisions based on the samples collected from the exploration phase, and commitment to a single action during the exploitation phase is required. No samples from the exploitation phase can be utilized although the agent may determine the length of both phases. It is clear that explore-then-commit algorithms can be easily designed from best arm identification algorithms, i.e., one would first run the best arm identification algorithm in the exploration phase, and then commit to the action $\hat{a}$ found by the algorithm. However, it is shown in [22] that these type of BAI-ETC algorithms is essentially regret sub-optimal. To be specific, the regret lower bound of such algorithms is asymptotically twice as large as the upper bound of optimal regret minimization algorithms such as UCB and Thompson Sampling even with careful tuning. A recent work [23] provides a double explore-then-commit algorithm called DETC which is shown to be asymptotically regret optimal. But the algorithm itself is very complex and requires multiple stages of exploration and exploitation. This trait makes its sample complexity to identify the optimal action large, i.e., $\Omega(\log^2 T)$. Moreover, in empirical studies, the DETC algorithm incurs very large regret which is no match for the vanilla UCB algorithm. Another work [24] proposes an exploration algorithm with inflated UCB, which can be naturally adapted to an ETC algorithm. Even though the sample complexity is order optimal, i.e., $\mathcal{O}(\log T)$, the regret performance is essentially sub-optimal due to the inflation of UCB bonus.

### A.4 Offline Stochastic Bandits and Reinforcement Learning

Our action identification policy is inspired by the principle of pessimism widely adopted in offline stochastic bandit problems [26, 27, 25] and reinforcement learning problems [28, 30, 29]. In offline bandit problems, the agent is not allowed to interact with the environment at will. Instead, it is provided with a training dataset which contains action and reward pairs collected from the same bandit problem. Based on this dataset, the agent is asked to choose the optimal action. It is shown in [25] that greedily selecting the action with largest empirical mean would fail to produce the optimal action in some bandit problems. This failure results from the randomness of samples in the dataset and the imbalanced number of samples for each action. Instead, choosing the action with largest lower confidence bound to combat the imbalanced uncertainty between estimations of different actions leads to better performance as justified in [27, 25]. Similar to the UCB algorithm, the penalty term between the empirical mean and the lower confidence bound for each action decreases as the number of pulls increases, so the action which has not been tried frequently will suffer large penalty. In this way, the agent will avoid selecting an action which has large uncertainty to enhance stability. Our proposed algorithm incorporates the idea of pessimism in action identification.

## B   Proofs of Main Results for EOCP and EOCP-UG

In this section, we provide the proofs of main results presented in Section. 4. Throughout the proof section, we let $W_{a,i}$ be the $i$-th sample from pulling arm $a$ the $i$-th time. Let $\bar{r}_{a,s}$ be the empirical mean of arm $a$ after it has been pulled $s$ times. Before proving the theorems, we provide several lemmas which gathers specific large deviation results useful in our analyses. Then we prove the regret performance based on the concentration lemmas.

### B.1   Concentration Inequalities

We first present two concentration lemmas which characterize the sum and mean of empirical realizations of independent sub-Gaussian random variables as follows:

**Lemma 1 (Theorem 9.2 in [38])** *Let $W_1, W_2, \cdots, W_T$ be a sequence of independent $\sigma$-subgaussian random variables with $\mathbb{E}[W_1] = 0$. Then, for any $\delta > 0$, we have:*

$$\mathbb{P}\left(\exists s \leqslant T, \sum_{i=1}^{s} W_i \geqslant \delta\right) \leqslant \exp\left(-\frac{\delta^2}{2T\sigma^2}\right). \tag{2}$$

**Lemma 2 (Lemma C.3 in [23])** *Let $T_1 \leqslant T_2 \leqslant T$ be two real numbers in $\mathbb{R}^+$. Let $W_1, W_2, \cdots, W_T$ be a sequence of identically and independently distributed random variable according to a $\sigma$-subgaussian distribution with $\mathbb{E}[W_1] = 0$. Then, for any $\delta > 0$, we have:*

$$\mathbb{P}\left(\exists T_1 \leqslant s \leqslant T_2, \frac{\sum_{i=1}^{s} W_i}{s} \geqslant \delta\right) \leqslant \exp\left(-\frac{T_1\delta^2}{2\sigma^2}\right) \tag{3}$$

The proofs of aforementioned concentration lemmas can be found in the references respectively. Now we present the concentration results for our design of confidence bonuses as follows:

**Lemma 3** *Let $W_1, W_2, \cdots, W_T$ be identically and independently distributed 1-sub-Gaussian random variables with $\mathbb{E}[W_1] = 0$. Let $T_1 \leqslant T_2 \leqslant T$, then the following holds:*

*(a) if $l \geqslant 2$, $\mathbb{P}\left(\exists s \in (T_1, T_2], \frac{\sum_{i=1}^{s} W_i}{s} + \sqrt{\frac{2l}{s}} \leqslant 0\right) \leqslant \frac{\min\{T_2 - T_1, el(\log T_2 - \log T_1) + e\}}{\exp(l)}$;*

$$\tag{4a}$$

*(b) if $\delta \in (0, \sqrt{3}]$, $\mathbb{P}\left(\exists s \in [T_1, T_2], \frac{\sum_{i=1}^{s} W_i}{s} + \sqrt{\frac{2l}{s}} + \delta \leqslant 0\right) \leqslant \dfrac{4}{\delta^2 \exp\left(\left(\sqrt{l} + \delta\sqrt{\frac{T_1}{2}}\right)^2\right)}$;*

$$\tag{4b}$$

*(c) if $l \geqslant \frac{T_1\delta^2}{2}$, $\displaystyle\sum_{s=T_1+1}^{T_2} \mathbb{P}\left(\frac{\sum_{i=1}^{s} W_i}{s} + \sqrt{\frac{2l}{s}} \geqslant \delta\right) \leqslant \frac{2l + \sqrt{4\pi l} + 2}{\delta^2} + 1 - T_1.$ $\tag{4c}$*

The proof of the Lemma. 3 will be delayed to Sec. D.

## B.2 Proof of Regret Optimality for EOCP with Pre-Determined Stopping Time

We prove the following Theorem which characterizes the finite-time performance of Algorithm. 1. Theorem. 1 can be directly derived.

**Theorem 5** *If $l = \log(T) + 4\sqrt{2\log(T)}$ and $T \geqslant \max\{16, A, 16l\Delta_{\text{lb}}^{-2}\}$, the expected regret of the EOCP algorithm in Algorithm. 1 with pre-determined stopping time is upper bounded by:*

$$\text{Reg}_{\boldsymbol{\mu}}^{\text{EOCP}}(T) \leqslant \sum_{a:\Delta_a>0}\left(\frac{2\log T}{\Delta_a} + \frac{(8 + \sqrt{20\pi})\sqrt{\log T}}{\Delta_a} + \frac{2}{\Delta_a} + \Delta_a\right) + o(1).$$

**Remark:** The asymptotic regret upper bound is clear from Theorem.5 by letting $T$ increases to infinity, i.e.,

$$\limsup_{T \to \infty} \frac{\text{Reg}_{\boldsymbol{\mu}}^{\text{EOCP}}(T)}{\log T} \leqslant \limsup_{T \to \infty} \sum_{a:\Delta_a>0}\left(\frac{2}{\Delta_a} + \frac{(8 + \sqrt{20\pi})}{\Delta_a\sqrt{\log T}} + \frac{2}{\Delta_a \log T} + \frac{\Delta_a}{\log T}\right) + o(1)$$

$$= \sum_{a:\Delta_a>0} \frac{2}{\Delta_a}.$$

**Proof.** Without loss of generality, assume action 1 is the unique optimal action. From the regret decomposition lemma [38, Lemma 4.5], we can decompose the regret of Algorithm. 1 to the number of pulls for each sub-optimal arm as follows:

$$\text{Reg}_{\boldsymbol{\mu}}^{\text{EOCP}}(T) = \sum_{a:\Delta_a>0} \Delta_a \mathbb{E}[N_T(a)].$$

Then, the key to bound the total regret is to bound the number of pulls for each sub-optimal arms. Since our EOCP algorithm has a clear separation of exploration and exploitation phases, so for any sub-optimal action $a$, we can bound its pulls in different phases as follows:

$$\mathbb{E}[N_a(T)] \leqslant \underbrace{\mathbb{E}\left[N_a(T_{\mathrm{c}})\right]}_{I_1} + (T - T_{\mathrm{c}}) \underbrace{\mathbb{P}(\hat{a} = a)}_{I_2},$$

where $T_{\mathrm{c}}$ is the end of exploration phase and $\hat{a}$ is the action that the algorithm commits to. Notice that the bound on $I_2$ gives the confidence level result provided in Corollary. 1.

**Bounding $I_1$:** We first bound $I_1$, which is similar to the proof of bounding the regret for the traditional UCB algorithm [12, 10, 22]. We have:

$$I_1 = \mathbb{E}\left[\sum_{t=1}^{T_{\mathrm{c}}} \mathbb{1}_{A_t = a}\right] = 1 + \underbrace{\mathbb{E}\left[\sum_{t=A+1}^{T_{\mathrm{c}}} \mathbb{1}_{A_t = a} \mathbb{1}_{\mathsf{UCB}_{t-1}(a) \geqslant \mu_1}\right]}_{A_1} + \underbrace{\mathbb{E}\left[\sum_{t=A+1}^{T_{\mathrm{c}}} \mathbb{1}_{A_t = a} \mathbb{1}_{\mathsf{UCB}_{t-1}(a) < \mu_1}\right]}_{A_2}.$$

Notice that $A_t = a$ indicates $\mathsf{UCB}_{t-1}(1) < \mathsf{UCB}_{t-1}(a)$ due to the dynamic of UCB exploration, we can bound the last term as:

$$A_2 \leqslant \sum_{t=A+1}^{T_{\mathrm{c}}} \mathbb{E}\left[\mathbb{1}_{\mathsf{UCB}_{t-1}(1) < \mu_1}\right] = \sum_{t=A+1}^{T_{\mathrm{c}}} \mathbb{P}\left(\mathsf{UCB}_{t-1}(1) < \mu_1\right).$$

Event $\{\mathsf{UCB}_{t-1}(1) < \mu_1\}$ means that at round $t$, the mean estimation $\bar{r}_{t-1}(1)$ of the optimal arm from previous pulls is lower than $\mu_1 - b_{t-1}(1)$, which incurs a large deviation, so we can bound this event with a union over all rounds in the exploration phase:

$$\{\mathsf{UCB}_{t-1}(1) < \mu_1\} \subset \left\{\exists t' \in [A, T_{\mathrm{c}}), \mathsf{UCB}_{t'}(1) < \mu_1\right\},$$

which doesn't depend on round number $t$ any more, so we have:

$$\sum_{t=A+1}^{T_{\mathrm{c}}} \mathbb{P}\left(\mathsf{UCB}_{t-1}(1) < \mu_1\right) \leqslant \sum_{t=A+1}^{T_{\mathrm{c}}} \mathbb{P}\left(\exists t' \in [A, T_{\mathrm{c}}), \mathsf{UCB}_{t'}(1) < \mu_1\right)$$

$$= (T_{\mathrm{c}} - A)\mathbb{P}\left(\exists t' \in [A, T_{\mathrm{c}}), \mathsf{UCB}_{t'}(1) < \mu_1\right)$$

$$= (T_{\mathrm{c}} - A)\mathbb{P}\left(\exists t' \in [A, T_{\mathrm{c}}), \bar{r}_{t'}(1) + \sqrt{\frac{2l}{N_{t'}(1)}} < \mu_1\right).$$

It is worth-noting that if there is no pulls for arm 1 between a time interval, its empirical mean and number of pulls will remain the same. So we have:

$$\left\{\exists t' \in [A, T_{\mathrm{c}}), \bar{r}_{t'}(1) + \sqrt{\frac{2l}{N_{t'}(1)}} < \mu_1\right\} \subset \left\{\exists s \in [1, N_{T_{\mathrm{c}}}(1)), \bar{r}_{s,1} + \sqrt{\frac{2l}{s}} < \mu_1\right\}.$$

Notice that $N_{T_{\mathrm{c}}}(1) \leqslant T_{\mathrm{c}} - A + 1$, so we can bound the probability as follows:

$$\mathbb{P}\left(\exists t' \in [A, T_{\mathrm{c}}), \bar{r}_{t'}(1) + \sqrt{\frac{2l}{N_{t'}(1)}} < \mu_1\right)$$

$$\leqslant \mathbb{P}\left(\exists s \in [1, T_{\mathrm{c}} - A + 1), \bar{r}_{s,1} - \mu_1 + \sqrt{\frac{2l}{s}} < 0\right)$$

$$\leqslant \mathbb{P}\left(\exists s \in [1, T_{\mathrm{c}} - A + 1), \frac{\sum_{i=1}^{s}(W_{1,i} - \mu_1)}{i} + \sqrt{\frac{2l}{s}} < 0\right)$$

$$\leqslant \frac{T_{\mathrm{c}} - A + 1}{\exp(l)},$$

where the last inequality uses Eq. (4a) of Lemma 3 and the fact that $W_{1,i} - \mu_1$ is 1-sub-Gaussian with zero mean. Notice that $\sqrt{\log T} \geqslant \log\log T$ when $T$ is large and we have:

$$\exp(l) = \exp\left(\log T + 4\sqrt{2\log T}\right) \geqslant \exp(\log T + 4\log\log T) = \frac{1}{T\log^4 T}.$$

So putting the above bound back to the bound of $A_2$, we have:

$$A_2 \leqslant \frac{(T_c - A + 1)^2}{T \log^2 T} = \frac{\left( \frac{8A\left( \log T + 4\sqrt{2 \log T} \right)}{\Delta_{lb}^2} + 1 \right)^2}{T \log^4 T} = o\left( \frac{1}{T} \right),$$

where the last inequality is due to the fact that when $T \geqslant 3$, $\sqrt{\log T} \leqslant \log T$. Next, we attempt to bound the middle term $A_1$ as follows:

$$A_1 = \mathbb{E}\left[ \sum_{t=A+1}^{T_c} \mathbb{1}_{A_t=a} \mathbb{1}_{\mathsf{UCB}_{t-1}(a) \geqslant \mu_1} \right] = \mathbb{E}\left[ \sum_{t=A+1}^{T_c} \mathbb{1}_{A_t=a} \mathbb{1}_{\bar{r}_{t-1}(a) + \sqrt{\frac{2l}{N_{t-1}(a)}} \geqslant \mu_1} \right].$$

Notice that the number of pulls $N_t(a)$ will only increase by 1 every time there is new pull, i.e., when $A_t = a$. Otherwise, the term inside summation is 0. So instead of counting on the time step $t$, we can count over the number of pulls over arm $a$ as follows:

$$\mathbb{E}\left[ \sum_{t=A+1}^{T_c} \mathbb{1}_{A_t=a} \mathbb{1}_{\bar{r}_{t-1}(a) + \sqrt{\frac{2l}{N_{t-1}(a)}} \geqslant \mu_1} \right] = \mathbb{E}\left[ \sum_{s=1}^{N_{T_c}(a)} \mathbb{1}_{\bar{r}_{s,a} + \sqrt{\frac{2l}{s}} \geqslant \mu_1} \right]$$

$$\leqslant \sum_{s=1}^{T_c} \mathbb{P}\left( \bar{r}_{s,a} - \mu_a + \sqrt{\frac{2l}{s}} \geqslant \Delta_a \right),$$

where the last inequality is due to $N_{T_c}(a) \leqslant T_c$. Then, by Eq. (4c) of Lemma. 3, we have:

$$\sum_{s=1}^{T_c} \mathbb{P}\left( \bar{r}_{s,a} - \mu_a + \sqrt{\frac{2l}{s}} \geqslant \Delta_a \right) = \sum_{s=1}^{T_c} \mathbb{P}\left( \frac{\sum_{i=1}^{s}(W_{a,i} - \mu_a)}{i} + \sqrt{\frac{2l}{s}} \geqslant \Delta_a \right)$$

$$\leqslant \frac{2\left( \log T + 4\sqrt{2 \log T} \right) + \sqrt{4\pi \left( \log T + 4\sqrt{2 \log T} \right)} + 2}{\Delta_a^2} + 1$$

$$\leqslant \frac{2 \log T}{\Delta_a^2} + \frac{(8 + \sqrt{20\pi})\sqrt{\log T}}{\Delta_a^2} + \frac{2}{\Delta_a^2} + 1,$$

where the first inequality is due to the fact that $W_{a,i} - \mu_a$ is 1-sub-Gaussian with zero mean, and the last inequality is due to $\sqrt{2 \log T} \leqslant \log T$ when $T \geqslant 9$. Therefore, combining the bounds on $A_1$ and $A_2$, we can bound $I_1$ as follows:

$$I_1 \leqslant \frac{2 \log T}{\Delta_a^2} + \frac{(8 + \sqrt{20\pi})\sqrt{\log T}}{\Delta_a^2} + \frac{2}{\Delta_a^2} + 1 + o(1).$$

**Bounding $I_2$:** The bound on $I_2$ gives the confidence level result provided in Corollary. 1. After the exploration phase, recall that we select the arm with the largest lower confidence bound to commit to. The idea of bounding $I_2$ is very similar to the regret bound in [2] where since we use UCB to explore in the exploration phase, the optimal arm should be pulled very often such that its bonus becomes very small when exploration phase ends. Then selecting the arm with largest LCB will ensure we select the optimal arm with high probability. This is because the LCB of the optimal arm is larger than the true means of any sub-optimal arms, and the true means of sub-optimal arms are larger than their respective LCB, both with high probability. We can first bound the probability of selecting a sub-optimal arm $a$ as:

$$\mathbb{P}(\hat{a} = a) \leqslant \mathbb{P}\left( \mathsf{LCB}_{T_c}(a) \geqslant \mathsf{LCB}_{T_c}(1) \right)$$

$$\leqslant \underbrace{\mathbb{P}\left( \mathsf{LCB}_{T_c}(a) \geqslant \mathsf{LCB}_{T_c}(1), N_{T_c}(1) > \frac{8l}{\Delta_{lb}^2} \right)}_{B_1} + \underbrace{\mathbb{P}\left( N_{T_c}(1) \leqslant \frac{8l}{\Delta_{lb}^2} \right)}_{B_2}.$$

We first bound the term $B_2$ which states that during the first exploration phase with UCB exploration, the optimal arm is under-pulled, which means that one of the sub-optimal arms have been pulled with larger number of times. Therefore, we have:

$$\mathbb{P}\left( N_{T_c}(1) \leqslant \frac{8l}{\Delta_{lb}^2} \right) = \mathbb{P}\left( \sum_{a:\Delta_a>0} N_{T_c}(a) > T_c - \frac{8l}{\Delta_{lb}^2} \right).$$

Recall that $T_c = \frac{8Al}{\Delta_{lb}^2} + A$, so we have:

$$\mathbb{P}\left(N_{T_c}(1) \leqslant \frac{8l}{\Delta_{lb}^2}\right) = \mathbb{P}\left(\sum_{a:\Delta_a>0} N_{T_c}(a) > \frac{8(A-1)l}{\Delta_{lb}^2} + A\right)$$

$$\leqslant \mathbb{P}\left(\exists a > 1, N_{T_c}(a) > \frac{8l}{\Delta_{lb}^2} + 1\right).$$

Then by union bound, we have:

$$\mathbb{P}\left(\exists a > 1, N_{T_c}(a) > \frac{8l}{\Delta_{lb}^2} + 1\right) \leqslant \sum_{a:\Delta_a>0} \mathbb{P}\left(N_{T_c}(a) > \frac{8l}{\Delta_{lb}^2} + 1\right).$$

Consider a fixed sub-optimal arm $a$, then for each probability inside the summation, we have:

$$\mathbb{P}\left(N_{T_c}(a) > \frac{8l}{\Delta_{lb}^2} + 1\right)$$

$$\leqslant \mathbb{P}\left(\exists t \in [A+1, T_c], N_{t-1}(a) = \left\lceil \frac{8l}{\Delta_{lb}^2} \right\rceil, A_t = 2\right)$$

$$\leqslant \mathbb{P}\left(\exists t \in [A+1, T_c], N_{t-1}(a) = \left\lceil \frac{8l}{\Delta_{lb}^2} \right\rceil, \mathsf{UCB}_{t-1}(1) \leqslant \mathsf{UCB}_{t-1}(a)\right)$$

$$\leqslant \underbrace{\mathbb{P}\left(\exists t \in [A+1, T_c], N_{t-1}(a) = \left\lceil \frac{8l}{\Delta_{lb}^2} \right\rceil, \mu_1 \leqslant \mathsf{UCB}_{t-1}(a)\right)}_{B_3}$$

$$+ \underbrace{\mathbb{P}\left(\exists t \in [A+1, T_c], \mathsf{UCB}_{t-1}(1) \leqslant \mu_1\right)}_{B_4}.$$

Notice that $B_4$ can be bounded from concentration lemma. First, we switch the count from time step $t$ to the number of pulls for arm 1 since the empirical estimation $\bar{r}_{t-1}(1)$ and count for pulls $N_{t-1}(1)$ won't change unless there is a new pull. Then, we will apply Eq. (4a) to the probability as follows:

$$B_4 = \mathbb{P}\left(\exists t \in [A+1, T_c], \bar{r}_{t-1}(1) - \mu_1 + \sqrt{\frac{2l}{N_{t-1}(1)}} \leqslant 0\right)$$

$$\leqslant \mathbb{P}\left(\exists s \in [1, T_c - A + 1), \bar{r}_{s,1} - \mu_1 + \sqrt{\frac{2l}{s}} \leqslant 0\right)$$

$$\leqslant \mathbb{P}\left(\exists s \in [1, T_c - A + 1), \frac{\sum_{i=1}^{s}(W_{1,i} - \mu_1)}{s} + \sqrt{\frac{2l}{s}} \leqslant 0\right)$$

$$\leqslant \frac{T_c - A}{\exp l},$$

where the last inequality is due to the fact that $W_{1,i} - \mu_1$ is 1-sub-Gaussian with zero mean. Similar to the procedure of bounding $I_1$, notice that $\exp(l) \geqslant T \log^4 T$, we have:

$$B_4 \leqslant \frac{8A\left(\log T + 4\sqrt{2\log T}\right)}{\Delta_{lb}^2} \frac{1}{T\log^4 T} = o\left(\frac{1}{T}\right).$$

On the other hand, let $\gamma = \left\lceil \frac{8l}{\Delta_{lb}^2} \right\rceil$, and we also change the count from time step $t$ to the number of pulls for arm $a$. Then, $B_3$ can be expressed as follows:

$$B_3 = \mathbb{P}\left(\exists t \in [A+1, T_c - 1], N_{t-1}(a) = \gamma, \mu_1 \leqslant \bar{r}_{t-1}(a) + \sqrt{\frac{2l}{N_{t-1}(a)}}\right)$$

$$= \mathbb{P}\left(\bar{r}_{\gamma,a} + \sqrt{\frac{2l}{\gamma}} > \mu_1\right).$$

Notice that $\gamma \geqslant \frac{8l}{\Delta_{\mathrm{lb}}^2} \geqslant \frac{8l}{\Delta_a^2}$, so we have $\sqrt{\frac{2l}{\gamma}} \leqslant \sqrt{\frac{2l\Delta_a^2}{8l}} = \frac{\Delta_a}{2}$. Then $B_3$ can be bounded as follows:

$$B_3 \leqslant \mathbb{P}\left(\bar{r}_{\gamma,a} + \frac{\Delta_a}{2} > \mu_1\right) = \mathbb{P}\left(\bar{r}_{\gamma,a} - \mu_a > \frac{\Delta_a}{2}\right) \leqslant \exp\left(-\frac{\gamma\Delta_a^2}{8}\right) \leqslant \frac{1}{\exp(l)} \leqslant \frac{1}{T\log^4 T},$$

where the first inequality uses Hoeffding's inequality and the second inequality uses the lower bound on $\gamma$ mentioned above. The last inequality is due to the fact that $\exp(l) \geqslant T\log^4 T$. So combine $B_3$ and $B_4$ together, we can bound $B_2$ as follows:

$$B_2 \leqslant \sum_{a:\Delta_a>0} (B_3 + B_4) = o\left(\frac{1}{T}\right).$$

Next, we attempt to bound $B_1$. Notice that $B_1$ can also be bounded as follows:

$$B_1 = \mathbb{P}\left(\mathsf{LCB}_{T_c}(a) \geqslant \mathsf{LCB}_{T_c}(1), N_{T_c}(1) > \frac{8l}{\Delta_{\mathrm{lb}}^2}\right)$$

$$\leqslant \underbrace{\mathbb{P}\left(\mu_a \geqslant \mathsf{LCB}_{T_c}(1), N_{T_c}(1) > \frac{8l}{\Delta_{\mathrm{lb}}^2}\right)}_{B_5} + \underbrace{\mathbb{P}\left(\mu_a \leqslant \mathsf{LCB}_{T_c}(a)\right)}_{B_6}.$$

The term $B_6$ can be expressed by counting the number of pulls of arm $a$ as follows:

$$B_6 = \mathbb{P}\left(\mu_a \leqslant \bar{r}_{T_c}(a) - \sqrt{\frac{2l}{N_{T_c}(a)}}\right)$$

$$\leqslant \mathbb{P}\left(\exists s \in [1, T_c - A + 1], \mu_a - \bar{r}_{s,a} + \sqrt{\frac{2l}{s}} \leqslant 0\right)$$

$$= \mathbb{P}\left(\exists s \in [1, T_c - A + 1], \frac{\sum_{i=1}^s (\mu_a - W_{a,i})}{s} + \sqrt{\frac{2l}{s}} \leqslant 0\right).$$

Notice that $(\mu_a - W_{a,i})$ is 1-sub-Gaussian with zero mean, so we can apply concentration lemma Eq. (4a) from Lemma. (3) as follows:

$$B_6 \leqslant \frac{T_c - A + 1}{\exp(l)} = \frac{8A\left(\log T + 4\sqrt{2\log T}\right)}{\Delta_{\mathrm{lb}}^2 \exp(l)} + \frac{1}{\exp(l)} = o\left(\frac{1}{T}\right),$$

where the last inequality is due to $\log T \geqslant \sqrt{2\log T}$ and $\exp(l) \geqslant T\log^4 T$ when $T \geqslant 9$. Similarly, term $B_5$ can be expressed as:

$$B_5 = \mathbb{P}\left(\mu_a \geqslant \bar{r}_{T_c}(1) - \sqrt{\frac{2l}{N_{T_c}(1)}}, N_{T_c}(1) > \frac{8l}{\Delta_{\mathrm{lb}}^2}\right).$$

First notice that when $N_{T_c}(1) > \frac{8l}{\Delta_{\mathrm{lb}}^2}$, we have the bonus term $b_{T_c}(1) = \sqrt{\frac{2l}{N_{T_c}(1)}} \leqslant \frac{\Delta_{\mathrm{lb}}}{2} \leqslant \frac{\Delta_a}{2}$. Then we have:

$$B_5 \leqslant \mathbb{P}\left(\mu_a \geqslant \bar{r}_{T_c}(1) - \frac{\Delta_a}{2}, N_{T_c}(1) > \frac{8l}{\Delta_{\mathrm{lb}}^2}\right) = \mathbb{P}\left(\mu_1 - \bar{r}_{T_c}(1) \geqslant \frac{\Delta_a}{2}, N_{T_c}(1) > \frac{8l}{\Delta_{\mathrm{lb}}^2}\right).$$

Then it is equivalent to count over the number of pulls for arm 1:

$$B_5 \leqslant \mathbb{P}\left(\exists s \in \left[\frac{8l}{\Delta_{\mathrm{lb}}^2}, T\right], \mu_1 - \bar{r}_{s,1} \geqslant \frac{\Delta_a}{2}\right)$$

$$= \mathbb{P}\left(\exists s \in \left[\frac{8l}{\Delta_{\mathrm{lb}}^2}, T\right], \frac{\sum_{i=1}^s (\mu_1 - W_{1,i})}{s} \geqslant \frac{\Delta_a}{2}\right).$$

Then, by the maximal concentration Eq. (3) from Lemma. 2, we can bound $B_5$ as follows:

$$\mathbb{P}\left(\exists s \in \left[\frac{8l}{\Delta_{\mathrm{lb}}^2}, T\right], \frac{\sum_{i=1}^s (\mu_1 - W_{1,i})}{s} \geqslant \frac{\Delta_a}{2}\right) \leqslant \exp\left(-\frac{\frac{8l}{\Delta_{\mathrm{lb}}^2}\left(\frac{\Delta_a}{2}\right)^2}{2}\right) = \frac{1}{\exp(l)} \leqslant \frac{1}{T\log^4 T}.$$

Therefore, collecting the bounds for $B_5$ and $B_6$, we have a bound for $B_1$ as follows:

$$B_1 \leqslant B_5 + B_6 = o\left(\frac{1}{T}\right).$$

And therefore from the bounds of $B_1$ and $B_2$, we can bound term $I_2$ as follows:

$$I_2 \leqslant B_1 + B_2 = o\left(\frac{1}{T}\right).$$

This means the the confidence of selecting the wrong action to commit to is $o(T^{-1})$ as indicated in Corollary. 1. Finally, putting the bounds on $I_1$ and $I_2$ together, we have:

$$\mathbb{E}[N_a(T)] \leqslant I_1 + T I_2$$
$$\leqslant \frac{2\log T}{\Delta_a^2} + \frac{(8 + \sqrt{20\pi})\sqrt{\log T}}{\Delta_a^2} + \frac{2}{\Delta_a^2} + 1 + o(1).$$

Therefore, by the regret decomposition lemma, we can bound the total regret as follows:

$$\mathsf{Reg}_{\boldsymbol{\mu}}(T) \leqslant \sum_{a:\Delta_a>0} \left(\frac{2\log T}{\Delta_a} + \frac{(8 + \sqrt{20\pi})\sqrt{\log T}}{\Delta_a} + \frac{2}{\Delta_a} + \Delta_a + o(1)\right)$$
$$= \sum_{a:\Delta_a>0} \left(\frac{2\log T}{\Delta_a} + \frac{(8 + \sqrt{20\pi})\sqrt{\log T}}{\Delta_a} + \frac{2}{\Delta_a} + \Delta_a\right) + o(1).$$

### B.3 Proof of Regret Optimality for EOCP-UG with Adaptive Stopping Time

We prove the following theorem which characterizes the finite-time performance of Algorithm. 2. Theorem. 2 can be derived directly.

**Theorem 6** *If $l = \log(T) + 4\sqrt{2\log(T)}$ and $T \geqslant \max\{16, A, 16l\Delta_{\min}^{-2}\}$, the regret of* EOCP-UG *algorithm in Algorithm. 2 with adaptive stopping time can be upper bounded as:*

$$\mathsf{Reg}_{\boldsymbol{\mu}}^{\mathsf{EOCP\text{-}UG}}(T) \leqslant \sum_{a:\Delta_a>0} \left(\frac{2\log T}{\Delta_a} + \frac{(8 + \sqrt{20\pi})\sqrt{\log T}}{\Delta_a}\right) + \mathcal{O}(1).$$

**Remark:** The asymptotic regret upper bound is clear from Theorem.6 by letting $T$ increases to infinity, i.e.,

$$\limsup_{T\to\infty} \frac{\mathsf{Reg}_{\boldsymbol{\mu}}^{\mathsf{EOCP\text{-}UG}}(T)}{\log T} \leqslant \limsup_{T\to\infty} \sum_{a:\Delta_a>0} \left(\frac{2}{\Delta_a} + \frac{(8 + \sqrt{20\pi})}{\Delta_a\sqrt{\log T}}\right) + \mathcal{O}(\log^{-1}(T)) = \sum_{a:\Delta_a>0} \frac{2}{\Delta_a}.$$

**Proof.** Without loss of generality, let action 1 be the unique optimal action. The first step for proving the regret performance is regret decomposition lemma [38, Lemma 4.5]. We also decompose the regret of Algorithm. 2 into the number of pulls for each sub-optimal arm as follows:

$$\mathsf{Reg}_{\boldsymbol{\mu}}(T) = \sum_{a:\Delta_a>0} \Delta_a \mathbb{E}[N_T(a)].$$

Then, for a specific sub-optimal arm $a$, we bound the number of pulls. Since our EOCP-UG algorithm has a clear separation of exploration and exploitation phases, we can bound the pulls in the two phases respectively. However, the unique characteristic of unknown gap scenario is we don't have a fixed end time of exploration phase. Recall that $T_c \leqslant T$ is the stopping time that the exploration phase ends and $\hat{a}$ is the arm we choose for commitment, so we can decompose the number of pulls into two phases as:

$$\mathbb{E}[N_a(T)] = \mathbb{E}[N_a(T_c)] + \mathbb{E}[(T - T_c)\mathbb{1}_{\hat{a}=a}] \leqslant \underbrace{\mathbb{E}[N_a(T_c)]}_{I_1} + T\underbrace{\mathbb{P}(\hat{a} = a)}_{I_2}.$$

Notice that upper bound of $I_2$ gives the confidence level result in Corollary. 2. We then bound the two terms $I_1$ and $I_2$ separately. In order to simplify the proof, we first prove a lemma which characterizes a high probability upper bound for the stopping time $T_c$. This lemma also proves the sample complexity to commitment result in Corollary. 2. The proof of Lemma. 4 will be delayed.

**Lemma 4** *If $l = \log(T) + 4\sqrt{2\log(T)}$ and $T \geq \max\{16, A, 16l\Delta_{\min}^{-2}\}$, our stopping time $T_c$ for exploration of Algorithm. 2 is upper bounded with high probability:*

$$\mathbb{P}\left(T_c \geq \sum_{a:\Delta_a > 0} \frac{8(l+1)^2}{\Delta_a^2} + A(l+2)\right) \leq \frac{10eA}{T\log^2 T}.$$

**Bounding $I_1$:** With the help of Lemma. 4, we have a high probability upper bound for the exploration phase. We let $T_c^u = \sum_{a:\Delta_a > 0} \frac{8(l+1)^2}{\Delta_a^2} + A(l+2)$ to be the high probability upper bound of our stopping time to end the exploration phase. Notice that $N_a(T_c) \leq T$, we have:

$$I_1 = \mathbb{E}[N_a(T_c)\mathbb{1}_{T_c \geq T_c^u}] + \mathbb{E}[N_a(T_c)\mathbb{1}_{T_c \leq T_c^u}] \leq T\mathbb{P}(T_c \geq T_c^u) + \mathbb{E}[N_a(T_c)\mathbb{1}_{T_c \leq T_c^u}]$$

$$\leq \underbrace{\mathbb{E}[N_a(T_c)\mathbb{1}_{T_c \leq T_c^u}]}_{I_3} + \frac{10eA}{\log^2 T}.$$

Then, bounding $I_3$ is similar to bounding $I_1$ for Theorem. 5. We decompose $I_3$ as follows:

$$I_3 = \mathbb{E}\left[\mathbb{1}_{T_c \leq T_c^u}\sum_{t=1}^{T_c}\mathbb{1}_{A_t = a}\right]$$

$$= 1 + \underbrace{\mathbb{E}\left[\mathbb{1}_{T_c \leq T_c^u}\sum_{t=A+1}^{T_c}\mathbb{1}_{A_t=a}\mathbb{1}_{\mathsf{UCB}_{t-1}(a) \geq \mu_1}\right]}_{A_1} + \underbrace{\mathbb{E}\left[\mathbb{1}_{T_c \leq T_c^u}\sum_{t=A+1}^{T_c}\mathbb{1}_{A_t=a}\mathbb{1}_{\mathsf{UCB}_{t-1}(a) < \mu_1}\right]}_{A_2},$$

In order to bound $A_1$ and $A_2$, we assume there is a virtual process that after the end time $T_c$ of exploration phase, it continues to select the arm with largest UCB and receive the corresponding reward until time $T_c^u$. This is only a virtual process used in our proof, while in reality our algorithm will stop exploration after stopping time $T_c$. We will use $\mathbb{E}'$ and $\mathbb{P}'$ to denote the expectation and probability over this virtual process. Then, we can bound $A_2$ as follows:

$$A_2 \leq \mathbb{E}\left[\mathbb{1}_{T_c \leq T_c^u}\sum_{t=A+1}^{T_c}\mathbb{1}_{\mathsf{UCB}_{t-1}(1) < \mu_1}\right] \leq \mathbb{E}'\left[\sum_{t=A+1}^{T_c^u}\mathbb{1}_{\mathsf{UCB}_{t-1}(1) < \mu_1}\right]$$

$$= \sum_{t=A+1}^{T_c^u}\mathbb{P}'\left(\mathsf{UCB}_{t-1}(1) < \mu_1\right).$$

where the first inequality is because $A_t = a$ indicates $\mathsf{UCB}_{t-1}(1) < \mathsf{UCB}_{t-1}(a)$ due to the dynamic of UCB exploration. The second inequality is because $\mathbb{E}$ and $\mathbb{E}'$ are totally the same for the first $T_c$ time steps. This step also allows us to bound $A_2$ over the events on a different probability measure $\mathbb{E}'$ and $\mathbb{P}'$. Event $\{\mathsf{UCB}_{t-1}(1) < \mu_1\}$ means that at time step $t$, the mean estimation $\bar{r}_{t-1}(1)$ of the optimal arm from previous pulls is lower than $\mu_1 - b_{t-1}(1)$, which incurs a large deviation, so we can bound this event with a union over all time steps:

$$\{\mathsf{UCB}_{t-1}(1) < \mu_1\} \subset \left\{\exists t' \in [A, T_c^u), \mathsf{UCB}_{t'}(1) < \mu_1\right\},$$

which doesn't depend on time step $t$ any more, so we have:

$$\sum_{t=A+1}^{T_c^u}\mathbb{P}'\left(\mathsf{UCB}_{t-1}(1) < \mu_1\right) \leq \sum_{t=A+1}^{T_c^u}\mathbb{P}'\left(\exists t' \in [A, T_c^u), \mathsf{UCB}_{t'}(1) < \mu_1\right)$$

$$= (T_c^u - A)\mathbb{P}'\left(\exists t' \in [A, T_c^u), \mathsf{UCB}_{t'}(1) < \mu_1\right)$$

$$= (T_c^u - A)\mathbb{P}'\left(\exists t' \in [A, T_c^u), \bar{r}_{t'}(1) + \sqrt{\frac{2l}{N_{t'}(1)}} < \mu_1\right).$$

If there is no pulls for arm 1 in a time interval, the empirical mean and number of pulls will remain the same. So instead of counting on the time steps, we can count the number of pulls for arm 1:

$$\left\{\exists t' \in [A, T_c^u), \bar{r}_{t'}(1) + \sqrt{\frac{2l}{N_{t'}(1)}} < \mu_1\right\} \subset \left\{\exists s \in [1, N_{T_c^u}(a)), \bar{r}_{s,1} + \sqrt{\frac{2l}{s}} < \mu_1\right\}.$$

Notice that $N_{T_c^u}(a) \leqslant T_c^u - A + 1$, so we can bound the probability as follows:

$$\mathbb{P}' \left( \exists t' \in [A, T_c^u), \bar{r}_{t'}(1) + \sqrt{\frac{2l}{N_{t'}(1)}} < \mu_1 \right)$$

$$\leqslant \mathbb{P}' \left( \exists s \in [1, T_c^u - A + 1), \bar{r}_{s,1} - \mu_1 + \sqrt{\frac{2l}{s}} < 0 \right)$$

$$\leqslant \mathbb{P}' \left( \exists s \in [1, T_c^u - A + 1), \frac{\sum_{i=1}^s (W_{1,i} - \mu_1)}{i} + \sqrt{\frac{2l}{s}} < 0 \right)$$

$$\leqslant \frac{T_c^u - A}{\exp(l)},$$

where the last inequality uses Eq. (4a) of Lemma 3 and the fact that $W_{1,i} - \mu_1$ is 1-sub-Gaussian with zero mean. Notice that $\exp(l) \geqslant T \log^4 T$, so putting the above bound back to the bound of $A_2$, we have:

$$A_2 \leqslant \frac{\left( \sum_{a:\Delta_a>0} \frac{8l^2}{\Delta_a^2} + Al - A \right)^2}{T \log^4 T} \leqslant \frac{\left( \sum_{a:\Delta_a>0} \frac{200 \log^2 T}{\Delta_a^2} + 5A \log T \right)^2}{T \log^4 T} = \mathcal{O}(T^{-1}).$$

where the first inequality is due to the fact that when $T \geqslant 3$, $\sqrt{\log T} \leqslant \log T$. Next, we attempt to bound $A_1$ as follows:

$$A_1 = \mathbb{E} \left[ \mathbb{1}_{T_c \leqslant T_c^u} \sum_{t=A+1}^{T_c^u} \mathbb{1}_{A_t = a} \mathbb{1}_{\mathsf{UCB}_{t-1}(a) \geqslant \mu_1} \right] \leqslant \mathbb{E}' \left[ \sum_{t=A+1}^{T_c^u} \mathbb{1}_{A_t = a} \mathbb{1}_{\bar{r}_{t-1}(a) + \sqrt{\frac{2l}{N_{t-1}(a)}} \geqslant \mu_1} \right],$$

where the inequality is also due to the fact that the virtual and real processes are identical before time $T_c$. Notice that the number of pulls $N_t(a)$ will only increase by 1 every time there is new pull, i.e., when $A_t = a$. Otherwise, the term inside summation is 0. So instead of counting on the time step $t$, we can count over the number of pulls over arm $a$ as follows:

$$\mathbb{E}' \left[ \sum_{t=A+1}^{T_c^u} \mathbb{1}_{A_t = a} \mathbb{1}_{\bar{r}_{t-1}(a) + \sqrt{\frac{2l}{N_{t-1}(a)}} \geqslant \mu_1} \right] = \mathbb{E}' \left[ \sum_{s=1}^{N_{T_c^u}(a)} \mathbb{1}_{\bar{r}_{s,a} + \sqrt{\frac{2l}{s}} \geqslant \mu_1} \right]$$

$$\leqslant \sum_{s=1}^{T_c^u} \mathbb{P}' \left( \bar{r}_{s,a} - \mu_a + \sqrt{\frac{2l}{s}} \geqslant \Delta_a \right),$$

where the last inequality is due to $N_{T_c^u}(a) \leqslant T_c^u$. Then, by Eq. (4c) of Lemma. 3, we have:

$$\sum_{s=1}^{T_c^u} \mathbb{P}' \left( \bar{r}_{s,a} - \mu_a + \sqrt{\frac{2l}{s}} \geqslant \Delta_a \right) = \sum_{s=1}^{T_c^u} \mathbb{P}' \left( \frac{\sum_{i=1}^s (W_{a,i} - \mu_a)}{i} + \sqrt{\frac{2l}{s}} \geqslant \Delta_a \right)$$

$$\leqslant \frac{2 \left( \log T + 4\sqrt{2 \log T} \right) + \sqrt{4\pi \left( \log T + 4\sqrt{2 \log T} \right) + 2}}{\Delta_a^2} + 1$$

$$\leqslant \frac{2 \log T}{\Delta_a^2} + \frac{(8 + \sqrt{20\pi})\sqrt{\log T}}{\Delta_a^2} + \frac{2}{\Delta_a^2} + 1,$$

where the first inequality is due to the fact that $W_{a,i} - \mu_a$ is 1-sub-Gaussian with zero mean, and the last inequality is due to $\sqrt{2 \log T} \leqslant \log T$ when $T \geqslant 9$. Therefore, combining the bounds on $A_1$ and $A_2$, we can bound $I_3$ as follows:

$$I_3 \leqslant \frac{2 \log T}{\Delta_a^2} + \frac{(8 + \sqrt{20\pi})\sqrt{\log T}}{\Delta_a^2} + \frac{2}{\Delta_a^2} + 1 + \mathcal{O}(T^{-1}).$$

Therefore, a similar bound can be established on $I_1$ as follows:

$$I_1 \leqslant \frac{2 \log T}{\Delta_a^2} + \frac{(8 + \sqrt{20\pi})\sqrt{\log T}}{\Delta_a^2} + \frac{2}{\Delta_a^2} + 1 + \frac{10eA}{\log^2 T} + \mathcal{O}(T^{-1}).$$

**Bounding $I_2$:** Recall that our stopping criterion is when there exists an arm $\tilde{a}$ whose number of pulls is significantly larger than other arms, i.e., $N_{T_c}(\tilde{a}) \geqslant l \max_{a \neq \tilde{a}} N_{T_c}(a)$. Therefore, its bonus $b_{T_c}(\tilde{a})$ should be very small compared to other arms. Also recall that we select the arm $\hat{a}$ with highest LCB to commit to, so the proof follows two steps. First we will show that with high probability the arm $\tilde{a}$ with most number of pulls is the best arm. Then we will show that under this circumstance, the maximum LCB arm is also the best arm. Therefore, consider an arbitrary sub-optimal arm $a$:

$$\mathbb{P}\left(\tilde{a} = a\right) \leqslant \mathbb{P}\left(N_{T_c}(a) \geqslant \lceil lN_{T_c}(1)\rceil + 1\right).$$

Notice that if $N_{T_c}(a) \geqslant \lceil lN_{T_c}(1)\rceil + 1$ at the end time $T_c$, we must have pulled arm $a$ at time step $T_c$. This also means that the number of pulls $N_{T_c-1}(a)$ for arm $a$ after time step $T_c - 1$ is exactly $\lceil lN_{T_c}(1)\rceil$ and and $N_{T_c-1}(1) = N_{T_c}(1)$. It also means that arm $a$ has the largest UCB. So we have:

$$\mathbb{P}\left(N_{T_c}(a) \geqslant \lceil lN_{T_c}(1)\rceil + 1\right)$$
$$\leqslant \mathbb{P}\left(N_{T_c-1}(a) = \lceil lN_{T_c-1}(1)\rceil, \mathsf{UCB}_{T_c-1}(a) \geqslant \mathsf{UCB}_{T_c-1}(1)\right)$$
$$\leqslant \mathbb{P}\left(\exists t \leqslant T_c, N_t(a) = \lceil lN_t(1)\rceil, \bar{r}_t(a) + \sqrt{\frac{2l}{N_t(a)}} \geqslant \bar{r}_t(1) + \sqrt{\frac{2l}{N_t(1)}}\right).$$

Notice that we can separate the two random quantities with union bound as follows:

$$\mathbb{P}\left(N_{T_c}(a) \geqslant \lceil lN_{T_c}(1)\rceil + 1\right)$$

$$\leqslant \underbrace{\mathbb{P}\left(\exists t \leqslant T_c, \bar{r}_t(a) \geqslant \mu_a + \sqrt{\frac{2l}{N_t(a)}}\right)}_{B_1}$$

$$+ \underbrace{\mathbb{P}\left(\exists t \leqslant T_c, N_t(a) = \lceil lN_t(1)\rceil, \mu_a + 2\sqrt{\frac{2l}{N_t(a)}} \geqslant \bar{r}_t(1) + \sqrt{\frac{2l}{N_t(1)}}\right)}_{B_2}.$$

We first bound $B_1$. Notice that we can switch from counting of time step $t$ to count the number of pulls for arm $a$. It is clear that $N_t(a) \leqslant T_c \leqslant T$ when $t \leqslant T_c$, so we have:

$$B_1 \leqslant \mathbb{P}\left(\exists s \leqslant T, \bar{r}_{s,a} \geqslant \mu_a + \sqrt{\frac{2l}{s}}\right)$$
$$= \mathbb{P}\left(\exists s \leqslant T, \frac{\sum_{i=1}^s(\mu_a - W_{a,i})}{s} + \sqrt{\frac{2l}{s}} \leqslant 0\right)$$
$$\leqslant \frac{el\log T + e}{\exp l},$$

where the last inequality uses Eq. (4a) of Lemma. 3. Also notice that $\exp(l) \geqslant T \log^4 T$, so we have:

$$B_1 \leqslant \frac{10e\log^2 T + e}{T\log^4 T} = o\left(\frac{1}{T}\right).$$

Next, we bound the term $B_2$, we can rearrange the terms inside $B_2$ as follows:

$$B_2 = \mathbb{P}\left(\exists t \leqslant T_c, N_t(a) = \lceil lN_t(1)\rceil, \bar{r}_t(1) - \mu_1 + \left(1 - \frac{2}{\sqrt{l}}\right)\sqrt{\frac{2l}{N_t(1)}} + \Delta_a \leqslant 0\right).$$

Denote $\alpha = 1 - \frac{2}{\sqrt{l}}$, and switch the count from time step $t$ to the number of pulls for arm $1$, we can bound $B_2$ as follows:

$$B_2 \leqslant \mathbb{P}\left(\exists s \leqslant T, \bar{r}_{s,1} - \mu_1 + \sqrt{\frac{2\alpha^2 l}{s}} + \Delta_a \leqslant 0\right)$$
$$= \mathbb{P}\left(\exists s \leqslant T, \frac{\sum_{i=1}^s(W_{1,i} - \mu_1)}{s} + \sqrt{\frac{2\alpha^2 l}{s}} + \Delta_a \leqslant 0\right)$$
$$\leqslant \frac{4}{\Delta_a^2 \exp\left(\alpha^2 l\right)}.$$

When $T$ is large enough, we have $\alpha^2 l = l - 4\sqrt{l} + 4 \geqslant \log T + 4$, so we have:

$$B_2 \leqslant \frac{4}{\Delta_a^2 T}.$$

Therefore, combining $B_1$ and $B_2$ together, we can bound the probability as follows:

$$\mathbb{P}\left(\tilde{a} = a\right) \leqslant \mathbb{P}\left(N_{T_c}(a) \geqslant \lceil l N_{T_c}(1) \rceil + 1\right) \leqslant B_1 + B_2 \leqslant \mathcal{O}\left(\frac{1}{T}\right).$$

Recall that arm 1 is the optimal arm. Therefore, by a union bound, we can characterize the probability that of event $\{\tilde{a} \neq 1\}$ as follows:

$$\mathbb{P}\left(\tilde{a} \neq 1\right) \leqslant \sum_{a:\Delta_a > 0} \mathbb{P}\left(\tilde{a} = a\right) \leqslant \mathcal{O}\left(\frac{1}{T}\right).$$

Next, we investigate the probability of choosing the wrong arm $\hat{a}$ to commit to. Recall that for the arm which we commit to, we choose the one with the largest LCB. Consider any sub-optimal arm $a$, if we wrongly choose the arm $\hat{a} = a$, it means its LCB should be larger than the LCB of the optimal arm, which with high probability has the largest number of pulls. So, we have:

$$\mathbb{P}\left(\hat{a} = a\right) \leqslant \mathbb{P}\left(\hat{a} = a, \tilde{a} = 1\right) + \mathbb{P}\left(\tilde{a} \neq 1\right) \leqslant \mathbb{P}\left(\mathsf{LCB}_{T_c}(a) \geqslant \mathsf{LCB}_{T_c}(1), \tilde{a} = 1\right) + \mathcal{O}\left(\frac{1}{T}\right).$$

When $\tilde{a} = 1$, arm 1 has $l$ times more pulls than arm $a$ when the exploration phase stops, so we have:

$$\mathbb{P}\left(\mathsf{LCB}_{T_c}(a) \geqslant \mathsf{LCB}_{T_c}(1), \tilde{a} = 1\right)$$
$$\leqslant \mathbb{P}\left(\bar{r}_{T_c}(a) - \sqrt{\frac{2l}{N_{T_c}(a)}} \geqslant \bar{r}_{T_c}(1) - \sqrt{\frac{2l}{N_{T_c}(1)}}, N_{T_c}(1) \geqslant l N_{T_c}(a)\right).$$

Similarly, we separate the two random variables as follows:

$$\mathbb{P}\left(\mathsf{LCB}_{T_c}(a) \geqslant \mathsf{LCB}_{T_c}(1), \tilde{a} = 1\right)$$
$$\leqslant \mathbb{P}\left(\bar{r}_{T_c}(a) - \sqrt{\frac{2l}{N_{T_c}(a)}} \geqslant \mu_1 - 2\sqrt{\frac{2l}{N_{T_c}(1)}}, N_{T_c}(1) \geqslant l N_{T_c}(a)\right)$$
$$+ \mathbb{P}\left(\bar{r}_{T_c}(1) \leqslant \mu_1 - \sqrt{\frac{2l}{N_{T_c}(1)}}\right)$$
$$\leqslant \underbrace{\mathbb{P}\left((\mu_a - \bar{r}_{T_c}(a)) + \left(1 - \frac{2}{\sqrt{l}}\right)\sqrt{\frac{2l}{N_{T_c}(a)}} + \Delta_a \leqslant 0\right)}_{B_3} + \underbrace{\mathbb{P}\left(\bar{r}_{T_c}(1) \leqslant \mu_1 - \sqrt{\frac{2l}{N_{T_c}(1)}}\right)}_{B_4}.$$

For $B_4$, notice that $T_c$ is a random stopping time, so we bound the probability over all time step when the number of pulls is larger than $l$ as follows:

$$B_4 \leqslant \mathbb{P}\left(\exists s \in [l, T], \bar{r}_{s,1} - \mu_1 + \sqrt{\frac{2l}{s}} \leqslant 0\right) = \mathbb{P}\left(\exists s \in [l, T], \frac{\sum_{i=1}^s (W_{1,i} - \mu_1)}{s} + \sqrt{\frac{2l}{s}} \leqslant 0\right).$$

Since $W_{1,i} - \mu_1$ is 1-subgaussian, we can use Eq. (4a) from the concentration Lemma. 3 to bound $B_4$ as follows:

$$B_4 \leqslant \mathbb{P}\left(\exists s \in [l, T], \frac{\sum_{i=1}^s (W_{1,i} - \mu_1)}{s} + \sqrt{\frac{2l}{s}} \leqslant 0\right) \leqslant \frac{el \log T + e}{\exp l} = o\left(\frac{1}{T}\right),$$

where the last inequality is due to $\sqrt{2 \log T} \leqslant \log T$ when $T \geqslant 9$ and $\exp(l) \geqslant T \log^4 T$. On the other hand, bounding $B_3$ is similar to the proof of probability upper bound regarding $\tilde{a}$. recall that $\alpha = \left(1 - \frac{2}{\sqrt{l}}\right)$, and notice that $T_c$ is a random variable, so we use a union over all possible arm pulls

to bound the event on time step $T_c$ as follows:

$$B_3 \leqslant \mathbb{P}\left(\exists s \leqslant T, (\mu_a - \bar{r}_{s,a}) + \sqrt{\frac{2\alpha^2 l}{s}} + \Delta_a \leqslant 0\right)$$

$$\leqslant \mathbb{P}\left(\exists s \leqslant T, \frac{\sum_{i=1}^{s}(\mu_a - W_{a,i})}{s} + \sqrt{\frac{2\alpha^2 l}{s}} + \Delta_a \leqslant 0\right)$$

$$\leqslant \frac{4}{\Delta_a^2 \exp(\alpha^2 l)},$$

where the last inequality uses the second concentration inequality Eq. (4b) from Lemma. 3 and the fact that $\mu_a - W_{a,i}$ is 1-subgaussian. Then when $T$ is large enough, we have:

$$B_3 \leqslant \frac{4}{\Delta_a^2 T}.$$

Combining $B_3$ and $B_4$, we can bound the probability that we select the wrong arm as:

$$\mathbb{P}\left(\mathsf{LCB}_{T_c}(a) \geqslant \mathsf{LCB}_{T_c}(1), \tilde{a} = 1\right) \leqslant B_3 + B_4 \leqslant \frac{4}{\Delta_a^2 T} + o\left(\frac{1}{T}\right) \leqslant \mathcal{O}\left(\frac{1}{T}\right).$$

Therefore, we finally bound $I_2$ as follows:

$$I_2 = \mathbb{P}(\hat{a} = a) \leqslant \mathbb{P}\left(\hat{a} = a, \tilde{a} = 1\right) + \mathbb{P}\left(\tilde{a} \neq 1\right) = \mathcal{O}\left(\frac{1}{T}\right).$$

Here, the bound on $I_2$ shows that the confidence level of our algorithm is $\mathcal{O}(T^{-1})$ indicated in Corollary. 2. Combining $I_1$ and $I_2$, we can bound $\mathbb{E}[N_a(T)]$ in finite time as follows:

$$\mathbb{E}[N_a(T)] \leqslant I_1 + TI_2 \leqslant \frac{2\log T}{\Delta_a^2} + \frac{(8 + \sqrt{20\pi})\sqrt{\log T}}{\Delta_a^2} + \mathcal{O}(1).$$

So by the regret decomposition lemma, we can bound the regret performance as:

$$\mathsf{Reg}_{\boldsymbol{\mu}}^{\mathsf{EOCP\text{-}UG}}(T) \leqslant \sum_{a:\Delta_a>0}\left(\frac{2\log T}{\Delta_a} + \frac{(8 + \sqrt{20\pi})\sqrt{\log T}}{\Delta_a}\right) + \mathcal{O}(1).$$

### B.4 Proof of Lemma. 4 and Corollary. 2

Consider a specific sub-optimal arm $a$. We first show that it can only be pulled $\mathcal{O}(\log T)$ with high probability during the exploration phase due to the dynamic of our exploration strategy, i.e., the UCB exploration strategy. To be specific, we intend to show the following inequality:

$$\mathbb{P}\left(\exists t \leqslant T_c, N_t(a) \geqslant \frac{8l}{\Delta_a^2} + 1\right) \leqslant \frac{10e}{T\log^2 T}.$$

Since during the exploration phase, we select the arm with the largest UCB to explore, if there exists a time $t$ which the number of pulls for action $a$ is larger than $\frac{8l}{\Delta_a^2} + 1$, which means that there exists a time $t' \in [A, t)$, where at time $t' + 1$ the number of pulls for previous rounds $N_{t'}(a)$ is exactly $\frac{8l}{\Delta_a^2}$ and arm $a$ has the largest UCB. Therefore, we can bound the probability as:

$$\mathbb{P}\left(\exists t \leqslant T_c, N_t(a) \geqslant \frac{8l}{\Delta_a^2} + 1\right) \leqslant \mathbb{P}\left(\exists t' \leqslant t \leqslant T, N_{t'}(a) = \frac{8l}{\Delta_a^2}, a = \arg\max_{a'} \mathsf{UCB}_{t'}(a')\right).$$

Therefore, the UCB of arm $a$ should be larger than the UCB of the optimal arm 1, i.e., $\mathsf{UCB}_{t'}(a) \geqslant \mathsf{UCB}_{t'}(1)$. So we can further derive an upper bound as follows:

$$\mathbb{P}\left(\exists t' \leqslant t \leqslant T, N_{t'}(a) = \left\lceil \frac{8l}{\Delta_a^2} \right\rceil, a = \arg\max_{a'} \mathsf{UCB}_{t'}(a')\right)$$

$$\leqslant \mathbb{P}\left(\exists t' \leqslant T, N_{t'}(a) = \left\lceil \frac{8l}{\Delta_a^2} \right\rceil, \mathsf{UCB}_{t'}(a) \geqslant \mathsf{UCB}_{t'}(1)\right)$$

$$= \mathbb{P}\left(\exists t' \leqslant T, N_{t'}(a) = \left\lceil \frac{8l}{\Delta_a^2} \right\rceil, \bar{r}_{t'}(a) + \sqrt{\frac{2l}{N_{t'}(a)}} \geqslant \bar{r}_{t'}(1) + \sqrt{\frac{2l}{N_{t'}(1)}}\right)$$

$$\leqslant \mathbb{P}\left(\exists t' \leqslant T, \bar{r}_{\left\lceil \frac{8l}{\Delta_a^2} \right\rceil, a} + \frac{\Delta_a}{2} \geqslant \bar{r}_{t'}(1) + \sqrt{\frac{2l}{N_{t'}(1)}}\right).$$

Then, we can separate the two empirical means $\bar{r}_{\left\lceil \frac{8l}{\Delta_a^2} \right\rceil, a}$ and $\bar{r}_{t'}(1)$ with a union bound as follows:

$$\mathbb{P}\left(\exists t' \leqslant T, \bar{r}_{\left\lceil \frac{8l}{\Delta_a^2} \right\rceil, a} + \frac{\Delta_a}{2} \geqslant \bar{r}_{t'}(1) + \sqrt{\frac{2l}{N_{t'}(1)}}\right)$$

$$\leqslant \underbrace{\mathbb{P}\left(\exists t' \leqslant T, \bar{r}_{t'}(1) + \sqrt{\frac{2l}{N_{t'}(1)}} \leqslant \mu_a + \Delta_a\right)}_{A_1} + \underbrace{\mathbb{P}\left(\bar{r}_{\left\lceil \frac{8l}{\Delta_a^2} \right\rceil, a} \geqslant \mu_a + \frac{\Delta_a}{2}\right)}_{A_2}.$$

The term $A_2$ can be easily bounded through Hoeffding's inequality as:

$$A_2 = \mathbb{P}\left(\frac{\sum_{i=1}^{\left\lceil \frac{8l}{\Delta_a^2} \right\rceil}(W_{a,i} - \mu_a)}{\left\lceil \frac{8l}{\Delta_a^2} \right\rceil} \geqslant \frac{\Delta_a}{2}\right) \leqslant \exp\left(-\left\lceil \frac{8l}{\Delta_a^2} \right\rceil \frac{\Delta_a^2}{8}\right) \leqslant \frac{1}{\exp(l)},$$

where the first inequality is due to the fact that $W_{a,i} - \mu_a$ is 1-subgaussian. For term $A_1$, we notice that the empirical estimation $\bar{r}_{t'}(1)$ and the counter $N_{t'}(1)$ will remain the same if there is no pull for arm 1, so we can switch the count of time step $t'$ to the count of number of pulls for $N'_t(1)$. To be specific,

$$A_1 \leqslant \mathbb{P}\left(\exists s \leqslant T, \bar{r}_{s,1} + \sqrt{\frac{2l}{s}} \leqslant \mu_a + \Delta_a\right) = \mathbb{P}\left(\exists s \leqslant T, \frac{\sum_{i=1}^s(W_{1,i} - \mu_1)}{s} + \sqrt{\frac{2l}{s}} \leqslant 0\right).$$

Since $W_{1,i} - \mu_1$ is 1-subgaussian, we can use the concentration result Eq. (4a) of Lemma. 3 to bound $A_1$ as follows:

$$\mathbb{P}\left(\exists s \leqslant T, \frac{\sum_{i=1}^s(W_{1,i} - \mu_1)}{s} + \sqrt{\frac{2l}{s}} \leqslant 0\right) \leqslant \frac{el \log T + e}{\exp(l)}.$$

Therefore, we have can bound the probability of overpull for sub-optimal arm $a$ as follows:

$$\mathbb{P}\left(\exists t \leqslant T_{\mathrm{c}}, N_t(a) \geqslant \frac{8l}{\Delta_a^2} + 1\right) \leqslant A_1 + A_2 \leqslant \frac{e(\log T + 4\sqrt{2 \log T})\log T + 2e}{\exp(l)} \leqslant \frac{10e}{T \log^2 T},$$

where the third inequality is due to $\log T \geqslant \sqrt{2 \log T}$ when $T \geqslant 9$ and the last inequality is due to $\exp(l) \geqslant T \log^4 T$. Recall that in Algorithm. 2, the exploration phase will stop if there exists an arm $\tilde{a}$ whose pulls $N_{T_{\mathrm{c}}}(\tilde{a})$ at the stopping time have exceeds $l$ times of all other arms, i.e.,

$$l \max_{a \neq \tilde{a}} N_{T_{\mathrm{c}}}(a) + 2 \geqslant N_{T_{\mathrm{c}}}(\tilde{a}) > l \max_{a \neq \tilde{a}} N_{T_{\mathrm{c}}}(a) + 1.$$

So if $T_c$ is larger than $\sum_{a:\Delta_a>0} \frac{8(l+1)^2}{\Delta_a^2} + A(l+2)$, it means that there exists at least one sub-optimal arm $a$ whose pulls $N_{T_c}(a)$ is larger than $\frac{8l}{\Delta_a^2} + 1$. So we have:

$$\mathbb{P}\left(T_c \geqslant \sum_{a:\Delta_a>0} \frac{8(l+1)^2}{\Delta_a^2} + A(l+2)\right) \leqslant \mathbb{P}\left(\exists a : \Delta_a > 0, N_{T_c}(a) \geqslant \frac{8l}{\Delta_a^2} + 1\right)$$

$$\leqslant \sum_{a:\Delta_a>0} \mathbb{P}\left(\exists t \leqslant T_c, N_t(a) \geqslant \frac{8l}{\Delta_a^2} + 1\right)$$

$$\leqslant \frac{10eA}{T\log^2 T},$$

where the second inequality, we use union bound over $a$. Then Corollary. 2 can be easily proved with:

$$\mathsf{SCC}_{\boldsymbol{\mu}}^{\mathsf{EOCP\text{-}UG}}(T) = \mathbb{E}[T_c]$$

$$= \mathbb{E}\left[T_c \mathbb{1}_{T_c \geqslant \sum_{a:\Delta_a>0} \frac{8(l+1)^2}{\Delta_a^2} + A(l+2)}\right] + \mathbb{E}\left[T_c \mathbb{1}_{T_c < \sum_{a:\Delta_a>0} \frac{8(l+1)^2}{\Delta_a^2} + A(l+2)}\right]$$

$$\leqslant T\mathbb{P}\left(T_c \geqslant \sum_{a:\Delta_a>0} \frac{8(l+1)^2}{\Delta_a^2} + A(l+2)\right) + \sum_{a:\Delta_a>0} \frac{8(l+1)^2}{\Delta_a^2} + A(l+2)$$

$$\leqslant \sum_{a:\Delta_a>0} \frac{8\log^2 T + 80\log^{\frac{3}{2}} T + 200\log T}{\Delta_a^2} + 6A\log T + \frac{10eA}{\log^2 T}.$$

Therefore, taking $T$ to infinity, we have:

$$\limsup_{T\to\infty} \frac{\mathsf{SCC}_{\boldsymbol{\mu}}^{\mathsf{EOCP\text{-}UG}}(T)}{\log^2 T} = \sum_{a:\Delta_a>0} \frac{8}{\Delta_a^2}.$$

### B.5 Proof of Theorem. 3

We first prove the fundamental limits of sample complexity until commitment in the pre-determined stopping time setting. Suppose the bandit problem (instance) is as follows: the reward expectation of action 1 is $\mu_1$ and the reward expectation of action 2 is $\mu_2$. We assume $\mu_1 > \mu_2$ and $\Delta = \mu_1 - \mu_2$. Consider another bandit instance with reward expectations $\lambda_1$ and $\lambda_2$ for the two actions respectively, and $\lambda_1 + \Delta \leqslant \lambda_2$. By the "transportation" lemma [32, Lemma. 1] when $T \geqslant 10$, we have for any stopping time $\tau$ that:

$$\mathbb{E}_{\boldsymbol{\mu}}[N_1(\tau)]\mathsf{KL}(\mu_1, \lambda_1) + \mathbb{E}_{\boldsymbol{\mu}}[N_2(\tau)]\mathsf{KL}(\mu_2, \lambda_2) \geqslant \log\left(\frac{T}{2.4}\right).$$

Since under Gaussian bandit, the KL divergence is simply the squared norm, we have:

$$\mathbb{E}_{\boldsymbol{\mu}}[N_1(\tau)]\frac{(\mu_1 - \lambda_1)^2}{2} + \mathbb{E}_{\boldsymbol{\mu}}[N_2(\tau)]\frac{(\mu_2 - \lambda_2)^2}{2} \geqslant \log\left(\frac{T}{2.4}\right).$$

Since the inequality holds for any $\lambda_1$ and $\lambda_2$ such that that $\lambda_1 + \Delta \leqslant \lambda_2$, we can minimize the LHS to obtain tighter bounds. So minimizing the LHS over $\lambda_1$ and $\lambda_2$ gives:

$$\lambda_1 = \frac{\mu_1\mathbb{E}_{\boldsymbol{\mu}}[N_1(\tau)] + (\mu_2 - \Delta)\mathbb{E}_{\boldsymbol{\mu}}[N_2(\tau)]}{\mathbb{E}_{\boldsymbol{\mu}}[N_1(\tau)] + \mathbb{E}_{\boldsymbol{\mu}}[N_2(\tau)]}, \quad \lambda_2 = \frac{(\mu_1 + \Delta)\mathbb{E}_{\boldsymbol{\mu}}[N_1(\tau)] + \mu_2\mathbb{E}_{\boldsymbol{\mu}}[N_2(\tau)]}{\mathbb{E}_{\boldsymbol{\mu}}[N_1(\tau)] + \mathbb{E}_{\boldsymbol{\mu}}[N_2(\tau)]}.$$

So plug the optimization result into the "transportation" lemma, we have:

$$\frac{2\Delta^2\mathbb{E}_{\boldsymbol{\mu}}[N_2(\tau)]^2}{(\mathbb{E}_{\boldsymbol{\mu}}[N_1(\tau)] + \mathbb{E}_{\boldsymbol{\mu}}[N_2(\tau)])^2} + \frac{2\Delta^2\mathbb{E}_{\boldsymbol{\mu}}[N_1(\tau)]^2}{(\mathbb{E}_{\boldsymbol{\mu}}[N_1(\tau)] + \mathbb{E}_{\boldsymbol{\mu}}[N_2(\tau)])^2} \geqslant \log\left(\frac{T}{2.4}\right).$$

Therefore, rearranging the terms in the inequality, we can derive a lower bound on $\mathbb{E}_{\boldsymbol{\mu}}[N_1(\tau)]$ as follows:

$$\mathbb{E}_{\boldsymbol{\mu}}[N_1(\tau)] \geqslant \frac{2\log\left(\frac{T}{2.4}\right)\mathbb{E}_{\boldsymbol{\mu}}[N_2(\tau)]}{2\Delta^2\mathbb{E}_{\boldsymbol{\mu}}[N_2(\tau)] - 2\log\left(\frac{T}{2.4}\right)}.$$

Let $\tau = T_c$ be the fixed length of exploration. Thus, we can derive the following lower bound on the expectation of stopping time as follows:

$$\mathbb{E}_{\boldsymbol{\mu}}[T_c] = \mathbb{E}_{\boldsymbol{\mu}}[N_1(T_c)] + \mathbb{E}_{\boldsymbol{\mu}}[N_2(T_c)] \geqslant \frac{2\Delta^2 \left(\mathbb{E}_{\boldsymbol{\mu}}[N_2(T_c)]\right)^2}{2\Delta^2 \mathbb{E}_{\boldsymbol{\mu}}[N_2(T_c)] - 2\log\left(\frac{T}{2.4}\right)}.$$

According to our assumption, the algorithm has $c$-logarithm regret violation with $\mathcal{O}(T^{-1})$ confidence, which means the the number of pulls for the sub-optimal action is upper and lower bounded when $T$ is large enough. So we have:

$$\left| \mathbb{E}_{\boldsymbol{\mu}}\left[N_2(T)\right] - \frac{2\log(T)}{\Delta^2} \right| = \mathcal{O}(\log^c T).$$

Recall that $\hat{a}$ is the action the algorithm chooses to commit to and action $1$ is the optimal action in instance $\boldsymbol{\mu}$, so we have $\mathbb{E}\left[N_2(T)\right] \leqslant \mathbb{E}_{\boldsymbol{\mu}}\left[N_2(T_c)\right] + T\mathbb{P}_{\boldsymbol{\mu}}\left(\hat{a} \neq 1\right)$. Since the algorithm has $\mathcal{O}(T^{-1})$ confidence, so we have $\mathbb{P}_{\boldsymbol{\mu}}\left(\hat{a} \neq 1\right) = \mathcal{O}(T^{-1})$ and $\mathbb{E}_{\boldsymbol{\mu}}\left[N_2(T)\right] \leqslant \mathbb{E}_{\boldsymbol{\mu}}\left[N_2(T_c)\right] + \mathcal{O}(1)$. So we can lower bound the numerator as follows:

$$\begin{aligned}
2\Delta^2 \left(\mathbb{E}_{\boldsymbol{\mu}}[N_2(T_c)]\right)^2 &\geqslant 2\Delta^2 \left(\mathbb{E}_{\boldsymbol{\mu}}[N_2(T)] - \mathcal{O}(1)\right)^2 \\
&\geqslant 2\Delta^2 \left(\frac{4}{\Delta^4}\log^2 T - \mathcal{O}(\log^{1+c} T)\right) \\
&= \frac{8}{\Delta^2}\log^2 T - \mathcal{O}(\log^{1+c} T).
\end{aligned}$$

Notice that $\mathbb{E}_{\boldsymbol{\mu}}[N_2(T_c)] \leqslant \mathbb{E}_{\boldsymbol{\mu}}[N_2(T)]$. For the denominator, we can derive an upper bound similarly as follows:

$$\begin{aligned}
2\Delta^2 \mathbb{E}_{\boldsymbol{\mu}}[N_2(T_c)] - 2\log\left(\frac{T}{2.4}\right) &\leqslant 2\Delta^2 \mathbb{E}_{\boldsymbol{\mu}}[N_2(T)] - 2\log\left(\frac{T}{2.4}\right) \\
&\leqslant 2\Delta^2 \frac{2}{\Delta^2}\log T + \mathcal{O}(\log^c T) - 2\log\left(\frac{T}{2.4}\right) \\
&\leqslant 2\log T + \mathcal{O}(\log^c T),
\end{aligned}$$

where the second inequality uses the fact that the algorithm has $c$-logarithm regret violation. So combining both bounds in the numerator and the denominator, we have:

$$\mathbb{E}_{\boldsymbol{\mu}}[T_c] \geqslant \frac{\frac{8}{\Delta^2}\log^2 T - \mathcal{O}(\log^{1+c} T)}{2\log T + \mathcal{O}(\log^c T)} = \Omega\left(\frac{\log T}{\Delta^2}\right),$$

which concludes the proof for the pre-determined stopping time setting. In the adaptive stopping time setting, we can use the same procedure to prove the lower bound of sample complexity until commitment, so we also create another bandit instance with reward expectations $\lambda_1$ and $\lambda_2$. Unlike in the pre-determined stopping time setting, now $\lambda_1$ and $\lambda_2$ satisfies $\lambda_1 \leqslant \lambda_2$. We also use the "transportation" lemma and optimize over $\lambda_1$ and $\lambda_2$ to get a lower bound for $\mathbb{E}[T_c]$ as follows:

$$\mathbb{E}_{\boldsymbol{\mu}}[T_c] = \mathbb{E}_{\boldsymbol{\mu}}[N_1(T_c)] + \mathbb{E}_{\boldsymbol{\mu}}[N_2(T_c)] \geqslant \frac{\Delta^2 \left(\mathbb{E}_{\boldsymbol{\mu}}[N_2(T_c)]\right)^2}{\Delta^2 \mathbb{E}_{\boldsymbol{\mu}}[N_2(T_c)] - 2\log\left(\frac{T}{2.4}\right)}.$$

Similar to the proof of pre-determined stopping time setting, we utilize the fact that $\mathbb{E}_{\boldsymbol{\mu}}\left[N_2(T_c)\right] \leqslant \mathbb{E}_{\boldsymbol{\mu}}\left[N_2(T)\right] \leqslant \mathbb{E}_{\boldsymbol{\mu}}\left[N_2(T_c)\right] + \mathcal{O}(1)$ and the algorithm has $c$-logarithm regret violation to bound the numerator and denominators separately. For the numerator, we have:

$$\begin{aligned}
\Delta^2 \left(\mathbb{E}_{\boldsymbol{\mu}}[N_2(T_c)]\right)^2 &\geqslant \Delta^2 \left(\mathbb{E}_{\boldsymbol{\mu}}[N_2(T)] - \mathcal{O}(1)\right)^2 \\
&\geqslant \Delta^2 \left(\frac{4}{\Delta^4}\log^2 T - \mathcal{O}(\log^{1+c} T)\right) \\
&= \frac{4}{\Delta^2}\log^2 T - \mathcal{O}(\log^{1+c} T).
\end{aligned}$$

For the denominator, we use the fact that the algorithm has $c$-logarithm regret violation:

$$\begin{aligned}
\Delta^2 \mathbb{E}_{\boldsymbol{\mu}}[N_2(T_c)] - 2\log\left(\frac{T}{2.4}\right) &\leqslant \Delta^2 \frac{2}{\Delta^2}\log T + \mathcal{O}(\log^c T) - 2\log\left(\frac{T}{2.4}\right) \\
&\leqslant \mathcal{O}(\log^c T).
\end{aligned}$$

So combining both bounds in the numerator and the denominator, we have:

$$\mathbb{E}_{\boldsymbol{\mu}}[T_c] \geqslant \frac{\frac{4}{\Delta^2}\log^2 T - \mathcal{O}(\log^{1+c} T)}{\mathcal{O}(\log^c T)} = \Omega\left(\frac{\log^{2-c} T}{\Delta^2}\right).$$

## C   Proofs of Main Results for KL-EOCP

Now we are ready to prove the regret and sample complexity until commitment results for our KL-EOCP Algorithms, i.e., Algorithm. 3. Without loss of generality, suppose action 1 is the unique optimal action. Recall that for any time index $t$, $\mathsf{UCB}_t(a)$ and $\mathsf{LCB}_t(a)$ denotes be KL upper and lower confidence bound of action $a$ as:

$$\mathsf{UCB}_t(a) = \underset{\mu \geqslant \bar{r}_t(a)}{\arg\max}\left\{N_t(a)\mathsf{KL}(\bar{r}_t(a), \mu) \leqslant l\right\}, \quad \mathsf{LCB}_t(a) = \underset{\mu \leqslant \bar{r}_t(a)}{\arg\min}\left\{N_t(a)\mathsf{KL}(\bar{r}_t(a), \mu) \leqslant l\right\}.$$

Throughout the proof section, we let $W_{a,i}$ be the $i$-th sample from pulling arm $a$ the $i$-th time. Recall that $T_c$ is the length of exploration phase. Let $\bar{r}_{a,s}$ be the empirical mean of arm $a$ after it has been pulled $s$ times. Moreover, we let:

$$\mathsf{UCB}_{s,a} = \underset{\mu \geqslant \bar{r}_{s,a}}{\arg\max}\left\{s\mathsf{KL}(\bar{r}_{s,a}, \mu) \leqslant l\right\}, \quad \mathsf{LCB}_{s,a} = \underset{\mu \leqslant \bar{r}_{s,a}}{\arg\min}\left\{s\mathsf{KL}(\bar{r}_{s,a}, \mu) \leqslant l\right\}.$$

In order to incorporate this new definition of UCB and LCB, we need new sets of concentration inequalities, which will be summarized in the next subsection. The proof will be delayed to Sec. D.

### C.1   Concentration Inequalities for Natural Exponential Families

Since we no longer assume the distributions $[\nu_1, \cdots, \nu_A]$ are subgaussian any more, the concentration inequalities from Sec. B.1, especially Lemma. 3, no longer holds. Therefore, we need a set of new concentrations specialized in the natural exponential family regime. To be specific, we want to bound the probability that the KL divergence of empirical estimations and the true expectation is very large. First, we provide a concentration lemma which is analogous to the Hoeffding's inequality widely used in the subgaussian scenario as follows:

**Lemma 5** *Let $W_1, W_2, \cdots, W_T$ be identically and independently distributed random variables with common expectation $\mu = \mathbb{E}[W_1]$ and sampled from a distribution $\nu$ that belongs to a canonical exponential family $\mathcal{P}$. For any $s \leqslant T$, Let $S_s = \sum_{i=1}^{s} W_i$ and $\bar{\mu}_s = \frac{S_s}{s}$ denote the sum and the empirical mean of the first $s$ samples. For any $\delta > 0$, we have:*

$$\mathbb{P}\left(\bar{\mu}_s \leqslant \mu, s\mathsf{KL}\left(\bar{\mu}_s, \mu\right) \geqslant \delta\right) \leqslant \exp(-\delta), \tag{5a}$$

$$\mathbb{P}\left(\bar{\mu}_s \geqslant \mu, s\mathsf{KL}\left(\bar{\mu}_s, \mu\right) \geqslant \delta\right) \leqslant \exp(-\delta). \tag{5b}$$

Based on Lemma. 5 we propose the following lemmas characterizing the any-time concentration property of random variables.

**Lemma 6** *Let $W_1, W_2, \cdots, W_T$ be identically and independently distributed random variables with common expectation $\mu = \mathbb{E}[W_1]$ and sampled from a distribution $\nu$ that belongs to a canonical exponential family $\mathcal{P}$. For any $s \leqslant T$, Let $S_s = \sum_{i=1}^{s} W_i$ and $\quad \bar{\mu}_s = \frac{S_s}{s}$ denote the sum and the empirical mean of the first $s$ samples. Let $T_1 \leqslant T_2 \leqslant T$ be two real numbers in $\mathbb{R}^+$. For any $l > 2$, the following holds:*

$$\mathbb{P}\left(\exists s \in [T_1, T_2], \bar{\mu}_s \leqslant \mu, s\mathsf{KL}\left(\bar{\mu}_s, \mu\right) \geqslant l\right) \leqslant \min\left\{\frac{T_2 - T_1 + 1}{\exp(l)}, \frac{el\left(\log T_2 - \log T_1\right) + e}{\exp(l)}\right\}, \tag{6a}$$

$$\mathbb{P}\left(\exists s \in [T_1, T_2], \bar{\mu}_s \geqslant \mu, s\mathsf{KL}\left(\bar{\mu}_s, \mu\right) \geqslant l\right) \leqslant \min\left\{\frac{T_2 - T_1 + 1}{\exp(l)}, \frac{el\left(\log T_2 - \log T_1\right) + e}{\exp(l)}\right\}. \tag{6b}$$

**Lemma 7** *Let $W_1, W_2, \cdots, W_T$ be identically and independently distributed random variables with common expectation $\mu = \mathbb{E}[W_1]$ and sampled from a distribution $\nu$ that belongs to a canonical*

exponential family $\mathcal{P}$. For any $s \leqslant T$, Let $S_s = \sum_{i=1}^s W_i$ and $\bar{\mu}_s = \frac{S_s}{s}$ denote the sum and the empirical mean of the first $s$ samples. Let $\mathsf{UCB}_s = \arg\max_{\mu \geqslant \bar{r}_s} \{s\mathsf{KL}(\bar{\mu}_s, \mu) \leqslant l\}$ be the upper confidence bound for empirical mean. For any $l > 2$, any $T_1 \leqslant T$, any $\mu' > \mu$, and any $\varepsilon > 0$, the following holds:

$$\sum_{s=1}^{T_1} \mathbb{P}\left(\mathsf{UCB}_s \geqslant \mu'\right) \leqslant \frac{(1+\varepsilon)l}{\mathsf{KL}(\mu, \mu')} + \frac{\beta_2(\varepsilon)}{T^{\beta_1(\varepsilon)}}, \tag{7}$$

where $\beta_1(\varepsilon) = \mathcal{O}(\varepsilon^2)$ and $\beta_2(\varepsilon) = \mathcal{O}(\varepsilon^{-2})$ are constants.

## C.2 Proof of Regret Optimality for KL-EOCP

In this section, we provide a complete proof of Theorem. 4. We prove the following Theorem which characterizes the finite-time performance of Algorithm. 3, and Theorem. 4 can be derived directly.

**Theorem 7** *If $l = \log(T) + 4\sqrt{2\log(T)}$ and $T \geqslant \max\{16, A, 8l\mathsf{KL}_{\mathrm{lb}}^{-2}\}$, the regret of* KL-EOCP *algorithm in Algorithm. 3 can be upper bounded as:*

$$\mathsf{Reg}_{\boldsymbol{\mu}}^{\mathsf{KL\text{-}EOCP}}(T) \leqslant \sum_{a:\Delta_a > 0} \left( \frac{\Delta_a \log T}{\mathsf{KL}(\mu_a, \mu_1)} + \frac{10\Delta_a \log^{\frac{3}{4}} T}{\mathsf{KL}(\mu_a, \mu_1)} \right) + o(1).$$

**Remark:** The asymptotic result is clear from Theorem.7 by letting $T$ increases to infinity, i.e.,

$$\limsup_{T \to \infty} \frac{\mathsf{Reg}_{\boldsymbol{\mu}}^{\mathsf{KL\text{-}EOCP}}(T)}{\log T} \leqslant \sum_{a:\Delta_a > 0} \frac{\Delta_a}{\mathsf{KL}(\mu_a, \mu_1)}.$$

**Proof.** Without loss of generality, let arm 1 be the unique best arm. From the regret decomposition lemma [38, Lemma 4.5], we can decompose the regret of Algorithm. 3 to the number of pulls for each sub-optimal arm as follows:

$$\mathsf{Reg}_{\boldsymbol{\mu}}(T) = \sum_{a:\Delta_a > 0} \Delta_a \mathbb{E}[N_T(a)].$$

Then, the key to bound the total regret is to bound the number of pulls for each sub-optimal arms. Since our KL-EOCP algorithm has a clear separation of exploration and exploitation phases, so for any sub-optimal action $a$, we can bound its pulls in different phases as follows:

$$\mathbb{E}[N_a(T)] = \underbrace{\mathbb{E}\left[N_a(T_c)\right]}_{I_1} + \underbrace{(T - T_c)\,\mathbb{P}(\hat{a} = a)}_{I_2},$$

where recall that $T_c$ is the end of exploration phase and $\hat{a}$ is the arm that the algorithm commits to during exploitation phase.

**Bounding $I_1$:** We first bound $I_1$, which is similar to the proof of bounding the regret for the traditional KL-UCB algorithm [12]. We have:

$$I_1 = \mathbb{E}\left[\sum_{t=1}^{T_c} \mathbb{1}_{A_t = a}\right] = 1 + \underbrace{\mathbb{E}\left[\sum_{t=A+1}^{T_c} \mathbb{1}_{A_t = a}\mathbb{1}_{\mathsf{UCB}_{t-1}(a) \geqslant \mu_1}\right]}_{A_1} + \underbrace{\mathbb{E}\left[\sum_{t=A+1}^{T_c} \mathbb{1}_{A_t = a}\mathbb{1}_{\mathsf{UCB}_{t-1}(a) < \mu_1}\right]}_{A_2},$$

Notice that $A_t = a$ indicates $\mathsf{UCB}_{t-1}(1) < \mathsf{UCB}_{t-1}(a)$ due to the dynamic of UCB exploration, we can bound the last term as:

$$A_2 \leqslant \sum_{t=A+1}^{T_c} \mathbb{E}\left[\mathbb{1}_{\mathsf{UCB}_{t-1}(1) < \mu_1}\right] = \sum_{t=A+1}^{T_c} \mathbb{P}\left(\mathsf{UCB}_{t-1}(1) < \mu_1\right).$$

Event $\{\mathsf{UCB}_{t-1}(1) < \mu_1\}$ means that at time step $t$, the mean estimation $\bar{r}_{t-1}(1)$ of the optimal arm from previous pulls is lower than $\mu_1 - b_{t-1}(1)$, which incurs a large deviation, so we can bound this event with a union over all time steps:

$$\{\mathsf{UCB}_{t-1}(1) < \mu_1\} \subset \left\{\exists t' \in [A, T_c), \mathsf{UCB}_{t'}(1) < \mu_1\right\},$$

which doesn't depend on time step $t$ any more, so we have:

$$\sum_{t=A+1}^{T_c} \mathbb{P}\left(\mathsf{UCB}_{t-1}(1) < \mu_1\right) \leqslant \sum_{t=A+1}^{T_c} \mathbb{P}\left(\exists t' \in [A, T_c), \mathsf{UCB}_{t'}(1) < \mu_1\right)$$
$$= (T_c - A)\mathbb{P}\left(\exists t' \in [A, T_c), \mathsf{UCB}_{t'}(1) < \mu_1\right).$$

For two time steps $T_c < t_2$, if there is no pulls for arm 1 between them, then the term $\mathsf{UCB}_{t'}(1)$ will remain the same through $t' \in [T_c, t_2]$. So instead of counting on the time steps, we can count the number of pulls for arm 1 instead. We have:

$$\left\{\exists t' \in [A, T_c), \mathsf{UCB}_{t'}(1) < \mu_1\right\} \subset \left\{\exists s \in [1, N_{T_c}(a)), \mathsf{UCB}_{s,1} < \mu_1\right\}.$$

Notice that $N_{T_c}(a) \leqslant T_c - A + 1$, so we can bound the probability as follows:

$$\mathbb{P}\left(\exists t' \in [A, T_c), \mathsf{UCB}_{t'}(1) < \mu_1\right) \leqslant \mathbb{P}\left(\exists s \in [1, T_c - A + 1), \mathsf{UCB}_{s,1} < \mu_1\right)$$
$$= \mathbb{P}\left(\exists s \in [1, T_c - A + 1), \max_{\mu \geqslant \bar{r}_{s,1}}\left\{\mathsf{KL}(\bar{r}_{s,1}, \mu) \leqslant \frac{l}{s}\right\} < \mu_1\right).$$

Notice that for any $s$ and under event $\{\mathsf{UCB}_{s,1} < \mu_1\}$, we have $\mu_1 \geqslant \mathsf{UCB}_{s,1} \geqslant \bar{r}_{s,1}$. By definition of $\mathsf{UCB}_{s,1}$, we also have $\mathsf{KL}\left(\bar{r}_{s,1}, \mathsf{UCB}_{s,1}\right) = \frac{l}{s}$. Therefore, we come to the conclusion that $\mathsf{KL}\left(\bar{r}_{s,1}, \mu\right) \geqslant \frac{l}{s}$. using Eq. (6a) of Lemma 6, we have:

$$\mathbb{P}\left(\exists s \in [1, T_c - A + 1), \max_{\mu \geqslant \bar{r}_{s,1}}\left\{\mathsf{KL}(\bar{r}_{s,1}, \mu) \leqslant \frac{l}{s}\right\} < \mu_1\right)$$
$$\leqslant \mathbb{P}\left(\exists s \in [1, T_c - A + 1), s\mathsf{KL}\left(\bar{r}_{s,1}, \mu\right) \geqslant l\right)$$
$$\leqslant \frac{T_c - A}{\exp(l)}.$$

Using the fact that $\exp(l) \geqslant T \log^4 T$ when $T \geqslant 16$ and putting the above bound back to the bound of $A_2$, we have:

$$A_2 \leqslant \frac{(T_c - A)^2}{T \log^4 T} = \frac{\left(\frac{8A\left(\log T + 4\sqrt{2\log T}\right)}{\mathsf{KL}_{\mathrm{lb}}^2}\right)^2}{T \log^2 T} = o\left(\frac{1}{T}\right),$$

where the last inequality is due to the fact that when $T \geqslant 3$, $\sqrt{\log T} \leqslant \log T$. Next, we attempt to bound the middle term $A_1$ as follows:

$$A_1 = \mathbb{E}\left[\sum_{t=A+1}^{T_c} \mathbb{1}_{A_t=a} \mathbb{1}_{\mathsf{UCB}_{t-1}(a) \geqslant \mu_1}\right] = \mathbb{E}\left[\sum_{t=A+1}^{T_c} \mathbb{1}_{A_t=a} \mathbb{1}_{\mathsf{UCB}_{N_{t-1}(a),a} \geqslant \mu_1}\right].$$

Notice that the number of pulls $N_{t-1}(a)$ will only increase by 1 every time there is new pull, i.e., when $A_t = a$. Otherwise, the term inside summation is 0. So instead of counting on the time step $t$, we can count over the number of pulls over arm $a$ as follows:

$$\mathbb{E}\left[\sum_{t=A+1}^{T_c} \mathbb{1}_{A_t=a} \mathbb{1}_{\mathsf{UCB}_{N_{t-1}(a),a} \geqslant \mu_1}\right] = \mathbb{E}\left[\sum_{s=1}^{N_{T_c}(a)} \mathbb{1}_{\mathsf{UCB}_{s,a} \geqslant \mu_1}\right] \leqslant \sum_{s=1}^{T_c} \mathbb{P}\left(\mathsf{UCB}_{s,a} \geqslant \mu_1\right),$$

where the last inequality is due to $N_{T_c}(a) \leqslant T_c$. Using Lemma. 7, we can bound the sum of probabilities. Specifically, for any $\varepsilon > 0$, there exists two constants $\beta_1(\varepsilon) = \mathcal{O}(\varepsilon^2)$ and $\beta_2(\varepsilon) = \mathcal{O}(\varepsilon^{-2})$ such that:

$$\sum_{s=1}^{T_c} \mathbb{P}\left(\mathsf{UCB}_{s,a} \geqslant \mu_1\right) \leqslant \frac{(1+\varepsilon)l}{\mathsf{KL}(\mu_a, \mu_1)} + \frac{\beta_2(\varepsilon)}{T^{\beta_1(\varepsilon)}}.$$

Let $\epsilon = \log^{-\frac{1}{4}} T$, and when $T \geqslant 16$, we have $\log T \geqslant \sqrt{\log T}$ and $4\sqrt{2\log T} \leqslant 5 \log^{\frac{3}{4}} T$, so we have:

$$\sum_{s=1}^{T_c} \mathbb{P}\left(\mathsf{UCB}_{s,a} \geqslant \mu_1\right) \leqslant \frac{\log T}{\mathsf{KL}(\mu_a, \mu_1)} + \frac{10 \log^{\frac{3}{4}} T}{\mathsf{KL}(\mu_a, \mu_1)} + o(1).$$

Therefore, combining the bounds on $A_1$ and $A_2$, we can bound $I_1$ as follows:

$$I_1 \leqslant \frac{\log T}{\mathsf{KL}(\mu_a, \mu_1)} + \frac{10 \log^{\frac{3}{4}} T}{\mathsf{KL}(\mu_a, \mu_1)} + o(1).$$

**Bounding $I_2$:** After the exploration phase, recall that we select the arm with largest LCB to commit to. The idea of proof is very similar to the regret bound in [2] and our proof of Theorem. 5. We can first bound the probability of selecting a sub-optimal arm $a$ as:

$$\mathbb{P}(\hat{a} = a) \leqslant \mathbb{P}\left(\mathsf{LCB}_{T_c}(a) \geqslant \mathsf{LCB}_{T_c}(1)\right)$$

$$= \mathbb{P}\left(\mathsf{LCB}_{T_c}(a) \geqslant \mathsf{LCB}_{T_c}(1), N_{T_c}(1) > \frac{4l}{\mathsf{KL}_{\mathrm{lb}}}\right)$$

$$+ \mathbb{P}\left(\mathsf{LCB}_{T_c}(a) \geqslant \mathsf{LCB}_{T_c}(1), N_{T_c}(1) \leqslant \frac{4l}{\mathsf{KL}_{\mathrm{lb}}}\right)$$

$$\leqslant \underbrace{\mathbb{P}\left(\mathsf{LCB}_{T_c}(a) \geqslant \mathsf{LCB}_{T_c}(1), N_{T_c}(1) > \frac{4l}{\mathsf{KL}_{\mathrm{lb}}}\right)}_{B_1} + \underbrace{\mathbb{P}\left(N_{T_c}(1) \leqslant \frac{4l}{\mathsf{KL}_{\mathrm{lb}}}\right)}_{B_2}.$$

We first bound the term $B_2$ which states that during the first exploration phase with UCB exploration, the optimal arm is under-pulled, which means that one of the sub-optimal arms have been pulled with larger number of times. Therefore, we have:

$$\mathbb{P}\left(N_{T_c}(1) \leqslant \frac{4l}{\mathsf{KL}_{\mathrm{lb}}}\right) = \mathbb{P}\left(\sum_{a:\Delta_a > 0} N_{T_c}(a) > T_c - \frac{4l}{\mathsf{KL}_{\mathrm{lb}}}\right).$$

Recall that $T_c = \frac{4Al}{\mathsf{KL}_{\mathrm{lb}}} + A$, so we have:

$$\mathbb{P}\left(N_{T_c}(1) \leqslant \frac{4l}{\mathsf{KL}_{\mathrm{lb}}}\right) = \mathbb{P}\left(\sum_{a:\Delta_a > 0} N_{T_c}(a) > \frac{4(A-1)l}{\mathsf{KL}_{\mathrm{lb}}} + A\right)$$

$$\leqslant \mathbb{P}\left(\exists a > 1, N_{T_c}(a) > \frac{4l}{\mathsf{KL}_{\mathrm{lb}}} + 1\right).$$

Then by union bound, we have:

$$\mathbb{P}\left(\exists a > 1, N_{T_c}(a) > \frac{4l}{\mathsf{KL}_{\mathrm{lb}}} + 1\right) \leqslant \sum_{a:\Delta_a > 0} \mathbb{P}\left(N_{T_c}(a) > \frac{4l}{\mathsf{KL}_{\mathrm{lb}}} + 1\right).$$

Consider a fixed sub-optimal arm $a$, then for each probability inside the summation, we have:

$$\mathbb{P}\left(N_{T_c}(a) > \frac{4l}{\mathsf{KL}_{\mathrm{lb}}} + 1\right)$$

$$\leqslant \mathbb{P}\left(\exists t \in [A+1, T_c], N_{t-1}(a) = \left\lceil \frac{4l}{\mathsf{KL}_{\mathrm{lb}}} \right\rceil, A_t = a\right)$$

$$\leqslant \mathbb{P}\left(\exists t \in [A+1, T_c], N_{t-1}(a) = \left\lceil \frac{4l}{\mathsf{KL}_{\mathrm{lb}}} \right\rceil, \mathsf{UCB}_{t-1}(1) \leqslant \mathsf{UCB}_{t-1}(a)\right)$$

$$\leqslant \underbrace{\mathbb{P}\left(\exists t \in [A+1, T_c], N_{t-1}(a) = \left\lceil \frac{4l}{\mathsf{KL}_{\mathrm{lb}}} \right\rceil, \mu_1 \leqslant \mathsf{UCB}_{t-1}(a)\right)}_{B_3}$$

$$+ \underbrace{\mathbb{P}\left(\exists t \in [A+1, T_c], \mathsf{UCB}_{t-1}(1) \leqslant \mu_1\right)}_{B_4}.$$

Notice that $B_4$ can be bounded from concentration lemma. First, we switch the count from time step $t$ to the number of pulls for arm 1 since the empirical estimation $\bar{r}_{t-1}(1)$ and count for pulls $N_{t-1}(1)$ won't change unless there is a new pull. Then, we will apply Eq. (4a) to the probability as follows:

$$B_4 \leqslant \mathbb{P}\left(\exists s \in [1, T_c - A + 1), \mathsf{UCB}_{s,1} \leqslant \mu_1\right)$$

$$\leqslant \mathbb{P}\left(\exists s \in [1, T_c - A + 1), \bar{r}_{s,1} \leqslant \mu_1, s\mathsf{KL}(\bar{r}_{s,1}, \mu_1) \geqslant l\right)$$

$$\leqslant \frac{T_c - A}{\exp l},$$

where the second inequality is because under event $\{\mathsf{UCB}_{s,1} \leqslant \mu_1\}$ and according to the definition of $\mathsf{UCB}_{s,1}$, we have $\bar{r}_{s,1} \leqslant \mathsf{UCB}_{s,1} \leqslant \mu_1$, and since $s\mathsf{KL}(\bar{r}_{s,1}, \mathsf{UCB}_{s,1}) = l$ with $\mu_1 > \mathsf{UCB}_{s,1}$, we have $s\mathsf{KL}(\bar{r}_{s,1}, \mu_1) \geqslant l$. The last inequality uses Lemma. 6. Similar to the procedure of bounding $I_1$, notice that $\exp(l) \geqslant T\log^4 T$, we have:

$$B_4 \leqslant \frac{4A\left(\log T + 4\sqrt{2\log T}\right)}{\mathsf{KL}_{\mathrm{lb}}} \frac{1}{T\log^4 T} \leqslant \frac{20A}{\mathsf{KL}_{\mathrm{lb}} T\log^3 T}.$$

On the other hand, let $\gamma = \left\lceil \frac{4l}{\mathsf{KL}_{\mathrm{lb}}} \right\rceil$, and we also change the count from time step $t$ to the number of pulls for arm $a$. Then, $B_3$ can be expressed as follows:

$$B_3 = \mathbb{P}\left(\exists t \in [A+1, T_{\mathrm{c}} - 1], N_{t-1}(a) = \gamma, \mu_1 \leqslant \mathsf{UCB}_{N_{t-1}(a),a}\right) = \mathbb{P}\left(\mu_1 \leqslant \mathsf{UCB}_{\gamma,a}\right).$$

For any pair of means $x, y \in [0,1]$, define $\mathsf{KL}^+(x,y) = \mathsf{KL}(x,y)\mathbb{1}_{x<y}$. Then, we have:

$$\mathbb{P}\left(\mu_1 \leqslant \mathsf{UCB}_{\gamma,a}\right) \leqslant \mathbb{P}\left(\mathsf{KL}^+(\bar{r}_{\gamma,a}, \mu_1) \leqslant \frac{l}{\gamma}\right).$$

Notice that $\gamma = \left\lceil \frac{4l}{\mathsf{KL}_{\mathrm{lb}}} \right\rceil \geqslant \frac{4l}{\mathsf{KL}(\mu_a,\mu_1)}$, we can also bound $B_3$ as follows:

$$B_3 \leqslant \mathbb{P}\left(\mathsf{KL}^+(\bar{r}_{\gamma,a}, \mu_1) \leqslant \frac{\mathsf{KL}(\mu_a,\mu_1)}{4}\right).$$

Recall that $\mu_a' \in (\mu_a, \mu_1)$ such that $\mathsf{KL}(\mu_a',\mu_1) = \frac{\mathsf{KL}(\mu_a,\mu_1)}{4}$, so we have $\bar{r}_{\gamma,a} \geqslant \mu_a'$ under the event that $\left\{\mathsf{KL}^+(\bar{r}_{\gamma,a}, \mu_1) \leqslant \frac{\mathsf{KL}(\mu_a,\mu_1)}{4}\right\}$, and thus $\mathsf{KL}(\bar{r}_{\gamma,a}, \mu_a) \geqslant \mathsf{KL}(\mu_a', \mu_a)$. By Lemma. 5, we have:

$$\mathbb{P}\left(\mathsf{KL}^+(\bar{r}_{\gamma,a}, \mu_1) \leqslant \frac{\mathsf{KL}(\mu_a,\mu_1)}{4}\right) \leqslant \mathbb{P}\left(\mathsf{KL}(\bar{r}_{\gamma,a}, \mu_a) \geqslant \mathsf{KL}(\mu_a', \mu_a)\right) \leqslant \exp(-\gamma \mathsf{KL}(\mu_a', \mu_a)).$$

Notice that $\gamma \mathsf{KL}(r(\gamma), \mu_a) \geqslant \frac{4l\mathsf{KL}(\mu_a', \mu_a)}{\mathsf{KL}_{\mathrm{lb}}} \geqslant l$ by the definition of $\mathsf{KL}_{\min}$, we then have:

$$B_3 \leqslant \exp(-l) \leqslant \frac{1}{T\log^4 T}.$$

The last inequality is due to the fact that $\exp(l) \geqslant T\log^4 T$. So combine $B_3$ and $B_4$ together, we can bound $B_2$ as follows:

$$B_2 \leqslant \sum_{a:\Delta_a > 0} (B_3 + B_4) \leqslant A\left(\frac{20A}{\mathsf{KL}_{\mathrm{lb}} T\log^3 T} + \frac{1}{T\log^4 T}\right) = o\left(\frac{1}{T}\right).$$

Next, we attempt to bound $B_1$. Notice that $B_1$ can also be bounded as follows:

$$B_1 = \mathbb{P}\left(\mathsf{LCB}_{T_{\mathrm{c}}}(a) \geqslant \mathsf{LCB}_{T_{\mathrm{c}}}(1), N_{T_{\mathrm{c}}}(1) > \frac{4l}{\mathsf{KL}_{\mathrm{lb}}}\right)$$

$$\leqslant \underbrace{\mathbb{P}\left(\mu_a \geqslant \mathsf{LCB}_{T_{\mathrm{c}}}(1), N_{T_{\mathrm{c}}}(1) > \frac{4l}{\mathsf{KL}_{\mathrm{lb}}}\right)}_{B_5} + \underbrace{\mathbb{P}\left(\mu_a \leqslant \mathsf{LCB}_{T_{\mathrm{c}}}(a)\right)}_{B_6}.$$

The term $B_6$ can be expressed by counting the number of pulls of arm $a$ as follows:

$$\begin{aligned}
B_6 &\leqslant \mathbb{P}\left(\exists s \in [1, T_{\mathrm{c}} - A + 1], \mathsf{LCB}_{s,a} \geqslant \mu_a\right) \\
&= \mathbb{P}\left(\exists s \in [1, T_{\mathrm{c}} - A + 1], \bar{r}_{s,a} \geqslant \mu_a, s\mathsf{KL}(\bar{r}_{s,a}, \mu_a) \geqslant l\right) \\
&\leqslant \frac{T_{\mathrm{c}} - A + 1}{\exp(l)},
\end{aligned}$$

where in the last inequality, we can apply Lemma. (6). Since $\log T \geqslant \sqrt{2\log T}$ and $\exp(l) \geqslant \frac{1}{T\log^4 T}$ when $T \geqslant 16$, we have:

$$B_6 \leqslant \frac{4A\left(\log T + 4\sqrt{2\log T}\right)}{\mathsf{KL}_{\mathrm{lb}} T\log^4 T} + \frac{1}{T\log^4 T} = o\left(\frac{1}{T}\right).$$

For any pair of means $x, y \in [0, 1]$, define $\mathsf{KL}^-(x, y) = \mathsf{KL}(x, y) \mathbb{1}_{x>y}$. Similarly, term $B_5$ can be expressed as:

$$B_5 \leqslant \mathbb{P}\left(\exists s > \frac{4l}{\mathsf{KL}_{\mathrm{lb}}}, \mu_a \geqslant \mathsf{LCB}_{s,1}\right)$$

$$\leqslant \mathbb{P}\left(\exists s > \frac{4l}{\mathsf{KL}_{\mathrm{lb}}}, s\mathsf{KL}^-\left(\bar{r}_{s,1}, \mu_a\right) \leqslant l\right)$$

$$\leqslant \mathbb{P}\left(\exists s > \frac{4l}{\mathsf{KL}_{\mathrm{lb}}}, \mathsf{KL}^-\left(\bar{r}_{s,1}, \mu_a\right) \leqslant \frac{\mathsf{KL}(\mu_a, \mu_1)}{4}\right).$$

Recall that $\mu'_a \in (\mu_a, \mu_1)$ satisfy $\mathsf{KL}\left(\mu'_a, \mu_a\right) = \frac{\mathsf{KL}(\mu_a, \mu_1)}{4}$, so under event $\{\mathsf{KL}^-\left(\bar{r}_{s,1}, \mu_a\right) \leqslant \frac{\mathsf{KL}(\mu_a, \mu_1)}{4}\}$ we can conclude $\bar{r}_{s,1} \leqslant \mu'_a$, which means $\mathsf{KL}\left(\bar{r}_{s,1}, \mu_1\right) \geqslant \mathsf{KL}\left(\mu'_a, \mu_1\right)$, so we have:

$$\mathbb{P}\left(\exists s > \frac{4l}{\mathsf{KL}_{\mathrm{lb}}}, \mathsf{KL}^-\left(\bar{r}_{s,1}, \mu_a\right) \leqslant \frac{\mathsf{KL}(\mu_a, \mu_1)}{4}\right)$$

$$\leqslant \mathbb{P}\left(\exists s > \frac{4l}{\mathsf{KL}_{\mathrm{lb}}}, \bar{r}_{s,1} \leqslant \mu_1, \mathsf{KL}\left(\bar{r}_{s,1}, \mu_1\right) \geqslant \mathsf{KL}\left(\mu'_a, \mu_1\right)\right)$$

$$\leqslant \mathbb{P}\left(\exists s > \frac{4l}{\mathsf{KL}_{\mathrm{lb}}}, \bar{r}_{s,1} \leqslant \mu_1, s\mathsf{KL}\left(\bar{r}_{s,1}, \mu_1\right) \geqslant \frac{4l\mathsf{KL}\left(\mu'_a, \mu_1\right)}{\mathsf{KL}_{\mathrm{lb}}}\right)$$

$$\leqslant \frac{el \log T + e}{\exp\left(\frac{4l\mathsf{KL}(\mu'_a, \mu_1)}{\mathsf{KL}_{\mathrm{lb}}}\right)},$$

where the last inequality comes from Lemma. 6. By the definition of $\mathsf{KL}_{\min}$, we know that $\frac{4l\mathsf{KL}\left(\mu'_a, \mu_1\right)}{\mathsf{KL}_{\mathrm{lb}}} \geqslant l$, so we have:

$$B_5 \leqslant \frac{el \log T + e}{\exp(l)} \leqslant \frac{10e \log^2 T}{T \log^4 T} = o\left(\frac{1}{T}\right).$$

Therefore, collecting the bounds for $B_5$ and $B_6$ and $B_2$, we have a bound for $I_2$ as follows:

$$I_2 \leqslant B_1 + B_2 \leqslant B_5 + B_6 + B_2 = o\left(\frac{1}{T}\right).$$

Finally, putting the bounds on $I_1$ and $I_2$ together, we have:

$$\mathbb{E}[N_a(T)] \leqslant I_1 + TI_2 \leqslant \frac{\log T}{\mathsf{KL}(\mu_a, \mu_1)} + \frac{10 \log^{\frac{3}{4}} T}{\mathsf{KL}(\mu_a, \mu_1)} + o(1).$$

Therefore, by the regret decomposition lemma, we can bound the total regret as follows:

$$\mathsf{Reg}_{\boldsymbol{\mu}}^{\mathsf{KL\text{-}EOCP}}(T) \leqslant \sum_{a:\Delta_a>0}\left(\frac{\Delta_a \log T}{\mathsf{KL}(\mu_a, \mu_1)} + \frac{10\Delta_a \log^{\frac{3}{4}} T}{\mathsf{KL}(\mu_a, \mu_1)}\right) + o(1).$$

# D  Proof of Concentration Inequalities

In this section, we provide the proof of the concentration inequalities presented in previous sections.

## D.1  Proof of Lemma. 3

The proof of Lemma. 3 relies on the maximal inequalities in Lemma. 1 and Lemma. 2, and the famous Hoeffding's inequality which we state here for the sake of completeness.

**Lemma 8 (Hoeffding's Inequality)** *Let $(W_i)_{i=1}^T$ be i.i.d. $\sigma$-sub-Gaussian random variables with $\mathbb{E}[W_1] = 0$, we have:*

$$\mathbb{P}\left(\frac{\sum_{i=1}^T W_i}{T} \geqslant \delta\right) \leqslant \exp\left(-\frac{T\delta^2}{2\sigma^2}\right).$$

**Eq. (4a) of Lemma. 3:** We now prove Eq. (4a) of Lemma. 3. We will first base our proof on the anytime concentration on union bound, which is tight only when $T_2 - T_1$ is relatively small. This procedure will result in the first term in the minimum at the RHS. To prove a stronger result when $T_2 - T_1$ is relatively large, we resort to the technique of "peeling device" which divide the time horizon into exponential grids, where we will perform maximal inequality inside each grid and a union bound over the grids. This will result in the second term in the minimum at the RHS. We first perform union bound on $s$ as follows:

$$\mathbb{P}\left(\exists s \in (T_1, T_2], \frac{\sum_{i=1}^{s} W_i}{s} + \sqrt{\frac{2l}{s}} \leqslant 0\right) = \mathbb{P}\left(\exists s \in (T_1, T_2], \sum_{i=1}^{s} W_i + \sqrt{2ls} \leqslant 0\right)$$

$$\leqslant \sum_{j=T_1+1}^{T_2} \mathbb{P}\left(s = j, \sum_{i=1}^{s} W_i + \sqrt{2ls} \leqslant 0\right).$$

Then, we can bound each probability with Hoeffding's concentration inequality as follows:

$$\mathbb{P}\left(s = j, \sum_{i=1}^{s} W_i + \sqrt{2ls} \leqslant 0\right) = \mathbb{P}\left(\sum_{i=1}^{j} W_i + \sqrt{2jl} \leqslant 0\right) \leqslant \exp\left(-\frac{(\sqrt{2jl})^2}{2j}\right) = \frac{1}{\exp(l)}.$$

Therefore, summing up the probabilities, we will have:

$$\mathbb{P}\left(\exists s \in (T_1, T_2], \frac{\sum_{i=1}^{s} W_i}{s} + \sqrt{\frac{2l}{s}} \leqslant 0\right) \leqslant \sum_{j=T_1+1}^{T_2} \frac{1}{\exp(l)} = \frac{T_2 - T_1}{\exp(l)}. \tag{8}$$

Next, we apply the peeling method to prove the second inequality. Take $\beta > 1$ to be a constant. Let $M = \lfloor \log_\beta \frac{T_2}{T_1} \rfloor$, we apply peeling method on $s$ and divide the time horizon over exponential grids $[T_1, T_1\beta], [T_1\beta, T_1\beta^2], \cdots, [T_1\beta^M, T_2]$ as follows:

$$\mathbb{P}\left(\exists s \in (T_1, T_2], \frac{\sum_{i=1}^{s} W_i}{s} + \sqrt{\frac{2l}{s}} \leqslant 0\right) = \mathbb{P}\left(\exists s \in (T_1, T_2], \sum_{i=1}^{s} W_i + \sqrt{2ls} \leqslant 0\right)$$

$$\leqslant \sum_{j=0}^{M} \mathbb{P}\left(\exists s \in [T_1\beta^j, T_1\beta^{j+1}], \sum_{i=1}^{s} W_i + \sqrt{2ls} \leqslant 0\right).$$

Since at each grid, $s \geqslant T_1\beta^j$, so we can upper bound each probability as:

$$\mathbb{P}\left(\exists s \in [T_1\beta^j, T_1\beta^{j+1}], \sum_{i=1}^{s} W_i + \sqrt{2ls} \leqslant 0\right) \leqslant \mathbb{P}\left(\exists s \in [T_1\beta^j, T_1\beta^{j+1}], \sum_{i=1}^{s} W_i + \sqrt{2T_1\beta^j l} \leqslant 0\right).$$

Let $\beta = \frac{l}{l-1}$, then according to the anytime concentration inequality from Lemma. 1, we have:

$$\mathbb{P}\left(\exists s \in [T_1\beta^j, T_1\beta^{j+1}], \sum_{i=1}^{s} W_i + \sqrt{2T_1\beta^j l} \leqslant 0\right) \leqslant \exp\left(-\frac{2T_1\beta^j l}{2T_1\beta^{j+1}}\right) = \exp\left(-\frac{l}{\beta}\right) = \frac{e}{\exp(l)}.$$

Then, summing up the probabilities, we can bound the total probability as follows:

$$\mathbb{P}\left(\exists s \in (T_1, T_2], \frac{\sum_{i=1}^{s} W_i}{s} + \sqrt{\frac{2l}{s}} \leqslant 0\right) \leqslant \frac{e(M+1)}{\exp(l)} \leqslant \frac{e(\log T_2 - \log T_1)}{\log \beta \exp(l)} + \frac{e}{\exp l},$$

where the last inequality is due to the definition of $M = \lfloor \log_\beta \frac{T_2}{T_1} \rfloor$. Notice that when $l \geqslant 2$, $\log \beta = \log\left(\frac{l}{l-1}\right) \geqslant \frac{1}{l}$, so we can further upper bound the probability as:

$$\mathbb{P}\left(\exists s \in (T_1, T_2], \frac{\sum_{i=1}^{s} W_i}{s} + \sqrt{\frac{2l}{s}} \leqslant 0\right) \leqslant \frac{el(\log T_2 - \log T_1)}{\exp(l)} + \frac{e}{\exp l}. \tag{9}$$

Finally, combining the two bounds Eq. (8) and Eq. (9) together, we can prove the first result of Lemma. 3 as follows:

$$\mathbb{P}\left(\exists s \in (T_1, T_2], \frac{\sum_{i=1}^s W_i}{s} + \sqrt{\frac{2l}{s}} \leqslant 0\right) \leqslant \min\left\{\frac{T_2 - T_1}{\exp(l)}, \frac{el\,(\log T_2 - \log T_1)}{\exp(l)} + \frac{e}{\exp l}\right\}.$$

**Eq. (4b) of Lemma. 3:** Next, we prove Eq. (4b) of Lemma. 3. Our result is only based on performing union bound, but one can also modify the proof of Eq. (4a) with peeling trick to prove Eq. (4b). However, the result from peeling trick is no better than simply performing union bound, at least not order-wise better. So for simplicity, we apply union bound to the probability as follows:

$$\mathbb{P}\left(\exists s \in [T_1, T_2], \frac{\sum_{i=1}^s W_i}{s} + \sqrt{\frac{2l}{s}} + \delta \leqslant 0\right) = \mathbb{P}\left(\exists s \in [T_1, T_2], \sum_{i=1}^s W_i + \sqrt{2ls} + \delta s \leqslant 0\right)$$

$$\leqslant \sum_{j=T_1}^{T_2} \mathbb{P}\left(s = j, \sum_{i=1}^s W_i + \sqrt{2ls} + \delta s \leqslant 0\right)$$

$$= \sum_{j=T_1}^{T_2} \mathbb{P}\left(\sum_{i=1}^j W_i + \sqrt{2jl} + \delta j \leqslant 0\right).$$

Then, we can apply Hoeffding's inequality to upper bound each probability as follows:

$$\mathbb{P}\left(\sum_{i=1}^j W_i + \sqrt{2jl} + \delta j \leqslant 0\right) \leqslant \exp\left(-\frac{\left(\sqrt{2jl} + \delta j\right)^2}{2j}\right)$$

$$= \exp\left(-\frac{2jl + 2\sqrt{2jl}\delta j + \delta^2 j^2}{2j}\right)$$

$$\leqslant \frac{\exp(-\frac{\delta^2}{2}j)}{\exp\left(l + \delta\sqrt{2T_1 l}\right)},$$

where the last inequality is due to $j \geqslant T_1$. So summing up all the probabilities, we can bound the anytime concentration as:

$$\mathbb{P}\left(\exists s \in [T_1, T_2], \frac{\sum_{i=1}^s W_i}{s} + \sqrt{\frac{2l}{s}} + \delta \leqslant 0\right) \leqslant \sum_{j=T_1}^{T_2} \frac{\exp(-\frac{\delta^2}{2}j)}{\exp\left(l + \delta\sqrt{2T_1 l}\right)}$$

$$\leqslant \frac{1}{\exp\left(l + \delta\sqrt{2T_1 l}\right)} \frac{\exp(-\frac{\delta^2 T_1}{2})}{1 - \exp\left(-\frac{\delta^2}{2}\right)}$$

$$\leqslant \frac{4}{\delta^2 \exp\left(\left(\sqrt{l} + \delta\sqrt{\frac{T_1}{2}}\right)^2\right)}.$$

where the last step is due to $1 - e^{-\frac{\delta^2}{2}} \geqslant \frac{\delta^2}{2}$ when $\delta \in [0, \sqrt{3}]$. Then we finish the proof of Eq. (4b) of Lemma. 3.

**Eq. (4c) of Lemma. 3:** Finally, we prove Eq. (4a) of Lemma. 3 which bounds the summation of probability for deviation events over the time horizon. Since $l \geqslant \frac{T_1 \delta^2}{2}$, we define $\gamma = \frac{2l}{\delta^2} \geqslant T_1$. Then, we can bound the probabilities when $s \leqslant \gamma$ by 1 as follows:

$$\sum_{s=T_1+1}^{T_2} \mathbb{P}\left(\frac{\sum_{i=1}^s W_i}{s} + \sqrt{\frac{2l}{s}} \geqslant \delta\right) \leqslant \gamma - T_1 + \sum_{s=\lceil\gamma\rceil}^{T_2} \mathbb{P}\left(\frac{\sum_{i=1}^s W_i}{s} + \sqrt{\frac{2l}{s}} \geqslant \delta\right).$$

When $s \geqslant \lceil\gamma\rceil$, we have: $\sqrt{\frac{2l}{s}} = \delta\sqrt{\frac{\gamma}{s}}$. So, for each probability inside the summation, we have:

$$\mathbb{P}\left(\frac{\sum_{i=1}^s W_i}{s} + \sqrt{\frac{2l}{s}} \geqslant \delta\right) = \mathbb{P}\left(\frac{\sum_{i=1}^s W_i}{s} \geqslant \delta - \sqrt{\frac{2l}{s}}\right) \leqslant \mathbb{P}\left(\frac{\sum_{i=1}^s W_i}{s} \geqslant \delta\left(1 - \sqrt{\frac{\gamma}{s}}\right)\right).$$

Then, we can bound the probability with Hoeffding's inequality as follows:

$$\mathbb{P}\left(\frac{\sum_{i=1}^{s} W_i}{s} \geqslant \delta\left(1 - \sqrt{\frac{\gamma}{s}}\right)\right) \leqslant \exp\left(-\frac{s\delta^2}{2}\left(1 - \sqrt{\frac{\gamma}{s}}\right)^2\right) = \exp\left(-\frac{\delta^2}{2}\left(\sqrt{s} - \sqrt{\gamma}\right)^2\right).$$

Notice that the upper bound function $\exp(-\frac{\delta^2}{2}(\sqrt{s} - \sqrt{\gamma})^2)$ on the RHS is uni-modal when $s \geqslant \gamma$. If a function $f(s)$ is uni-modal, then we can bound the summation $\sum_{s=\gamma}^{\infty}$ with the sum of $\max_s f(s)$ and integral $\int_{\gamma}^{\infty} f(s)ds$. Therefore, putting the upper bound back to the summation, we have:

$$\sum_{s=\lceil\gamma\rceil}^{T_2} \mathbb{P}\left(\frac{\sum_{i=1}^{s} W_i}{s} + \sqrt{\frac{2l}{s}} \geqslant \delta\right) \leqslant \sum_{s=\lceil\gamma\rceil}^{T_2} \exp\left(-\frac{\delta^2}{2}\left(\sqrt{s} - \sqrt{\gamma}\right)^2\right)$$

$$\leqslant \sum_{s=\lceil\gamma\rceil}^{\infty} \exp\left(-\frac{\delta^2}{2}\left(\sqrt{s} - \sqrt{\gamma}\right)^2\right)$$

$$\leqslant 1 + \int_{\gamma}^{\infty} \exp\left(-\frac{\delta^2}{2}\left(\sqrt{s} - \sqrt{\gamma}\right)^2\right) ds$$

$$= 1 + \frac{2}{\delta^2} + \frac{\sqrt{2\pi\gamma}}{\delta}.$$

So the whole term can be bounded by:

$$\sum_{s=T_1+1}^{T_2} \mathbb{P}\left(\frac{\sum_{i=1}^{s} W_i}{s} + \sqrt{\frac{2l}{s}} \geqslant \delta\right) \leqslant \gamma - T_1 + 1 + \frac{2}{\delta^2} + \frac{\sqrt{2\pi\gamma}}{\delta} = \frac{2l + \sqrt{4\pi l} + 2}{\delta^2} + 1 - T_1,$$

which completes the proof of Lemma. 3.

### D.2 Proof of Lemma. 5

We only prove the inequality when $\bar{\mu}_s \leqslant \mu$. The other inequality can be proved exactly the same way. For every $\lambda \in \mathbb{R}$, let $\phi_\mu(\lambda) = \log \mathbb{E}[\exp(\lambda X_1)]$ which is well-defined and finite by assumption. Let $W_0^\lambda = 1$ and for $s \geqslant 1$, we define $W_t^\lambda = \exp(\lambda S_s - s\phi_\mu(\lambda))$. We show that $\left(W_s^\lambda\right)_{s \geqslant 1}$ is a martingale with respect to the $\sigma$-field $\mathcal{F}_s = \sigma(W_1, \cdots, W_s)$. In fact,

$$\mathbb{E}\left[W_{s+1}^\lambda | \mathcal{F}_s\right] = \mathbb{E}\left[\exp\left(\lambda S_s + \lambda W_{s+1} - s\phi_\mu(\lambda) - \phi_\mu(\lambda)\right) | \mathcal{F}_s\right]$$

$$= \exp\left(\lambda S_s - s\phi_\mu(\lambda)\right) \mathbb{E}\left[\exp\left(\lambda W_{s+1} - \phi_\mu(\lambda)\right) | \mathcal{F}_s\right]$$

$$= W_s^\lambda \frac{\mathbb{E}\left[\exp\left(\lambda W_{s+1}\right)\right]}{\mathbb{E}\exp\left[\lambda X_1\right]}$$

$$= W_s^\lambda,$$

where the second equality is because $S_s$ is measurable w.r.t. $\mathcal{F}_s$, and the third equality uses the fact that $W_{s+1}$ is independent w.r.t. $\mathcal{F}_s$. Th last equality is due to the i.i.d. nature of $W_1$ and $W_{s+1}$. Let $x \in [0, \mu]$ be such that $\mathsf{KL}(x, \mu) = \delta/s$, and let $\lambda(x) = \theta^x - \theta^\mu$. It is worth-noting that since $\theta^\mu$ is a monotonically non-decreasing function since its inverse function $\mu = b'(\theta)$ is monotonically non-decreasing. So we have $\lambda(x) \leqslant 0$ since $x \leqslant \mu$. Observe that:

$$\bar{\mu}_s \leqslant \mu, \quad \mathsf{KL}\left(\bar{\mu}_s, \mu\right) \geqslant \frac{\delta}{s}, \quad \text{and,} \quad x \leqslant \mu, \quad \mathsf{KL}\left(x, \mu\right) = \frac{\delta}{s}.$$

Then it holds that $x \geqslant \bar{\mu}_s$. Notice that for natural parameter exponential family, $\phi_\mu(\lambda) = b(\lambda + \theta^\mu) - b(\theta^\mu)$. Hence on the event $\{\bar{\mu}_s < \mu\} \cap \{s\mathsf{KL}\left(\bar{\mu}_s, \mu\right) \geqslant \delta\}$, we have:

$$\lambda(x)\bar{\mu}_s - \phi_\mu(\lambda(x)) \geqslant \lambda(x)x - \phi_\mu(\lambda(x)) = x\left(\theta^x - \theta^\mu\right) - b(\theta^x) + b(\theta^\mu) = \mathsf{KL}(x, \mu) = \frac{\delta}{s},$$

where the first inequality is because $\lambda(x) < 0$, and the second last equality uses the expression of $\mathsf{KL}$ divergence for natural exponential families. Putting everything together, we have:

$$\mathbb{P}\left(\bar{\mu}_s \leqslant \mu, s\mathsf{KL}\left(\bar{\mu}_s, \mu\right) \geqslant \delta\right) \leqslant \mathbb{P}\left(\lambda(x)\bar{\mu}_s - \phi_\mu(\lambda(x)) \geqslant \frac{\delta}{s}\right)$$

$$= \mathbb{P}\left(W_s^{\lambda(x)} \geqslant \delta\right)$$

$$\leqslant \mathbb{E}[W_s^{\lambda(x)}] \exp(-\delta),$$

where the last inequality uses Markov inequality. Since $W_s^{\lambda(x)}$ is a martingale, so we have:

$$\mathbb{P}\left(\bar{\mu}_s \leqslant \mu, s\mathsf{KL}\left(\bar{\mu}_s, \mu\right) \geqslant \delta\right) \leqslant \mathbb{E}[W_0^{\lambda(x)}]\exp(-\delta) = \exp(-\delta).$$

## D.3   Proof of Lemma 6

We only prove the inequality when $\bar{\mu}_s \leqslant \mu$. The other inequality can be proved exactly the same way. The proof of Lemma 6 partly resembles the Proof of Lemma 5. However, to show an any-time concentration bound, we need either the union bound or a peeling trick. We first apply union bound as follows:

$$\mathbb{P}\left(\exists s \in (T_1, T_2], \bar{\mu}_s \leqslant \mu, s\mathsf{KL}\left(\bar{\mu}_s, \mu\right) \geqslant l\right) \leqslant \sum_{s=T_1+1}^{T_2} \mathbb{P}\left(\bar{\mu}_s \leqslant \mu, s\mathsf{KL}\left(\bar{\mu}_s, \mu\right) \geqslant l\right)$$
$$\leqslant (T_2 - T_1)\exp(-l),$$

which obtains the first term in the RHS of Lemma 6. Next, we apply the peeling trick. For every $\lambda \in \mathbb{R}$, let $\phi_\mu(\lambda) = \log \mathbb{E}[\exp(\lambda X_1)]$ which is well-defined and finite by assumption. Let $W_0^\lambda = 1$ and for $s \geqslant 1$, we define $W_t^\lambda = \exp(\lambda S_s - s\phi_\mu(\lambda))$. Recall that $\left(W_s^\lambda\right)_{s \geqslant 1}$ is a martingale with respect to the $\sigma$-field $\mathcal{F}_s = \sigma(W_1, \cdots, W_s)$. Take $\beta > 1$ to be a constant. Let $M = \lfloor \log_\beta \frac{T_2}{T_1} \rfloor$, we apply peeling method on $s$ and divide the time horizon over exponential grids $[T_1, T_1\beta], [T_1\beta, T_1\beta^2], \cdots, [T_1\beta^M, T_2]$ as follows:

$$\mathbb{P}\left(\exists s \in (T_1, T_2], \bar{\mu}_s \leqslant \mu, s\mathsf{KL}\left(\bar{\mu}_s, \mu\right) \geqslant l\right) \leqslant \sum_{i=0}^{M} \mathbb{P}\left(\exists s \in [T_1\beta^i, T_1\beta^{i+1}], \bar{\mu}_s \leqslant \mu, s\mathsf{KL}\left(\bar{\mu}_s, \mu\right) \geqslant l\right).$$

Let $s_i = T_1\beta^i$ and let $x \leqslant \mu$ such that $s\mathsf{KL}(x, \mu) = l$. let $\lambda(x) = \theta^x - \theta^\mu < 0$. Then, we have $\mathsf{KL}(x, \mu) = \lambda(x)x - \phi_\mu(\lambda(x))$. Consider $z$ such that $z_i < \mu$ and $\mathsf{KL}(z_i, \mu) = \frac{l}{s_i}$, so we have when $s \in [s_i, s_{i+1}]$

$$\mathsf{KL}(\bar{\mu}_s, \mu) \geqslant \frac{l}{s} \geqslant \frac{l}{s_{i+1}} = \mathsf{KL}(z_{i+1}, \mu).$$

So we can conclude that $\bar{\mu}_s \leqslant z_{i+1}$ Also, we have:

$$\mathsf{KL}(z_{i+1}, \mu) = \frac{l}{s_{i+1}} = \frac{1}{\beta}\frac{l}{s_i} \geqslant \frac{1}{\beta}\frac{l}{s}.$$

Therefore, we have:

$$\lambda(z_{i+1})\bar{\mu}_s - \phi_\mu(\lambda(z_{i+1})) \geqslant \lambda(z_{i+1})z_{i+1} - \phi_\mu(\lambda(z_{i+1})) = \mathsf{KL}(z_{i+1}, \mu) \geqslant \frac{l}{\beta s}.$$

So we can bound each probability as:

$$\mathbb{P}\left(\exists s \in [z_i, z_{i+1}], \bar{\mu}_s \leqslant \mu, s\mathsf{KL}\left(\bar{\mu}_s, \mu\right) \geqslant \delta\right) \leqslant \mathbb{P}\left(\lambda(z_{i+1})\bar{\mu}_s - \phi_\mu(\lambda(z_{i+1})) \geqslant \frac{l}{\beta s}\right)$$
$$= \mathbb{P}\left(W_s^{\lambda(z_{i+1})} \geqslant \exp\left(\frac{l}{\beta}\right)\right)$$
$$\leqslant \mathbb{E}[W_s^{\lambda(z_{i+1})}]\exp\left(-\frac{l}{\beta}\right),$$

where the last inequality uses Markov inequality. Since $W_s^{\lambda(z_{i+1})}$ is a martingale, so we have:

$$\mathbb{P}\left(\exists s \in [z_i, z_{i+1}], \bar{\mu}_s \leqslant \mu, s\mathsf{KL}\left(\bar{\mu}_s, \mu\right) \geqslant \delta\right) \leqslant \mathbb{E}[W_0^{\lambda(z_{i+1})}]\exp\left(-\frac{l}{\beta}\right) = \frac{e}{\exp(l)},$$

where in the last step, we choose $\beta = \frac{l}{l-1}$. Then, summing up the probabilities, we can bound the total probability as follows:

$$\mathbb{P}\left(\exists s \in (T_1, T_2], \bar{\mu}_s \leqslant \mu, s\mathsf{KL}\left(\bar{\mu}_s, \mu\right) \geqslant l\right) \leqslant \sum_{j=0}^{M} \frac{e}{\exp(l)} = \frac{e(M+1)}{\exp(l)} \leqslant \frac{e\left(\log T_2 - \log T_1\right)}{\log \beta \exp(l)} + \frac{e}{\exp l},$$

where the last inequality is due to the definition of $M = \lfloor \log_\beta \frac{T_2}{T_1} \rfloor$. Notice that when $l \geqslant 2$, $\log \beta = \log \left( \frac{l}{l-1} \right) \geqslant \frac{1}{l}$, so we can further upper bound the probability as:

$$\mathbb{P}\left(\exists s \in (T_1, T_2], \bar{\mu}_s \leqslant \mu, s\mathsf{KL}\left(\bar{\mu}_s, \mu\right) \geqslant l\right) \leqslant \frac{el\left(\log T_2 - \log T_1\right)}{\exp(l)} + \frac{e}{\exp l}. \tag{10}$$

### D.4  Proof of Lemma. 7

Our proof is based on the analysis of Theorem. 2 of [12]. For any pair of means $x, y \in [0, 1]$, define $\mathsf{KL}^+(x, y) = \mathsf{KL}(x, y)\mathbb{1}_{x < y}$. Then for a fixed $s$, under event $\{\mathsf{UCB}_s \geqslant \mu'\}$ we have either $\mu' < \bar{\mu}_s$, or $\mu' > \bar{\mu}_s$ but $s\mathsf{KL}(\bar{\mu}_s, \mu') \leqslant l$. So in general we can conclude that $s\mathsf{KL}^+(\bar{\mu}_s, \mu') \leqslant l$. Then we can bound the sum of probabilities as follows:

$$\sum_{s=1}^{T_1} \mathbb{P}\left(\mathsf{UCB}_s \geqslant \mu'\right) \leqslant \sum_{s=1}^{T_1} \mathbb{P}\left(s\mathsf{KL}^+(\bar{\mu}_s, \mu') \leqslant l\right).$$

Define $\gamma = \frac{(1+\varepsilon)l}{\mathsf{KL}^+(\mu, \mu')} = \frac{(1+\varepsilon)l}{\mathsf{KL}(\mu, \mu')}$, then if $T_1 > \gamma$, we can bound the first $\gamma$ terms in the summation with 1. If otherwise $T_1 \leqslant \gamma$, the whole summation is bounded by $\gamma$. So without loss with generality, assume $\gamma < T_1$ and let $\varepsilon > 0$ be a constant, we have:

$$\sum_{s=1}^{T_1} \mathbb{P}\left(s\mathsf{KL}^+(\bar{\mu}_s, \mu') \leqslant l\right) \leqslant \gamma + \sum_{s=\lceil \gamma \rceil}^{T_1} \mathbb{P}\left(s\mathsf{KL}^+(\bar{\mu}_s, \mu') \leqslant l\right)$$

$$\leqslant \gamma + \sum_{s=\lceil \gamma \rceil}^{T_1} \mathbb{P}\left(\gamma\mathsf{KL}^+(\bar{\mu}_s, \mu') \leqslant l\right)$$

$$= \gamma + \sum_{s=\lceil \gamma \rceil}^{T_1} \mathbb{P}\left(\mathsf{KL}^+(\bar{\mu}_s, \mu') \leqslant \frac{\mathsf{KL}(\mu, \mu')}{1 + \varepsilon}\right),$$

where the second inequality is due to $\mathsf{KL}^+(\bar{\mu}_s, \mu') > 0$. For any $s$, let $r(\varepsilon) \in (\mu, \mu')$ such that $\mathsf{KL}(r(\varepsilon), \mu') = \frac{\mathsf{KL}(\mu, \mu')}{1+\varepsilon}$. if $\mathsf{KL}^+(\bar{\mu}_s, \mu') \leqslant \frac{\mathsf{KL}(\mu, \mu')}{1+\varepsilon}$, we have $\bar{\mu}_s \geqslant r(\varepsilon)$. Hence,

$$\mathbb{P}\left(\mathsf{KL}^+(\bar{\mu}_s, \mu') \leqslant \frac{\mathsf{KL}(\mu, \mu')}{1 + \varepsilon}\right) \leqslant \mathbb{P}\left(\bar{\mu}_s \geqslant \mu, \mathsf{KL}(\bar{\mu}_s, \mu) \geqslant \mathsf{KL}(r(\varepsilon), \mu)\right) \leqslant \exp(-s\mathsf{KL}(r(\varepsilon), \mu)),$$

where the last inequality uses Lemma. 5. So we can bound the sum of probabilities as follows:

$$\sum_{s=1}^{T_1} \mathbb{P}\left(s\mathsf{KL}^+(\bar{\mu}_s, \mu') \leqslant l\right) \leqslant \gamma + \sum_{s=\lceil \gamma \rceil}^{\infty} \exp(-s\mathsf{KL}(r(\varepsilon), \mu)) \leqslant \gamma + \frac{\exp(-\gamma\mathsf{KL}(r(\varepsilon), \mu))}{1 - \exp(-\mathsf{KL}(r(\varepsilon), \mu))}.$$

Notice that $\exp(-\gamma\mathsf{KL}(r(\varepsilon), \mu)) = \exp\left(-l\frac{(1+\varepsilon)\mathsf{KL}(r(\varepsilon), \mu)}{\mathsf{KL}(\mu, \mu')}\right) \leqslant T^{-\beta_1(\varepsilon)}$, where $\beta_1(\varepsilon) = \frac{(1+\varepsilon)\mathsf{KL}(r(\varepsilon), \mu)}{\mathsf{KL}(\mu, \mu')}$. Let $\beta_2(\varepsilon) = \frac{1}{1 - \exp(-\mathsf{KL}(r(\varepsilon), \mu))}$. It is easy to check that $r(\varepsilon) = \mu + \mathcal{O}(\varepsilon)$, so we have $\beta_1(\varepsilon) = \mathcal{O}(\varepsilon^2)$ and $\beta_2(\varepsilon) = \mathcal{O}(\varepsilon^{-2})$. So we have:

$$\sum_{s=1}^{T_1} \mathbb{P}\left(\mathsf{UCB}_s \geqslant \mu'\right) \leqslant \sum_{s=1}^{T_1} \mathbb{P}\left(s\mathsf{KL}^+(\bar{\mu}_s, \mu') \leqslant l\right) \leqslant \frac{(1+\varepsilon)l}{\mathsf{KL}(\mu, \mu')} + \frac{\beta_2(\varepsilon)}{T^{\beta_1(\varepsilon)}}.$$

**Remark:** Let $\varepsilon = l^{-\frac{1}{4}}$, we have:

$$\sum_{s=1}^{T_1} \mathbb{P}\left(\mathsf{UCB}_s \geqslant \mu'\right) \leqslant \frac{(1+\varepsilon)l}{\mathsf{KL}(\mu, \mu')} + \frac{\beta_2(\varepsilon)}{T^{\beta_1(\varepsilon)}} \leqslant \frac{l + l^{\frac{3}{4}}}{\mathsf{KL}(\mu, \mu')} + \mathcal{O}\left(\frac{\sqrt{l}}{\exp\left(\sqrt{l}\right)}\right).$$

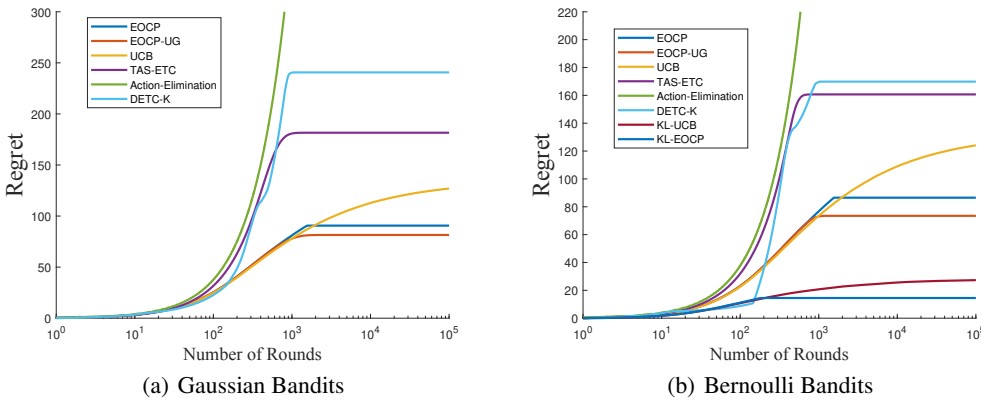

|   (a) Gaussian Bandits   |   (b) Bernoulli Bandits   |

Figure 2: Regret performance in bandit models with 4 arms.

# E    Numerical Experiments Beyond 2-Armed Bandits

In this section, we present numerical experiments based on bandits with more than two arms. We consider a four armed bandit model with expected rewards $[0.7, 0.2, 0.2, 0.2]$. We study the cumulative regret performance of all our proposed algorithms, i.e., EOCP in Algorithm. 1, EOCP-UG in Algorithm. 2, and KL-EOCP in Algorithm. 3, with both Gaussian and Bernoulli rewards. Since some baseline algorithms we chose in Section. 6 (e.g., DETC and BAI-ETC) does not directly generalize to the multi-arm model, we replace them with DETC-K [23] and TAS-ETC respectively, where the TAS-ETC algorithm first uses the famous Track and Stop [36] algorithm to identify the best arm and commits to it for the rest of horizon. Additional to all other baselines we used in Sec. 6, we also compare our algorithms to the Action-Elimination algorithm [42], which provably has both $\mathcal{O}(\log T)$ regret and $\mathcal{O}(\log T)$ commitment time. We choose the reward of all sub-optimal arms to be the same value so that BAI based algorithms can utilize a closed form expression to solve the optimal arm pull fraction $w^*$ and does not need to solve a non-convex minimax problem. We choose $\Delta_{\mathrm{lb}}$ to be $0.5$ for EOCP and KL-EOCP, and the results are shown in Fig. 2 and averaged over $10^4$ iterations.

In both Gaussian and Bernoulli bandits, we observe very similar trends compared to Fig. 1 in two-armed models. the TAS-ETC algorithm and DETC-K algorithm incurs high regret over the total time horizon due to their aggressive exploration at the beginning of the trials, similar to the performance of BAI-ETC and DETC in 2-armed models. The Action Elimination algorithm incurs even larger regret because of its almost uniform sampling rule at the beginning of the trials. It selects sub-optimal actions even more often than BAI based algorithms and could not identify the optimal action as quickly. On the other hand, our proposed algorithms take a more delicate exploration strategy and result in lower regret, which indicates that the EOCP algorithm and its variants generalize well beyond two-armed bandit models. The over-exploration phenomenon is also witnessed in this experiment the UCB algorithm and the KL-UCB algorithm continues to explore sub-optimal actions even after the EOCP algorithms have identified the best action. The regret of UCB algorithms continues to increase during the whole time horizon, which shows that over-exploration is a common and fundamental issue in optimistic algorithms. Preventing over-exploration will significantly boost the performance of algorithms not only in bandits but also in other online decision-making problems.

