# OpenReview forum: "Fast and Regret Optimal Best Arm Identification: Fundamental Limits and Low-Complexity Algorithms"
_NeurIPS.cc/2023/Conference — NeurIPS 2023 poster_

### Official Review · Reviewer_SqBM · 2023-06-21

**Soundness:** 3 good
**Presentation:** 4 excellent
**Contribution:** 3 good
**Rating:** 7
**Confidence:** 3

**Summary:**

This work focuses on simultaneously achieving regret minimization and best arm identification in multi-armed bandits, which is called Regret-Optimal Best Arm Identification (ROBAI). That is, the goal is to identify with high probability and as fast as possible / within a certain time frame the optimal arm, and play until round T. In practice, authors propose three algorithmic contributions, which have guarantees on the cumulative regret, the time before commitment to exploiting a single arm (called stopping time in the paper) and the error on the identification of the optimal arm. The first two algorithms are asymptotically optimal regretwise for Gaussian bandits, and respectively tackle the pre-determined stopping time case and the adaptive stopping time case. They consist in exploring arms using the UCB principle, and committing to the arm which maximizes the LCB. The last one is a variant of the previous algorithms which achieves asymptotical regret optimality for subgaussian distributions, by leveraging KL confidence intervals.

**Strengths:**

- Originality: This paper tackles an interesting problem which has been seldom considered. Although the algorithms reuse well-known principles (UCB / LCB), their analysis allow to prove very good results (regret optimality, interesting bound for adaptive “stopping” time). Table 1 is clear and shows that the work is well-grounded in prior literature.
- Quality: The results seem technically sound, although I did not check the appendix in detail.
- Clarity: The submission and the proof sketches are clearly written.
- Significance: This paper provides and substantiates an interesting insight on the behavior of UCB algorithms with respect to over-exploration

**Weaknesses:**

- Significance: I am not 100% convinced about the real-life applications of this framework. If the “stopping” time is fixed because of budget limits (predetermined setting), I don’t see what ROBAI brings more than classical regret minimization (i.e., why playing pessimistically at the end of the sampling phase is better), especially since it is not feasible in practice, as the related algorithm requires the knowledge of Delta_min to be optimal.

**Questions:**

-  What does ROBAI bring more than classical regret minimization (i.e., why playing pessimistically at the end of the sampling phase is better) in the predetermined stopping time setting?

**Limitations:**

This paper deals with theoretical work, and does not raise significant concerns about negative impacts.

---

> ### Author Rebuttal · Authors · 2023-08-08
>
> We thank the reviewers for the precious time they spent on reviewing our paper. We discuss the points raised by the reviewer below.
>
> ### Clarification on Pre-determined Stopping Time and Fixed-Budget ###
>
> We thank the reviewer for raising this question, and we would like to make a clarification on how the pre-determined stopping time setting is different from the fixed budget setting usually considered in the literature. The stopping time $T_c$ in our setting is chosen based on the parameters of the problem instead of being enforced by the environment. Generally, our algorithm is not designed for the fixed-budget setting because EOCP has to decide the stopping time based on the problem parameter. The goal of fixed budget is to find the best arm as accurately as possible with a given budget which does not adapt to the problem parameters. It is not clear whether our algorithm makes optimal decisions when the budget is different from the stopping time we need to have. For example, if the stopping budget $T_c$ is smaller than our stopping time then the algorithm may have to explore more aggressively than EOCP since the commitment needs to be made much earlier. However, if $T_c$ (the budget) is larger than the stopping time, then our algorithm can be used to stop earlier without using the full budget to achieve $T^{-1}$ confidence. It is an interesting question whether we should use the remaining budget for exploration or commit without further exploration. As we can see from the numerical example, over-exploration hurts the system's performance. This somewhat counter-intuitive observation could be an interesting implication for the fixed budget setting.
>
> ### Comparison to Regret Minimization in Pre-determined Stopping Time Setting ###
>
> We would also like to make a clarification on how ROBAI is different from classic regret minimization. In ROBAI, we care about the regret performance for the total horizon $[1,T]$, instead of just the regret before the pre-determined stopping time $[1:T_c]$. If there is no commitment requirement, the problem is same as regret minimization with horizon $T$. However, since ROBAI requires the algorithm to commit as soon as possible while maintaining asymptotic optimal regret, and also note that EOCP uses $T_c = \mathcal{O}(\log T)$ which is much smaller, the agent must identify the optimal action with sufficiently large confidence ($T^{-1}$) to have good regret in commitment. Playing pessimistic actions after sampling phase ensures selecting the best arm for commitment with this high probability. This guarantee cannot be achieved by simply playing an UCB action in commitment (optimistically selecting the arm which has highest UCB). We conducted an empirical experiment to compare the accuracy of different commitment strategies:
> ```
> | Algorithm                | Number of committing wrong arm | Accuracy |
> | ------------------------ | ------------------------------ | -------- |
> | EOCP                     | 11                             | 0.000011 |
> | UCB Explore + UCB Comm   | 17293                          | 0.0173   |
> ```
>
> It can be witnessed that LCB provides better accuracy than simple UCB commitment which ensures low regret. ROBAI with time horizon $T$ is also different from regret minimization with horizon $T_c$ if $T_c$ is fixed due to budget limits. For regret minimization problems with horizon $T_c$, the theoretical regret limit is $\mathcal{O}(\log T_c) = \mathcal{O}(\log \log (T))$, which is much smaller than ROBAI with time horizon $T$. To achieve this limit, the UCB algorithm should choose a different optimistic bonus (a bonus $b_t(a) = \sqrt{\frac{2l}{N_t(a)}}$ with an exploration function $l$ in the order of $\log T_c$ instead of $\log T$). With this much smaller bonus, even though the regret of this algorithm will match $\mathcal{O}(\log \log (T))$, it will not be possible to identify the best arm with $T^{-1}$ confidence at time-step $T_c$ (even with LCB commitment), since the sub-optimal arms have not been explored enough and the empirical reward estimation is not accurate enough. So, ROBAI requires more aggressive exploration compared to regret minimization with horizon $T_c$, but it is less agressive than regret minimization with horizon $T$ as there is an interesting over-exploration phenomenon shown in our Fig.1.
>
> We hope that our response addresses the reviewer's concerns about the the contributions of ROBAI, and we are happy to answer any additional questions or concerns.

---

> > ### Comment · Reviewer_SqBM · 2023-08-14
> > **Response to the rebuttal**
> >
> > I thank the authors for their detailed answers. My apologies, I had misunderstood the pre-determined stopping time setting and the differences with regret minimization. However, the answer to the former question raises a concern (already mentioned by Reviewer T4df in Weakness #2 and left unaddressed in the rebuttal) of practicality due to the necessary knowledge of the minimal gap for that specific setting. As such, I will keep for now the score as it is.

---

> > > ### Author Response · Authors · 2023-08-15
> > >
> > > We are glad that our response clarifies the difference between the two settings. For the concern the reviewer raised (and the weakness #2 by reviewer T4df),  we would like to comment that our algorithm only needs a value that is no larger than the minimum gap $\Delta_{\min}$ instead of the exact value. While it is not completely "model-free", it is much easier and practical than knowing the exact value of $\Delta_{\min}$. For example, we don't need to use binary search routines to estimate the exact value of $\Delta_{\min}$ as suggested by Reviewer T4df. In practice, the agent can choose a small number $\epsilon$ to replace $\Delta_{\min}$ in the algorithm. For any problem where $\Delta_{\min}$ is larger than $\epsilon$, the algorithm will be regret optimal and commits in $\mathcal{O} (\log T)$ rounds. In order to have an algorithm that works for all $\Delta_{\min},$ we proposed the adaptive stopping time setting and Algorithm 2, which however has a higher sample complexity.  We thank the reviewer again for the positive review and quick response!

---

### Official Review · Reviewer_T4df · 2023-07-06

**Soundness:** 3 good
**Presentation:** 3 good
**Contribution:** 2 fair
**Rating:** 6
**Confidence:** 3

**Summary:**

This paper studies the classical multi-armed bandits problem with the goal to design algorithms achieving tight regret bound and tight sample complexity to identify the best arm simultaneously. To this end, three algorithms are proposed based on upper confidence exploration and lower confidence commitment for three settings: Gaussian bandits with known and unknown suboptimality gap $\Delta_{\min}$, general bandits when suboptimality gap (in terms of KL divergence) is known. It is proved that all the three algorithms enjoy tight regret bound up to the constant in front of the dominating term ($\log T$), and they also have polylog sample complexity scaled linearly with $1 / \Delta_{\min}^2$. There is also a lower bound on the sample complexity given the regret to have a tight constant in the $\log T$. The lower bound states that the rate of sample complexity is tight when $\Delta_{\min}$ is given, and can also be tight when $\Delta_{\min}$ is unknown as long as the order of remaining terms in the regret is smaller than $\log T$.

**Strengths:**

1.The paper addresses the classical settings of regret minimization and best arm identification problems in bandit theory. The investigation of their combination opens up new avenues for fundamental methods in this field. The paper stands out for its clean and clear presentation, making it accessible to readers and facilitating understanding of the proposed techniques.

2.The paper introduces several novel techniques that enable the simultaneous bounding of regret and sample complexity. The incorporation of lower confidence bound (LCB) commitment following upper confidence bound (UCB) exploration is a particularly interesting departure from the traditional approach of UCB commitment. This novel technique offers fresh insights and potential improvements in bandit algorithms. Additionally, the stopping criterion employed in the unknown gap setting is highly appreciated in theory by leveraging the deep connections between the number of pulls across different arms and the total pulls of suboptimal arms.

3.The analysis of lower bounds in the paper sheds light on the fundamental limits and trade-offs involved in identifying the best arm. These findings provide valuable insights into the design of algorithms, emphasizing the need to balance regret and sample complexity. The paper's techniques offer the potential for achieving a near-perfect balance between these two factors in practical applications of multi-armed bandits.


**Weaknesses:**

1. The unified achievement of both optimal regret and optimal sample complexity is an important problem in theory and application. However, I am not fully convinced of the motivation of this paper to study the constant-level tight regret bound but rate-level tight sample complexity. It somehow looks like a beam search of different settings in multi-armed bandits that are not fully addressed, since algorithms with both rate-optimal regret and rate-optimal sample complexity have been extensively studied. The vanilla multi-armed bandits is a fundamental model in theory, and the algorithms of multi-armed bandits provide prominent insights in many other problems. However, the UCB and LCB algorithms, which are conceptual in nature, require significant domain-dependent modifications to be applicable in real-world scenarios. This lack of naturalness in the setting detracts from the paper, despite the appreciation for the introduction of new techniques.

2. Two of the three proposed algorithms require the minimal gap to be known, which I believe is not very possible in application (except for the synthetic task). There are two reasons for it. Consider some tasks that aim to identify the user's favorite items: the expected reward is 1 if the item is liked by the user, and 0 otherwise, with some noise to model random attributes of the user. In this case, the gap is known and large, which greatly simplifies the problem. In general, it takes a very short period of time to find the favorite items even in the presence of noises, so it is totally acceptable that the regret is rate-optimal instead of constant-optimal. When the gap is not determined manually and small, it is not likely one can obtain the exact value of the gap. People often guess this gap by some binary search routines. However, the Algorithm 2 suffers a $\log^2 T$ sample complexity, where $\log T$ is the dominating term. This means the overall sample complexity of this algorithm is even not rate-optimal, though the regret is constant-optimal. It would be better in this case to use other methods such as elimination to keep both regret and sample complexity rate-optimal.

3. The experimental results presented in the paper have limitations in their scope. Firstly, comparing and plotting cumulative rewards instead of regret would provide a more convincing demonstration, as the sum of rewards and regret is a linear function rather than a constant. Additionally, the experiments should be conducted in more diverse environments, including scenarios with a higher number of arms. The current results do not suggest a significant improvement over the vanilla UCB algorithm, raising questions about the practical advantage of the proposed EOCP and its variants (see the questions below).


**Questions:**

1. Why you only consider the setting where the regret should be constant-optimal while the sample complexity can be rate-optimal. From the perspective of theory, is it possible to obtain both constant-optimal algorithm or constant-optimal sample complexity and rate-optimal regret?

2. How is the identification accuracy of the vanilla UCB algorithm at the step it gets the same regret of EOCP? Is this accuracy comparable to the commitment accuracy of EOCP?

**Limitations:**

See above.

---

> ### Author Rebuttal · Authors · 2023-08-08
>
> We thank the reviewers for the precious time they spent on reviewing our paper. We discuss the points raised by the reviewer below.
>
> ### Motivation of Constant-level Tight Regret but Rate-level Tight Sample Complexity And Contributions ###
>
> Whether it is necessary for constant-level tight regret bound or we could just be satisfied with rate optimal regret bound depends on the application. In some applications it is better to use elimination based algorithms to maintain rate optimality for both measures. However, the applications that motivates our research are those where reward performance is more important than commitment time, even though commitment time is also important. One conceptual example would be investment strategies, where the reward is the profit of a strategy and the commitment time is the time that the company decides on the strategy. In this case, a commitment needs to be made as early as possible but the profit is more important so it makes sense to take more time to make a better decision. The same also applies to people's occupation choices, where the potential career achievement and the commitment time is when one figures out which career he/she wants to pursue. It makes sense that achieving life goals is more important than choosing one's major early. We believe in these cases, rate optimality in regret is not satisfactory enough. The trade-off between constant-level regret optimality and commitment time we discovered in our paper shed lights on these applications. On the other hand, even though the upper bound of Algorithm 2 is $\log^2 T$, the empirical stopping time is quite close to Algorithm 1 (Fig.2). In general, we don't necessarily disagree with the the comment raised in Weakness 2, but again, which regime is the best model to study is application dependent.
>
> Building upon the fundamental MAB model, one major contribution of our paper is to consider three fundamental questions in online learning in a single model: how to explore? when to commit? which arm to choose? Through our stylized model, we demonstrate that these three questions are fundamentally related. In the current literature, the first question is studied under online MAB, the third question is considered under offline MAB and the stopping time was not well studied directly. We are hoping our model motivates the research interest in looking at these three questions together instead of individually.
>
> ### Both Constant Optimality in regret and sample complexity? ###
>
> It is possible to achieve constant optimal sample complexity with rate optimal regret. Take Gaussian 2-armed bandit for example, this can be done by an explore-then-commit algorithm which uniformly samples two arms and starts commitment once enough statistical separation can be witnessed from the past samples with confidence $T^{-1}$, i.e., BAI-ETC [20]. Its sample complexity is asymptotically $8\log(T)/\Delta$ which is constant optimal (compared to Theorem 1 [34]), and its regret is order optimal (Theorem 5 [20]). It is not possible to achieve constant optimality for both, as they require different trade-offs in exploration and exploitation. According to (Theorem 1 [34]), any algorithm to achieve constant optimal sample complexity in Gaussian 2-armed bandits should pull the two arms equal likely in exploration in order to stop early, and this aggressive strategy completely ignores the exploration-exploitation trade-off to maintain low regret. Therefore, it incurs regret asymptotically larger than UCB algorithms (rigorously shown in [20]). Our paper is dual to [20], in the way that [20] contributes to the regret performance of algorithms with constant-level optimality in sample complexity, while we contributes to the sample complexity of algorithms with constant-level optimality in regret.
>
> ### Identification Accuracy of Vanilla UCB algorithm ###
> The vanilla UCB algorithm does not output a best arm for us to commit to, and our proposed algorithm's exploration strategy is in fact the UCB algorithm. One could modify the vanilla UCB algorithm to stop at a certain time, and then makes a decision on the best arm, just like we did in our paper. The accuracy in this case will depend on the design of the stopping time and the decision rule. We remark that our major contribution in terms of the algorithm is to design such a stopping rule and an action identification rule using LCB. We illustrates the improvement of LCB by comparing EOCP to an algorithm which uses the UCB exploration strategy and the same stopping time as our algorithm (which would results in the same regret in exploration), but uses UCB to identify the best arm instead of LCB. The accuracy is compared below:
>
> ```
> | Algorithm                | Number of committing wrong arm | Accuracy |
> | ------------------------ | ------------------------------ | -------- |
> | EOCP                     | 11                             | 0.000011 |
> | UCB Explore + UCB Comm   | 17293                          | 0.0173   |
> ```
> Time horizon is $10^3$ and experiment is done over $10^6$ iterations. It can be witnessed that LCB provides better accuracy than simple UCB commitment and UCB commitment does not satisfy the required $T^{-1} = 10^{-3}$ accuracy.
>
> ### Numerical Experiments with More Arms ###
> We conducted an experiment in a $4$-armed bandit model to compare our algorithms to existing algorithms in the literature, including an Action Elimination algorithm. The results are presented in our submitted rebuttal pdf file. In general, the observations we see in 2-armed bandit models can still be witnessed in this $4$-armed bandit model, which demonstrates the regret improvement of our algorithm and the over-exploration phenomenon is not unique to the 2-armed case.
>
> We hope that our response addresses the reviewer's concerns are happy to answer any additional questions or concerns. We would also be grateful if the reviewer could consider reevaluating the review and rating based on our response.

---

> ### Author Response · Authors · 2023-08-11
>
> Dear Reviewer T4df:
>
> We want to follow up to see whether our response addresses your concerns. Please don't hesitate to let us know if you have any other questions/comments. Thanks! We also want to comment on weakness #2 raised by the Reviewer.
>
> ### Require Known Minimum Gap  ###
>
> For the weakness #2 raised by the Reviewer,  we would like to comment that our algorithm only needs a value that is no larger than the minimum gap $\Delta_{\min}$ instead of the exact value. While it is not completely "model-free", it is much easier and practical than knowing the exact value of $\Delta_{\min}$. For example, we don't need to use binary search routines to estimate the exact value of $\Delta_{\min}$ as suggested by the Reviewer. In practice, the agent can choose a small number $\epsilon$ to replace $\Delta_{\min}$ in the algorithm. For any problem where $\Delta_{\min}$ is larger than $\epsilon$, the algorithm will be regret optimal and commits in $\mathcal{O} (\log T)$ rounds. In order to have an algorithm that works for all $\Delta_{\min},$ we proposed the adaptive stopping time setting and Algorithm 2, which however has a higher sample complexity. But on the other hand, according to our Theorem 3 in adaptive stopping time setting, it is not possible to maintain constant optimal regret and commit at $\mathcal{O} (\log T)$ rounds at the same time.

---

> > ### Comment · Reviewer_T4df · 2023-08-16
> >
> > I thank the authors for your efforts to address my concerns. I really appreciate the additional experimental results that illustrates the effectiveness of LCB commitment strategy in a UCB algorithm. I do agree this is a technically solid theory paper studying an important fundamental problem in bandits, so I decided to slightly raise my score. However, I am still concerned with the motivation and setting of the paper. The UCB (or LCB) is a conceputal algorithm useful in theory, but the industry uses a much more complicated algorithm pipeline for their purpose based on the principle of UCB (or LCB). Given the motivating example of the paper, I am not sure about why requiring the regret to be constant-level optimal is so important **in a theory paper**. Moreover, the reply to the Weakness #2 seems to contradict with the purpose of the paper. Say, you have an $\epsilon$ in your algorithm that equals $\Delta_{min}/2$, then the regret would not be constant-level optimal anymore, since now your have a dependency like $2\log(T)/\epsilon = 4\log(T)/\Delta_{min}$. I hope the reviewer to give some intuitions on how to decide the $\epsilon$ to keep the regret to be constant-level optimal.

---

> > > ### Author Response · Authors · 2023-08-16
> > >
> > > We are glad that our additional experimental results address the concerns. We thank the reviewer for the additional comments and for raising the score.
> > >
> > > We would like to clarify that substituting $\Delta_{\min}$ with $\epsilon$ will not affect the regret performance in Theorem 1. Namely, with $\epsilon$ smaller than $\Delta_{\min}$ in the algorithm, the regret performance remains to be $2\log (T) / \Delta_{\min}$. However, the sample complexity result (Corollary 1) will be affected. Instead of having $\mathcal{O} ( \log(T)/\Delta_{\min}^2)$ commitment time, it becomes $\mathcal{O}(\log(T)/\epsilon^2)$. The regret is still constant-level optimal.

---

### Official Review · Reviewer_WGiE · 2023-07-07

**Soundness:** 3 good
**Presentation:** 3 good
**Contribution:** 3 good
**Rating:** 6
**Confidence:** 3

**Summary:**

The work studies how to design an algorithm with an asymptotically optimal regret rate such that it will also commit to the best arm with high probability after a stopping time (e.g., O(logT)). The paper proposes two algorithms in the Gaussian bandits setting: one with pre-determined stopping time (EOCP), which requires the knowledge of the minimum reward gap, and another with adaptive stopping time (EOCP-UG). The authors prove both algorithms are asymptotically optimal and will commit to the best arm with confidence O(1/T) after O(log(T)) and O(log^2(T)) respectively. The paper further shows the corresponding lower bounds for commitment times (expected stopping time) and finds EOCP is sample optimal and EOCP-UG is nearly sample optimal. In addition, the authors extend the EOCP to KL-EOCP for general bandits, whose commitment time also matches the lower bound.


**Strengths:**

1) In general, the studied question is interesting and the paper is well-written and easy to follow.

2) The proposed algorithms are novel which are based on well-designed stopping rules.

3) The theoretical results and the proofs are clear. The numerical results also show the advantages of the proposed algorithms compared to the benchmark algorithms, including UCB and BAI algorithms in the literature.


**Weaknesses:**

1) The problem setting is the fixed-confidence BAI setting (note for the setting of EOCP, the stopping time is pre-determined by the algorithm instead of by the environment). It would be great if the authors can give some discussions on the fixed budget setting (the stopping time is given), like the challenges of extending the proposed algorithms.

2) In the general bandits setting, the authors only extend EOCP to KL-EOCP, which needs the KL_min of arms. The authors demonstrate if KL_min is not known, an extension of EOCP-UG can deal with this setting.  But there is no pseudo-code or provable results on it. More details or discussions would be helpful.

Minor comments:

In Table 1, it might be helpful to add another column ‘Setting’, since ‘Optimality’ is a bit confusing for explaining ‘Gaussian’ or ’General’.


**Questions:**

See above.

**Limitations:**

Na.

---

> ### Author Rebuttal · Authors · 2023-08-08
>
> We thank the reviewers for the precious time they spent on reviewing our paper. We discuss the points raised by the reviewer below.
>
> ### Fixed-budget Setting ###
>
> We thank the reviewer for the great question! Generally, our algorithm is not designed for the fixed-budget setting because EOCP has to decide the stopping time based on the problem parameters. For fixed budget, the goal is to find the best arm as accurately as possible with the given budget. The budget does not adapt to the problem parameters but the decision and learning process do. The challenge of extending our algorithm to the fixed budget setting is that it is not clear whether our algorithm makes optimal decisions when the budget is different from the stopping time we need to have.
>
> For example, in a two-armed Gaussian bandit example, if the stopping budget $T_c$ is much smaller than our stopping time, then the algorithm may have to explore more aggressively than EOCP since the commitment needs to be made much earlier. However, if $T_c$ (the budget) is larger than the stopping time, then our algorithm can be used to stop early without using the full budget. It is an interesting question whether we should use the remaining budget for exploration or commit without further exploration. As we can see from the numerical example in the paper, over-exploration hurts the system's performance. This somewhat counter-intuitive observation could be an interesting contribution (or at least an implication) to the fixed budget setting.
>
> ### KL-EOCP with Unknown Gap ###
>
> We provide a short psudo-code and theoretical guarantees below:
>
>
> ***
> > ### Algorithm: KL-EOCP-UG ###
>
> > 1: Initialize by pulling each arm once.
>
> >2: **While** $\max_a \min_a' N_{t-1}(a) - l N_{t-1}(a')\leq 1$ **do**
>
> >3: &nbsp;&nbsp;&nbsp;&nbsp; take action $A_{t+1} = \arg\max_a\mathsf{UCB}_{t-1}(a)$
>
> >5: **end while**
>
> >6: Let $T_c = t-1$, $\hat{a} = \arg\max_{a} \mathsf{LCB}_{T_c } (a)$
>
> >7: Commit to $\hat{a}$ for the rest of time horizon.
> ***
>
> where $\mathsf{UCB}_t(a)$ and $\mathsf{LCB}_t(a)$ are defined according to Line 4 and 7 in Algorithm 3. We can also derive the following theoretical guarantee for this algorithm which requires additional assumptions on the $b(\theta)$ function defining the KL-divergence (Line 270):
>
> ***
> > ### Aymptotic Optimality for KL-EOCP-UG ###
>
> >If we choose $l = \log(T) + 4\sqrt{2\log(T)}$, suppose $b(\theta)$ is strongly convex and smooth, the expected regret of the KL-EOCP-UG algorithm is asymptotically upper bounded by:
>
> >$\lim\sup_{T\to\infty} \frac{Reg(T)}{\log T} \leq \sum_{a:\Delta_a >0} \frac{\Delta_a}{\mathsf{KL}(\mu_a,\mu_1)}$
> ***
>
> ### Table 1 ###
>
> We will add another coloum in table 1 to make sure there is no confusion. We thank the reviewer for pointing it out.
>
> We hope that our response addresses the reviewer's questions regarding the fixed-budget setting and the KL-EOCP algorithm with unknown gaps, and we are happy to answer any additional questions or concerns. We would also be grateful if the reviewer could re-evaluate the rating and review based on our response.

---

> ### Author Response · Authors · 2023-08-18
>
> Dear Reviewer WGiE,
>
> We want to follow up to see whether our response addresses your concerns on fixed-budget bandits and algorithm in general bandit settings. Please don't hesitate to let us know if you have any other questions/comments. Thanks!

---

> ### Author Response · Authors · 2023-08-21
>
> Dear Reviewer WGiE,
>
> We want to follow up to see whether our response addresses your concerns. We are happy to answer any other questions/comments. Thanks!

---

### Official Review · Reviewer_tQKP · 2023-07-08

**Soundness:** 2 fair
**Presentation:** 3 good
**Contribution:** 2 fair
**Rating:** 3
**Confidence:** 4

**Summary:**

The paper delves into the study of the 'Explore Then Commit' (ETC) policy, where the algorithm is divided into two stages: exploration and commitment. During exploration, the algorithm is permitted to switch actions, while the commitment phase restricts the algorithm to pulling only the commit arm. The primary objective of the algorithm is to reduce the expected commit time, denoted as $T_c$, and ensure that at time $T_c$, $\hat a \neq a^*$ holds true with a probability of $O(1/T)$.

Three variations of the ETC policy are introduced in this paper: EOCP, EOCP-UG, and KL-EOCP.

For EOCP, it's structured for a known gap setting, achieving regret in the order of $2\log T/\Delta_i$. The authors assert that this outcome is asymptotically optimal. However, this claim appears to be incorrect. As per reference [On Explore-then-Commit Strategy], the asymptotic optimal regret for a known gap should be $\log T/(2\Delta_i)$. Furthermore, the authors have not discussed this setting's reference in a comprehensible manner. For instance, DETC with a known gap achieves exact asymptotic optimality.

For EOCP-UG, the authors show that it achieves asymptotic optimality with a commit time of $\log^2 T$.

As for KL-EOCP, it necessitates knowledge of certain parameters for the design of the pre-determined stopping time.

In summary, the claim that the EOCP and KL-EOCP algorithms are asymptotically optimal appears to be overstated. The upper bound for EOCP-UG and its associated lower bound is the same as DETC. As for the lower bound, the assumption seems excessively strong. The authors did not clearly indicate whether algorithms must adhere to the equation given in Line 241.

Primarily, this paper targets two-armed bandit problems, focusing on asymptotic regret. However, I am intrigued by the finite-time bound. I would appreciate an algorithm that is not only asymptotically optimal but also demonstrates a robust finite-time bound. In addition, the paper only considers a fixed horizon T setting. What about the case with an unknown $T$? From my understanding, DECT and its variant [Almost Optimal Anytime Algorithm for Batched Multi-Armed Bandits], are also suitable for an unknown T setting. Furthermore, the necessity of the assumption in Line 141 is not clearly articulated and requires clarification.

Regarding experimental results, the performance of ETC and DECT presented in this paper is inconsistent with previous papers. This discrepancy might be due to parameter adjustments in these algorithms. I would appreciate seeing empirical results with parameter tuning for both ETC and DETC.

----

# At the end of discussion phase

The authors fail to address my primary concerns. Below, I elaborate on these issues:

1. Line 174 of the manuscript asserts that in the pre-determined setting (where the gap $\Delta$ is known), $\frac{2\log T}{\Delta}$ is asymptotically optimal. This claim is misleading. Algorithm 4 in reference [21] shows that in the pre-determined setting with $T_c = \log^2 T$, there exists an algorithm with a regret of $\frac{\log T}{2\Delta}$. Moreover, the known lower bound (Theorem 6 in [20]) for known gap setting is $\frac{\log T}{2\Delta}$. Therefore, stating $ \frac{2\log T}{\Delta}$ as asymptotically optimal is an overstatement.

In their response, the authors attempt to equate the lower bound of the pre-determined setting with the lower bound of the known gap setting in the ETC strategy [20]. This comparison is problematic and has the potential to mislead other readers. Specifically, Table 1 in the authors' response erroneously claims that $4\log T/\Delta$ is the lower bound for the known gap setting, whereas this is actually the lower bound proven for the **unknown gap** setting in Theorem 4 of [20]. Additionally, they assert that $2\log T/\Delta$ is the lower bound for the known gap and pre-determined settings, which contradicts Algorithm 4 in [21] that shows regret $\frac{\log T}{2\Delta}$.


2. As I pointed out in my initial review, DECT [21] (Algorithm 5 and 2) can be directly applied to the unknown gap setting. The results for DECT in an unknown gap setting are identical to those presented in this paper.

Given the paper's significant overstatements and the paper's limited contributions, I am revising my score from 4 to 3.

**Strengths:**

In this paper, the authors propose an algorithm called Explore Optimistically then Commit Pessimistically (EOCP) to solve the Regret Optimal Best Arm Identification (ROBAI) problem. It first uses an optimistic modified UCB algorithm to explore actions with a slightly larger exploration function , and then commits to actions according to a pessimistic LCB algorithm when the exploration ends. The main contributions include designing new stopping rules with both pre-determined stopping time (vanilla EOCP) and adaptive stopping time (the EOCP-UG variant), which provably balance the trade-off between regret minimization and optimal action identification

**Weaknesses:**

Please refer to the Summary.

**Questions:**

Please refer to the Summary.

**Limitations:**

The main consideration of mine is that the proposed algorithm considers asymptotic bounds other than finite $T$ (probabily unknown). This limits the contribution of the paper heavily.

---

> ### Author Rebuttal · Authors · 2023-08-08
>
> We sincerely thank the reviewer for the comments/suggestions and included a detailed response below. We also would like to point out a couple of misunderstandings the reviewer had in the preliminary review (see 1 and 2 below).
>
> ### Limitation of Targeting Two-armed Bandits ###
>
> The main theorems (Theorem 1, 2, and 4) of our paper are for general MAB with more than 2 arms. Even though our Theorem 3 focuses on a 2-arm case, it can be generalized to models with more than two arms.
>
> ### Limitation of Asymptotic Regret and Lack of Finite-time Bounds ###
>
> Our asymptotic regret guarantees (Theorem 1,2 and 4) are derived from the finite-time bounds, specifically,  the bounds in Theorem 5, 6 and 7 in the complete version (included as the supplementary material of the original submission). We chose to keep the asymptotic results n our main body due to its simplicity but we are happy to add the finite-time bounds to the main body as well.
>
> ### Unknown $T$ ###
>
> Any algorithm for ROBAI should satisfy the two properties : (1) the termination time $T_c$ of exploration (the commitment time) is a stopping time (either pre-determined or depends on the samples before $T_c$). (2) the confidence of best arm is $T^{-1},$ in order to guarantee regret optimality. Theoretically, it is impossible to design an ETC algorithm satisfying the above two properties without knowing $T$, because no matter when the agent stops exploration and what confidence $\delta$ it holds, an adversary can always choose $T$ large enough such that $\delta> T^{-1/2}$, which violates property (2).
>
> The variant of DETC works for an unknown $T$ because its exploration termination time is not a valid stopping time. In their algorithm, they guess $T$ in epochs. In the $r$-th epoch, they assume $T=2^r$ and perform an exploration period with a commitment period. If after $t=2^r$, the interaction continues, they will re-guess $T=2^{r+1}$ and consider the commitment period in the $r$-th epoch as part of the exploration. In other words, the "commitment" continues to change in their algorithm. For our problem, once a commitment is made, the agent cannot change the decision. However, If we don't require $T_c$ to be a stopping time and allow the agent to change the commitment as in DETC,  it is straightforward to adapt our algorithms to the unknown $T$ setting using the exact same idea. The theoretical guarantees can also be straightforwardly extended.
>
> ### Overstating Asymptotic Optimality in Pre-determined Setting ###
>
> We respectfully disagree with the reviewer on this because the pre-determined setting is more difficult than the known gap setting studied in [20,21], and thus $\log T/2\Delta$ is not a tight lower bound.
>
> (1) Algorithms in the known gap setting, e.g., SPRT-BAI [20] and DETC-KG [21], use the exact value of $\Delta$ to design all sampling, stopping, and action decision rules so that the regret performance can be lower than $2\log T/\Delta$. Changing $\Delta$ to any other value (even a slight mismatch) would harm the performance. The is also implied by the proof of Theorem 6 from [20] that one would need the exact value of $\Delta$ to achieve $\log T / 2\Delta$ lower bound. Our algorithms in the pre-determined setting only requires a value smaller than $\Delta$ and only use it in stopping time. The theoretical regret performance would still be the same if we under-estimate $\Delta$.
>
> (2) In the known gap setting, the algorithm can use reward samples from exploration to design the stopping time, but in the pre-determined setting, $T_c$ must be pre-specified before the exploration starts.
>
> Based on the two comparisons above, we believe $\log T/2\Delta$ is not achievable in the pre-determined setting. In fact, this setting is more comparable to the fixed-design setting [20] whose regret lower bound is $4\log T/\Delta$. But it requires all actions to be pre-specified before exploration, which is harder. Overall, we believe $2\log T/ \Delta$ is a more valid lower bound since the algorithms do not entirely depend on $\Delta$, i.e., any hyper-parameter lower than $\Delta$ would suffice. We compare all scenarios as follows:
>
> ```
> | Setting                 | Lower Bound       | Knowledge of $\Delta$ (Where it is used)     | Stopping Rule |
> | ----------------------- | ----------------- | -------------------------------------------- | ------------- |
> | Fixed-Design            | 4 \log T / \Delta | Yes (Stopping Time)                          | Pre-determined|
> | Known Gap               | \log T/ 2\Delta   | Yes (Sampling, Stopping, Arm Identification) | Adaptive      |
> | Unknown Gap             | 2\log T/\Delta    | No                                           | Adaptive      |
> | Pre-determined Stopping | 2\log T/\Delta    | lower bound of $\Delta$ (Stopping Time)      | Pre-determined|
> | Adaptive Stopping       | 2\log T/\Delta    | No                                           | Adaptive      |
> ```
>
> ### Lower Bound Condition ###
>
> We require all ROBAI algorithms to satisfy the condition in line 241. It is not excessively strong because ROBAI focuses on asymptotic regret optimal algorithms, i.e., the regret dominating term is $2\log T/\Delta$. These algorithms naturally satisfy the condition in line 241 with a $c<1$.
>
> ### Empirical Results with Parameter Tuning  ###
> We tune the parameters for BAI-ETC and DETC, and the results are presented in our uploaded pdf file. After tuning $T_1$ and the confidence bounds, the performance of DETC beats BAI-ETC and almost matches UCB, but not our algorithms.
>
> Given we indeed provide general results for bandits with more than two arms and also finite-time bounds for all algorithms in the original paper, which are the main limitations the reviewer is concerned about, we would appreciate it if the reviewer could re-evaluate the rating and review based on our response. We are happy to address any additional questions and concerns.

---

> > ### Comment · Reviewer_tQKP · 2023-08-21
> >
> > I'm skeptical about the claim of asymptotic optimality in the pre-determined setting as presented in Line 174. In Algorithm 1, the gap, $\Delta$, is utilized to define the parameter $T_c$. This leads me to question the assertion that EOCP is asymptotically optimal. A more fitting comparison would be with the known gap lower bound rather than the unknown gap lower bound.
> >
> > In their response, the authors draw parallels between this setting and the fixed-design setting referenced in [20], where the regret lower bound is given as $\(4\log T/\Delta\)$. I find this comparison misleading for several reasons:
> >
> > 1. The ETC strategy in [20] ensures that both arms are pulled an equal number of times up until the point of commitment. Contrarily, the algorithms examined in this paper allow for varying numbers of pulls for each arm. As a result, the lower bound outlined in [20] for ETC isn't applicable to the present study. Invoking the result $\(4\log T/\Delta\)$ in this context is not justifiable.
> >
> > 2. Algorithm 1 inherently caters to the known gap setting since it mandates the specification of $T_{c}$ based on the known gap, $\Delta$. In contrast, $\(4\log T/\Delta\)$ is recognized as the lower bound for the ETC strategy with an unknown gap.

---

> > > ### Author Response · Authors · 2023-08-21
> > >
> > > We thank the reviewer for the response, and we want to point out the following misunderstanding the reviewer had regarding our rebuttal and our paper.
> > >
> > > ### Utilizing $\Delta$ in Designing Parameter $T_c$ ###
> > > We want to point out that our algorithm **only requires a lower bound of $\Delta$** to define $T_c$ instead of the exact value. While it is not completely "model-free", it is much easier and practical than knowing the exact value of $\Delta$. This lower bound does not provide all the information that $\Delta$ has. In practice, the agent can choose a small number to replace $\Delta$ in the algorithm. For any problem where $\epsilon$ is larger than $\Delta$, the algorithm will have exactly the same performance as in Theorem 1 and Corollary 1. On the other hand, ETC strategy in [20] in the known gap setting requires **the exact value of $\Delta$**, and replacing it with any value other than $\Delta$ would either harm the best arm identification accuracy or harm the sample complexity and regret. Based on this essential difference and the fact that our algorithm does not require the exact value of $\Delta$. We consider it not fair to compare our algorithm 1 to the lower bound $\log T/ 2\Delta$.
> > >
> > > ### Comparison to the Fixed-Design Setting ###
> > > It is true that in the fixed-design setting, all arms need to be pulled uniformly, and the commitment time is pre-determined, which results in the $4\log T/ \Delta$ regret. The reason why our algorithm 1 can achieve lower regret than $4\log T/ \Delta$ is due to the adaptive arm sampling rule. However, our design of commitment time is still pre-determined. We said that our pre-determined setting is more similar to the fixed-design setting, instead of the known gap setting. Based on the reviewer's argument, we found the known gap setting lower bound $\log T/2\Delta$ is as not applicable to the present study as the lower bound $4\log T/\Delta$ in fixed-design setting. The full comparsion of all settings are presented in the original rebuttal.
> > >
> > > Given we indeed provide general results for bandits with more than two arms and also finite-time bounds for all algorithms in the original paper, which are the main limitations the reviewer is concerned about, we would appreciate it if the reviewer could re-evaluate the rating and review based on our response.

---

> ### Author Response · Authors · 2023-08-11
>
> Dear Reviewer tQKP:
>
> We want to follow up to see whether our response addresses your concerns. Please don't hesitate to let us know if you have any other questions/comments. Thanks!

---

> ### Author Response · Authors · 2023-08-18
>
> Dear Reviewer tQKP:
>
> We want to follow up to see whether our response addresses your concerns and we are happy to answer any additional questions/comments. Thanks!

---

> ### Author Response · Authors · 2023-08-21
>
> Dear Reviewer tQKP:
>
> We want to follow up to see whether our response addresses your concerns. We are happy to answer any additional questions/comments. Thanks!

---

### Author Rebuttal · Authors · 2023-08-08

We thank the reviewers for their precious time spent on reviewing our paper. To address the questions and concerns raised in the preliminary reviews, we present additional numerical results in the uploaded pdf File. Due to the limited time, we can only provide results with Gaussian bandits, and the results for Bernoulli bandits will be added to the revision.

In the PDF file, Fig.1 is a comparison of EOCP with existing algorithms in the literature under bandit models with more than 2 arms, and Fig. 2 is a comparison of EOCP with the tuned versions of DETC and BAI-ETC. Numerical results show that EOCP and its variants still outperform existing algorithms, with a similar trend as shown in Fig.1 in our original paper. The extended results demonstrate EOCP's ability to generalize to more complex settings.

We hope that our response addresses the reviewers' questions regarding the numerical results of EOCP, and we are happy to answer additional questions and concerns.

---

### Decision · Program_Chairs · 2023-09-21

**Decision:**

Accept (poster)

**Comment:**

This is a paper that simultaneously tackles two fundamental problems in bandit theory -- regret minimization (and committing to an action after a pre-determined stopping time) and best arm identification. Both have been traditionally studied separately, but this paper brings them under the same analytical framework. Reviewers were generally in praise of the theoretical importance of the problem and also the novel use and combination of traditional technical tools. The results are clean and easy to interpret and collectively form a useful contribution to bandit theory. Committing to an action after a fixed stopping time also seems to have some interesting practical applications, but this is of secondary importance to the theoretical contributions contained herein.